# Interferon-γ selectively promotes survival of alveolar progenitor cells in a human lung organoid model

Antonella F M Dost [ID][1,2,✉], Katarína Balážová [ID][1,2,8], Carla Pou Casellas[1,3,8], Lisanne M van Rooijen[1,2], Wisse Epskamp [ID][1,2], Gijs J F van Son [ID][4], Willine J van de Wetering [ID][5], Carmen Lopez-Iglesias[5], Harry Begthel [ID][1,2], Peter J Peters[6], Niels Smakman[7], Johan H van Es[1,2] & Hans Clevers [ID][1,2,4✉]

## Abstract

**Disease of the lung alveoli is frequently associated with acute or chronic inflammation. At present, there are no effective therapies to support regeneration of the alveolar epithelium, and ongoing inflammation adds an additional layer of complexity to many lung diseases. Here, we describe a primary adult human organoid model for investigating how inflammation shapes alveolar regeneration. Unlike previous models, this system supports long-term expansion of newly identified human-specific alveolar progenitor cells and serum-free differentiation into alveolar type 1 (AT1)-like cells. Using this platform, we find that interferon-gamma (IFN-γ) exerts cytotoxic effects on mature AT1-like cells while promoting survival of alveolar progenitor cells mediated by BIRC3. This unexpected selective positive effect of IFN-γ on alveolar progenitors underscores the need for nuanced and context-dependent evaluation of the influence of pro-inflammatory cytokines on alveolar regeneration. Our organoid model provides a reductionist platform for mechanistic studies and discovery of strategies to enhance alveolar regeneration.**

**Subject Categories** Development; Respiratory System; Stem Cells & Regenerative Medicine

## Introduction

Lung diseases are among the leading causes of death worldwide, with rising incidence (Institute for Health Metrics and Evaluation, 2019). The primary risk factor for developing lung diseases such as interstitial lung disease or chronic obstructive pulmonary disease (COPD) is chronic exposure to airborne toxic particles, such as cigarette smoke, air pollutants, dust, fumes, and chemicals (Antuni

and Barnes, 2016; Vasarmidi et al, 2025). The pathophysiology of chronic lung diseases are complex and poorly understood, and often involves airway obstruction, extracellular matrix remodeling, mucociliary dysfunction, fibrosis, chronic inflammation, or emphysema, the destruction of the alveolar region (Hogg and Timens, 2009). These features can persist even after exposure cessation, making diseases such as COPD a progressive disease with no cure. Available treatments, such as bronchodilators and corticosteroids, can alleviate shortness of breath and reduce inflammation. However, inhibiting the inflammatory response can increase the susceptibility to pulmonary infections, a serious complication that can lead to disease exacerbation. These exacerbation periods are characterized by an acute worsening of COPD symptoms and accelerated lung function decline, representing an important window for therapeutic intervention (Wedzicha and Seemungal, 2007). Understanding the nuanced interplay between the pro-inflammatory cues present during exacerbations and lung epithelial cells is therefore critical. To date, there are no therapies that succeed in protecting the alveolar tissue and improving alveolar regeneration in COPD and other lung diseases (Ng-Blichfeldt et al, 2019).

Lung alveoli are the gas-exchanging units of the lungs. Two major epithelial cell types exist within the alveoli: alveolar type 2 (AT2) and type 1 (AT1) cells. AT2 cells produce pulmonary surfactant that lines the alveolar epithelium. This surfactant, comprised of lipids and proteins, prevents collapsing of the alveoli by decreasing surface tension and plays a role in innate immunity. AT1 cells are exceedingly thin and spread out, forming part of the air–blood-barrier where gas exchange occurs. Not much is known about how human lungs regenerate, as our research relies heavily on mouse models. While AT2 cells are generally thought of as stem cells that can self-renew and differentiate to AT1 cells (Barkauskas et al, 2013; Nabhan et al, 2018; Zacharias et al, 2018), recent advances in single-cell RNA sequencing (scRNA-Seq) have helped define new cell types or cell states that likely play a role in alveolar regeneration. Most noteworthy are the discovery of the AT2-signaling (AT2-s) population (Travaglini et al, 2020), and cells residing at the terminal and respiratory bronchioles (TRB), termed pre-terminal bronchioles (pre-TB) secretory, TRB-specific alveolar

[1]Hubrecht Institute - Royal Netherlands Academy of Arts and Sciences and University Medical Center Utrecht, Utrecht, The Netherlands. [2]Oncode Institute, Utrecht, The Netherlands. [3]Department of Nephrology and Hypertension, University Medical Center Utrecht, Utrecht, The Netherlands. [4]The Princess Máxima Center for Pediatric Oncology, Utrecht, The Netherlands. [5]Microscopy CORE Lab, Faculty of Health, Medicine and Life Sciences, Maastricht University, Maastricht, The Netherlands. [6]The Maastricht Multimodal Molecular Imaging Institute, Maastricht University, Maastricht, The Netherlands. [7]Department of Surgery, Diakonessenhuis, Utrecht, The Netherlands. [8]These authors contributed equally: Katarína Balážová, Carla Pou Casellas. ✉E-mail: Antonella.Dost@sund.ku.dk; H.Clevers@hubrecht.eu

type-0 (AT0) (Kadur Lakshminarasimha Murthy et al, 2022), and respiratory airway secretory (RAS) cells (Basil et al, 2022). One commonality of those populations is that they express lower levels of canonical AT2 markers and higher levels of secretory markers such as secretoglobin family 3 A member 2 (SCGB3A2) as compared to the bulk AT2 population. This is in line with alveolar regeneration in murine models, where bronchioalveolar stem cells and club cells can repair the alveolar epithelium (Kim et al, 2005; Liu et al, 2019; Salwig et al, 2019; Liu et al, 2020). Moreover, both in mouse models and in humans, it has been shown that an AT2-AT1 transitional cell state emerges during injury repair (Konkimalla et al, 2023; Strunz et al, 2020; Kobayashi et al, 2020; Choi et al, 2020). Compared to AT2 cells, AT0 cells have higher expression of AT1 markers, placing them between AT2, AT1, and secretory cells (Kadur Lakshminarasimha Murthy et al, 2022). How these independently identified populations relate to each other is not completely understood. Even less understood are the roles of these populations and the mechanisms of alveolar regeneration in disease contexts such as emphysema, a knowledge gap that is hindering the development of new therapies. To address this unmet therapeutical need, it is important to develop human models that accurately recapitulate alveolar regeneration in the context of inflammatory disease.

Organoids are three-dimensional structures derived from stem cells grown in basement membrane extract (BME), enabling us to study the stem cell biology of untransformed human cells in vitro. Organoids are grown in conditions that promote proliferation and stemness. Because the lungs are quiescent during homeostasis, lung organoid cultures can be seen as an injury model that recapitulates regeneration rather than homeostasis. In recent years, primary feeder-free human adult alveolar organoid (ALVO) models have emerged, greatly advancing our toolbox to study alveolar regeneration (Katsura et al, 2020; Salahudeen et al, 2020; Youk et al, 2020). However, these models have a few drawbacks. Firstly, they rely on positive selection of AT2 cells from whole lung tissue using the surface marker HTII-280. This strategy likely excludes some of the progenitor cells that are capable of alveolar regeneration, such as RAS cells. HTII-280 negative AT2 cells are observed in healthy human lungs and can give rise to AT2 cell-containing organoids (Evans and Lee, 2020; Basil et al, 2022). Secondly, the protocols rely on the addition of serum to generate AT1 cells and hence do not provide defined media conditions, a crucial requirement for controlled disease modeling and mechanistic studies. In this study, we developed a robust human ALVO model to overcome these challenges, and we used it to discover that the infection-related cytokine interferon-gamma (IFN-γ) has distinct and opposite effects on alveolar epithelial cell populations.

## Results

### Optimized ALVO conditions with HRG1 allow for long-term expansion in defined media

Current ALVO culturing protocols rely on positive selection of HTII-280 + AT2 cells (Fig. 1A). However, it is unknown whether this strategy excludes cells that are important for alveolar regeneration. To avoid excluding this population in our ALVO

cultures and to maximize cell outgrowth, we deviated from existing protocols (Fig. 1A). We processed distal non-tumor human lung tissue from lobectomies of lung cancer patients into near single-cell suspensions and plated all cells in submerged, serum-free organoid conditions. The omission of cell sorting shortens the processing time of the tissue and puts less stress on the cells. Furthermore, the presence of non-epithelial cells at passage (p)0 may support organoid outgrowth. Our p0 ALVO-media (ALVO-p0) is similar to the published SFFF media (Katsura et al, 2020) with the addition of HRG1 (Table EV1). After 14 days in p0, large organoids with either alveolar (SFTPC) or airway (KRT5) markers had formed (Fig. EV1A). Unlike organoids grown in airway organoid (AO) media (Sachs et al, 2019), the KRT5+ organoids appeared dense and did not express the club cell marker SCGB1A1 (Sachs et al, 2019). Next, we set out to test whether HT2-280 positive selection would lead to exclusion of alveolar-like cells. Moreover, we tested the role of the media composition on alveolar cell outgrowth. We enriched for epithelial cells and excluded airway progenitor cells by selecting for organoid-derived cells that expressed epithelial cell adhesion molecule (EPCAM) but not nerve growth factor receptor (NGFR) (Rock et al, 2009; Wijk et al, 2021) using fluorescence-activated cell sorting (FACS). We then gated for HT2-280+ (HT2+) and HT2-280- (HT2-) cells (Fig. EV1B) and plated the two populations into media that either contained HRG1 or EGF (Fig. EV1C). This contrasts with our ALVO-p0 media, which contained both. HRG1/NRG1 and its receptors ERBB3 and ERBB4 are expressed in lung alveolar cells, similar to EGF and its receptor EGFR/ERBB1 (Fig. EV1D). Moreover, HRG1 has been shown to have a positive effect on the growth and survival of AOs (Liu et al, 2024b). In accordance with this observation, the organoids grown in HRG1-containing media appeared larger and in higher numbers compared to EGF (Fig. EV1C). Moreover, the organoids expressed higher levels of the AT2 marker SFTPC and the secretory gene SCGB3A2 (Fig. EV1E), recently described as a marker for multiple alveolar progenitor populations (Kadur Lakshminarasimha Murthy et al, 2022; Basil et al, 2022). Both HT2+ and HT2- derived cells had high expression levels of SFTPC when cultured with HRG1 compared to the housekeeping gene ACTB and AO-derived cells (Fig. EV1E). The expression level of SCGB3A2 was comparable between the two populations in the presence of HRG1. Given these results and the observation that EGF can lead to reduced long-term outgrowth in other organoid systems (Bannier-Hélaouët et al, 2024), we removed EGF from ALVO expansion media (ALVO-EM; Table EV1). Moreover, due to the high expression of SFTPC and SCGB3A2 in the HT2- population, we decided to not exclude this population in our sorting strategy and only use EPCAM+ and NGFR- sorting for enrichment of alveolar progenitor populations. With some variation between organoid lines, the EPCAM+ fraction contained around 90% NGFR- cells (Fig. EV1F). These ≥p1 EPCAM + / NGFR- cells gave rise to organoids that did not stain for airway markers (KRT5, SCGB1A1), while staining for SFTPC (Fig. EV1A, bottom panel). Notably, the cultures also contained cells that appeared to be negative for all three markers.

Next, we tested ALVO-EM side-by-side with the commercially available alveolar organoid expansion (AvOE) media from STEMCELL Technologies (SCT). We found that after 14 days, the cells had a ~ 30-fold reduced gene expression of SFTPC and an ~11-fold higher expression of CAV1 in AvOE, indicating increased AT1 differentiation compared to our ALVO-EM (Fig. EV1G). We

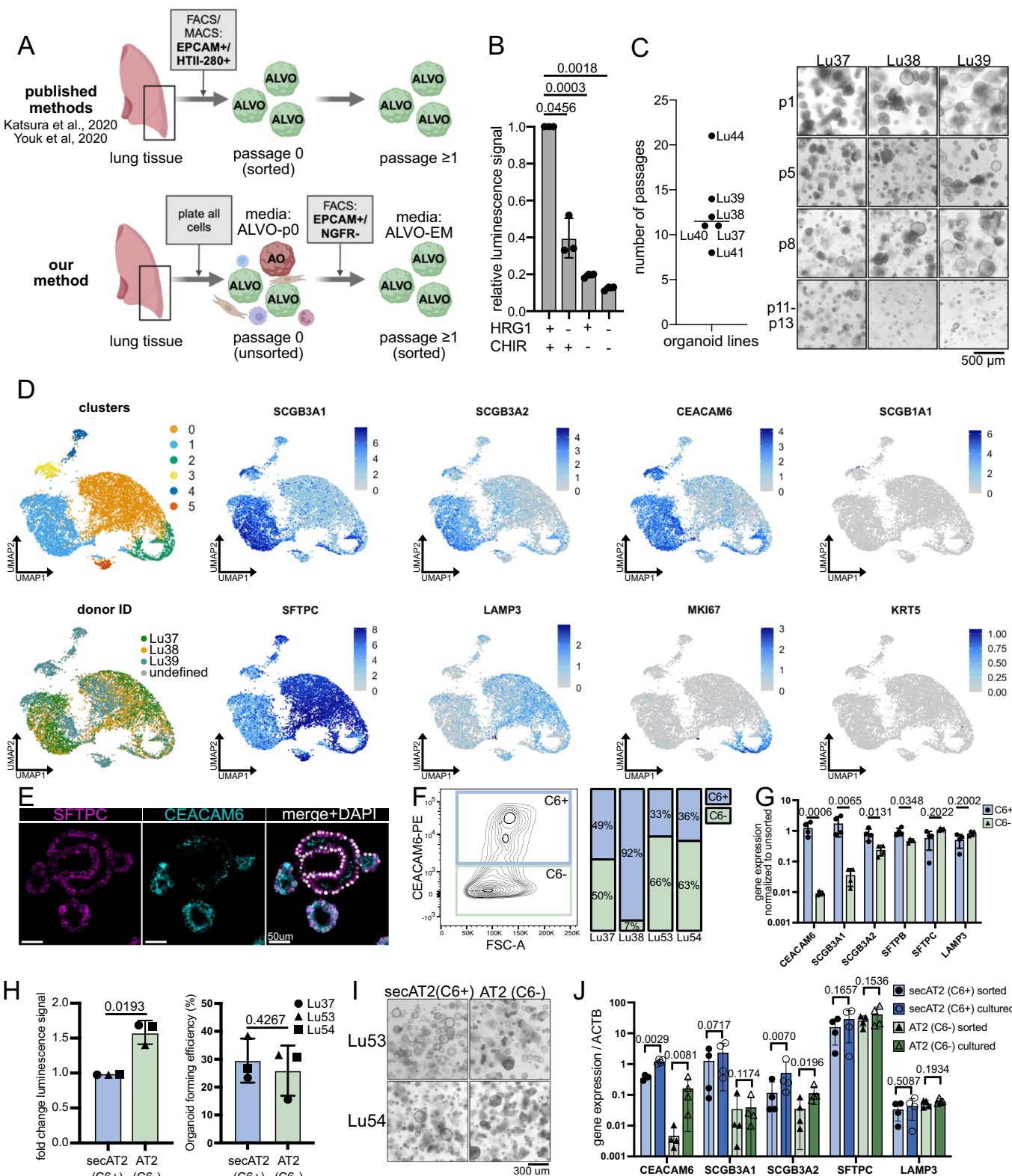

concluded that our media was better suited to prevent differentiation and promote expansion of alveolar progenitor cells within ALVOs.

Because the MAPK and Wnt pathways have been shown to be important drivers of proliferation in alveolar cells, we tested if this

was also the case in our ALVOs. Withdrawal of either HRG1, CHIR99021 (CHIR; GSK3-beta inhibitor/Wnt activator), or both from the ALVO-EM led to a significant reduction in cell numbers, indicating that indeed both pathways are essential for ALVO growth (Fig. 1B).

**Figure 1. Optimized ALVO conditions allow for long-term expansion of alveolar progenitor populations.**

(A) Schematic comparing published methods (Katsura et al, 2020 and Youk et al, 2020) with our method of generating ALVOs from lung tissue. (B) Viability assay (cell titer glo) of ALVOs cultured with the indicated factors for 14 days. Matched one-way ANOVA with Geisser–Greenhouse correction, followed by Dunnett's multiple comparisons versus control, was performed on log-transformed raw values. Data were shown as normalized mean ± SD. Exact adjusted p values are indicated. $N = 3$. (C) Left: Overview of the number of passages reached for the indicated ALVO lines. Median is indicated by a horizontal line. Right: Brightfield images of indicated ALVO lines at indicated passages. (D) UMAPs of scRNA-Seq data of p5 ALVOs in ALVO-EM showing cluster numbers, donor ID, and expression levels of indicated genes. (E) IF images of ALVOs in EM. (F) Flow cytometry plot and quantification of cells gated for single, DAPI- cells. Percentages of CEACAM6+ (C6+) and CEACAM6- (C6-) populations are indicated. (G) qPCR analysis of sorted C6+ and C6- cells, normalized to unsorted cells. A two-sided paired t-test was performed on log-transformed raw values. Data were shown as mean ± SD. Exact p values are indicated. $N = 4$. (H) Viability assay (cell titer glo) (left) and organoid-forming efficiency (right) of p4 sorted secAT2 (C6+) and AT2 (C6−) cells cultured for 12 days. A two-sided paired t-test was performed on log-transformed raw values. Data were shown as mean ± SD. Exact p values are indicated. $N = 3$. (I) Representative brightfield pictures of C6+ and C6- sorted cells. (J) qPCR analysis of sorted secAT2 (C6+) and AT2 (C6−) cells on day 0 and after culturing in ALVO-EM for 14 days. A two-sided paired t-test was performed on log-transformed raw values. Data were shown as mean ± SD. Exact p values are indicated. $N = 4$. Source data are available online for this figure.

Using the above-described strategy (Fig. 1A, bottom panel), we derived ALVO lines from multiple donors and passaged them in ALVO-EM conditions every 10–15 days. We achieved a median of 11 passages, with our best growing line still expanding at passage 21 (Fig. 1C). We monitored gene expression of alveolar markers over multiple passages and found that expression levels of AT2 markers were stably high, mostly above expression levels of the house-keeping gene beta-actin (ACTB), while levels of AT1 marker AGER remained low in the first ten passages (Fig. EV1H). Therefore, we conducted all further experiments within the first ten passages.

## ALVOs contain multiple alveolar progenitor populations

Because we observed heterogeneous SFTPC expression in the ALVOs (Fig. EV1A, bottom panel), we performed scRNA-Seq to further characterize the cell populations present in our cultures. We identified six clusters. Clusters 3, 4, and 5 were smaller and predominantly composed of cells from a single donor (Lu39), whereas clusters 0, 1, and 2 were larger and contained contributions from all three donors (Lu37, Lu38, and Lu39). Differential gene expression analysis of clusters 0, 1, and 2 revealed increased expression of SCGB3A1 and SCGB3A2 in cluster 1, while the expression of canonical AT2 markers (SFTPC and LAMP3) were reduced in these clusters (Fig. EV1I,D). Importantly, we did not find evidence of basal cell (KRT5) or club cell (SCGB1A1) markers. Cluster 0 had high expression of AT2 markers (SFTPC and LAMP3) and low expression of SCGB3A1 and SCGB3A2. Cluster 2 contained MKI67+ proliferating cells (Fig. EV1I,D). Intrigued by the presence of the SCGB3A1/3A2 high cluster, we looked for surface markers that could be used to sort out this population. We found that the expression of CEACAM6, previously described to be expressed by RAS cells (Basil et al, 2022), overlapped with the expression of SCGB3A1/3A2 (Fig. 1D). We confirmed with stainings that CEACAM6 protein is present in the ALVOs and that it is co-expressed with SFTPC in some cells (Fig. 1E). Next, we used FACS to sort out CEACAM6+ (C6 + ) and CEACAM6- (C6-) cells (Fig. 1F). The population sizes varied between organoid lines, reflecting heterogeneity common when working with human-derived cells. Gene expression analysis confirmed that the C6+ population was enriched for CEACAM6, SCGB3A1, SCGB3A2, and SFTPB while still maintaining high levels of AT2 markers SFTPC and LAMP3 (Fig. 1G); henceforth, we will refer to the C6+ population as secretory AT2 (secAT2) cells and the C6- population as AT2 cells. We plated the sorted secAT2 and AT2 cells in

organoid cultures to determine differences in cell growth between the populations. Both populations had high and comparable organoid-forming efficiencies (~15–38%), but AT2 cells yielded higher cell numbers, indicating increased proliferation compared to secAT2 cells (Fig. 1H). The secAT2-derived organoids were enriched for cystic organoids, while AT2-derived organoids showed more mixed morphologies (Fig. 1I). To check whether the two populations kept their identities over time, we investigated gene expression changes after 14 days in culture compared to freshly sorted cells. While expression of the AT2 markers (SFTPC and LAMP3) and SCGB3A1 remained stable in culture, the secAT2 markers CEACAM6 and SCGB3A2 were increased after 14 days in culture compared to day 0 (Fig. 1J). This could indicate selective pressure on AT2 cells to transition to a secAT2-like state.

## Maturation media supports AT2 surfactant secretion with tubular myelin formation

The ALVO-EM was optimized to sustain long-term proliferation and self-renewal of alveolar progenitor cells. These cells are known to express lower levels of surfactant proteins than their mature AT2 counterparts (Travaglini et al, 2020; Kadur Lakshminarasimha Murthy et al, 2022; Basil et al, 2022). To develop AT2-maturation media (AT2-MM), we screened for factors that increased the retention of lysotracker dye in the cells after initial organoid outgrowth (Fig. 2A). Lysotracker retention correlates with AT2 maturity, as it accumulates in the surfactant-producing and -storing organelles of the AT2 cell, the lamellar bodies (Van der Velden et al, 2013). We found that the addition of dexamethasone, cyclic adenosine monophosphate, and 3-isobutyl-1-methyl-xanthine, a cocktail termed "DCI", used in induced pluripotent stem cell (iPSC)-derived organoids to induce AT2 maturation (Jacob et al, 2017; Gonzales et al, 2002), led to the highest increase in lysotracker signal, followed by the cytokine interleukin (IL)-6 (Fig. 2B). The combined addition of DCI and IL-6 led to a higher increase in lysotracker signal compared to the single treatments (Fig. EV2A,B). To further confirm the increased maturation state of the AT2 cells, we performed electron microscopy and observed not only microvilli and lamellar bodies, but also pulmonary surfactant secreted into the lumen of the organoids (Fig. 2C). Remarkably, the secreted surfactant formed highly ordered structures, including mesh-like tubular myelin, which has important innate immunity functions (Fig. 2C) (Pison et al, 1994). However, analysis of genes involved in surfactant production (LAMP3 and NAPSA) and

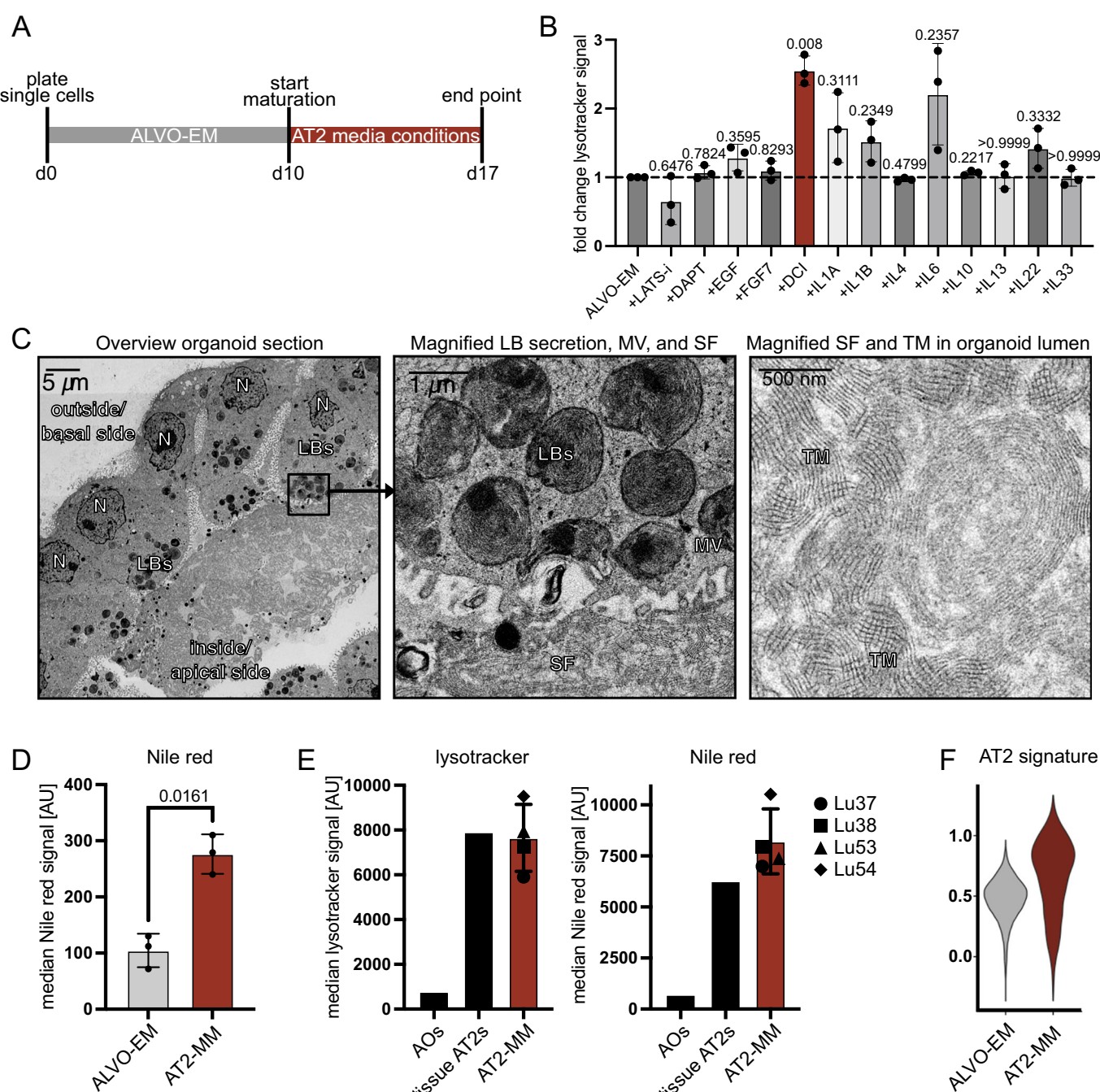

**Figure 2. Maturation media supports AT2 surfactant secretion with tubular myelin formation.**

(A) Timeline of media conditions used for ALVO cultures with indicated days (d). (B) Flow cytometry analysis of median lysotracker signal of organoids treated with indicated factors as outlined by the timeline in Fig. 2A. Matched one-way ANOVA with Geisser–Greenhouse correction, followed by Dunnett's multiple comparisons versus control, was performed on log-transformed raw values. Data were shown as normalized mean ± SD. Exact adjusted *p* values are indicated. $N = 3$. (C) Electron microscopy image of ALVOs exposed to DCI and IL-6 as outlined in Fig. 2A. Left: Part of an organoid containing mature AT2 cells and secreted surfactant inside the lumen. Center: Magnification of the apical side of one AT2 cell, showing MV and the process of the secretion of an LB into the lumen. Right: Magnification of the secreted SF and TM in the lumen of the organoid. N nucleus, LBs lamellar bodies, SF surfactant, MV microvilli, TM tubular myelin. (D) Flow cytometry analysis showing median Nile red signal of d17 organoids in ALVO-EM or AT2-MM as outlined in Fig. 2A. A two-sided paired *t*-test was performed on log-transformed raw values. Data were shown as mean ± SD. Exact *p* values are indicated. $N = 3$. (E) Flow cytometry analysis showing median Nile red signal of d17 organoids in AT2-MM compared to AOs (negative control) and primary tissue HT2-280 + AT2 cells. Data were shown as median ± SD. $N = 4$. (F) Violin plots showing scRNA-Seq expression data of a published AT2 signature (Burgess et al, 2024) in ALVOs in indicated media conditions. $N = 3$ biological replicates. Source data are available online for this figure.

 

surfactant proteins (*SFTPA1*, *SFTPD*, and *SFTPC*) showed that addition of DCI alone led to increased or unchanged expression of all markers, whereas IL-6 treatment—either alone or in combination with DCI—resulted in decreased *SFTPC* expression (Fig. EV2C). Because *SFTPC* is important for the integrity of the surfactant lipid layer, we decided not to include IL-6 in the AT2-MM (Table EV1) going forward. To confirm the increased AT2 maturity of the ALVOs in AT2-MM, we stained the surfactant-containing lamellar bodies of the organoids with the lipid dye Nile red, as pulmonary surfactant comprises of 90% lipids. As expected, we observed an increase in Nile red in AT2-MM compared to ALVO-EM (Fig. 2D). To confirm that Nile red staining intensity correlates with AT2 marker expression, we performed gene expression analysis on sorted Nile red-low and –high cells from AT2-MM–cultured organoids (Fig. EV2D,E). Nile red–high cells showed higher expression of AT2 markers than Nile red-low cells, and both populations expressed higher AT2 marker levels than AO-derived cells. In addition, comparison of Lysotracker and Nile red fluorescence intensities revealed comparable levels in AT2-MM organoid-derived cells and primary tissue-derived AT2 cells, whereas AO-derived cells exhibited substantially lower intensities (Fig. 2E). Next, we performed scRNA-Seq on the AT2-MM condition and found that the expression levels of a published AT2 signature (Burgess et al, 2024) was higher in AT2-MM than in ALVO-EM (Fig. 2F; Dataset EV1). Lastly, we found that both C6- and C6+ –derived cells achieve high Nile red signal when matured in AT2-MM compared to AO-derived cells (Fig. EV2F).

## Removal of proliferation factors and inhibition of LATS drives AT1 differentiation

To study alveolar regeneration in a controlled in vitro environment, defining media conditions to induce AT1 differentiation is crucial. Existing protocols use human serum (HS) to induce differentiation (Katsura et al, 2020). In iPSC- and fetal tissue-derived organoids, it was shown that a LATS-inhibitor (LATS-i) in combination with the withdrawal of Wnt-activators drives the AT1-phenotype (Burgess et al, 2024; Lim et al, 2025). To develop an AT1 differentiation medium, we removed all factors from the ALVO-EM that support proliferation and stemness to create a minimal base media (BM) (Table EV1). After expansion of the ALVOs for 10 days, we switched to the commercial SCT differentiation media (AvOD), BM, BM + LATS-i, or BM + HS for 7 days (Fig. 3A). We found that the BM was sufficient to induce the upregulation of AT1 markers (*AGER* and *CAV1*) and downregulation of AT2 markers (*SFTPA1* and *SFTPC*) (Fig. EV3A). This effect was strongest in the BM + LATS-i condition, indicating that the removal of proliferation and stemness factors in combination with LATS inhibition induced the AT1 program more strongly than HS and the commercial SCT media. Organoids in BM + LATS-i were smaller and more compact compared to ALVO-EM (Fig. 3B). When we tested YAP target gene expression (Stein et al, 2015), we found that *ANKRD1* and *CCN1* were upregulated in BM + LATS-i compared to ALVO-EM, suggesting that the differentiation was indeed correlated with active YAP signaling (Fig. EV3B). Notably, BM alone was sufficient to increase YAP target genes, indicating that the default AT1 differentiation trajectory of AT2 cells is concomitant with active YAP signaling (Fig. EV3B). To investigate the differentiation dynamics, we performed a qPCR time course for

14 days after switching to BM + LATS-i (Fig. EV3C). Expression levels of the AT2 marker *SFTPC* already decreased after 2 days in BM + LATS-i and kept decreasing over time. The AT1 markers *AGER* and *CAV1* increased after 2 days of differentiation and kept increasing until they reached a plateau around 7 days. When we stained the organoids for SFTPC and AGER protein, we found that SFTPC was entirely absent, while all organoids expressed high levels of AGER and GPRC5A (Fig. EV3D). Electron microscopy confirmed that the cells did not contain lamellar bodies (Fig. EV3E). Instead, we observed cell shapes that appeared to have a low cytoplasm-to-nucleus ratio, reminiscent of AT1 cells. However, we also observed large vacuoles in the cells, a sign of apoptosis and an unhealthy culture (Fig. EV3E). Furthermore, these organoids were very difficult to dissociate enzymatically and mechanically without harming the cells, making analysis that requires single-cell suspensions challenging.

## Inhibition of LATS in expansion conditions results in mixed AT1/AT2 organoids

We concluded that conditions that allow for both AT1 and AT2 cells to be present at the same time might lead to more physiologically relevant cultures. Therefore, we tested whether adding LATS-i to our ALVO-EM (= AT1/2-M; Table EV1) is sufficient to drive AT1 differentiation in some cells. After an initial expansion phase of 10 days, we added LATS-i for 7 days and observed a striking change in morphology, with darker, more compact organoids (Fig. 3B) and increased attachment to the plate and growth in 2D (Fig. 3C). In contrast to BM + LATS-i, these organoids were not smaller than organoids in ALVO-EM (Fig. 1B). ScRNA-Seq confirmed a decrease in the published AT2 signature and an increase in an AT1 signature (Burgess et al, 2024) in AT1/2-M compared to ALVO-EM (Fig. 3D; Dataset EV1). Despite the strong upregulation of AT1 markers (*AGER* and *CAV1*), some cells still expressed proliferation markers (*MKI67*), AT2 markers (*SFTPC* and *LAMP3*), and secAT2 markers (*SCGB3A1* and *SCGB3A2*) (Fig. 3E).

We confirmed with stainings that the organoids contained SFTPC-expressing cells, and cells with strong expression of the AT1 markers *AGER* and *GPRC5A*, indicating a mix of both cell types in the AT1/2-M culture conditions (Fig. EV3D). Because we detected proliferating cells and did not observe a decrease in the size of the organoids grown in AT1/2-M compared to ALVO-EM, we tested whether the organoids would grow in AT1/2-M added from day 0 of culture, when they were still single cells (Fig. 3F). Surprisingly, the organoids grew out well and stained for SFTPC and AGER after 14 days (Fig. 3F), indicating that inhibition of LATS triggers AT1 differentiation without inhibiting proliferation. In fact, a cell growth time course revealed that organoids in AT1/2-M grow faster than organoids in ALVO-EM (Fig. EV3F).

Two-dimensional (2D) air–liquid-interface (ALI) conditions can induce differentiation in other systems, including AOs (Sachs et al, 2019). Therefore, we tested whether ALI is sufficient to trigger AT1 differentiation in ALVOs. After 7 days of ALI culture with ALVO-EM at the bottom, we observed structured 3D growth. However, we did not detect AGER protein, indicating that 2D ALI cultures alone were not sufficient to induce AT1 marker expression (Fig. EV3F). We tested if AT1/2-M would lead to AGER induction and found that, similar to our 3D cultures, we detected SFTPC+ and AGER+

  

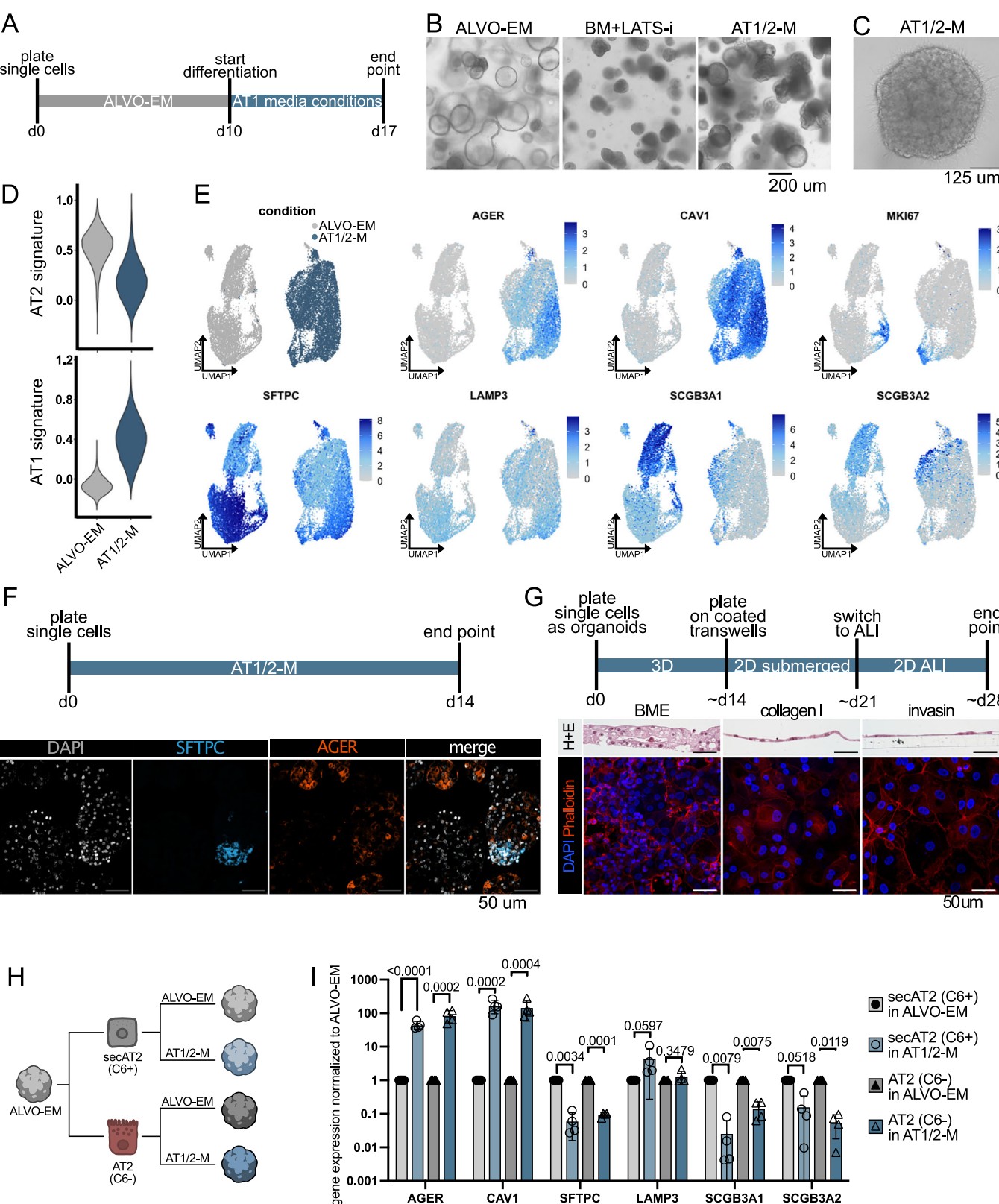

◄ **Figure 3. Inhibition of LATS in expansion conditions results in mixed AT1/AT2 organoids.**

(A) Timeline of media conditions used for ALVO cultures with indicated days (d). (B) Brightfield images of p7 ALVO organoids cultured in the indicated conditions as outlined in Fig. 3A. (C) Close-up brightfield image of organoid cultured in AT1/2-M as outlined in Fig. 3A with visible attachment to the plate. (D) Violin plots showing scRNA-Seq expression data of published AT2 and AT1 signatures (Burgess et al, 2024) in ALVOs in indicated media conditions. (E) UMAPs of scRNA-Seq data from ALVOs cultured in ALVO-EM or AT1/2-M, indicating media condition and expression levels of indicated genes. (F) IF images of ALVOs cultured in AT1/2-M for 14 days, as indicated by the timeline. DAPI = nuclei; SFTPC = AT2 marker; AGER = AT1 marker. (G) Cells were first cultured in 3D, then in 2D submerged, then in ALI transwells cultures in AT1/2-M as outlined in timeline. Side-view brightfield (H + E staining) and top-down IF images of ALI transwell cultures at the end point. DAPI = nuclei; Phalloidin = F-actin. (H) Schematic showing sorting and culturing strategy for qPCR data in Fig. 3I. (I) Gene expression analysis of sorted secAT2 (C6+) and AT2 (C6−) cells that were cultured in ALVO-EM or AT1/2-M for 14 days. Two-sided paired t-tests were performed on log-transformed raw values. Data are shown as normalized mean ± SD. Exact p values are indicated. N = 4. Source data are available online for this figure.

cells in ALI cultures (Fig. EV3G). Despite strong AGER expression, we did not observe the typical AT1 shape as found in vivo. Therefore, we grew the organoids in AT1/2-M conditions before we plated the cells in 2D ALI cultures (Fig. 3G). Because cell adhesion plays an important role in maintaining the AT1 cell shape, we tested different coatings of the transwells, including BME, invasin (Wijnakker et al, 2025), and collagen I. We found that the cells grew into 3D structures when plated on BME, and we did not observe cells with a morphology reminiscent of AT1 cells. In contrast, cells plated on invasin or collagen I formed a thin 2D monolayer and reached diameters of up to 100 um, similar to AT1 cells in vivo (Fig. 3G).

Given the cell heterogeneity in our ALVOs, we investigated whether both alveolar progenitor populations (secAT2 and AT2) present in ALVO-EM can differentiate to AT1 cells. We used the same sorting strategy as described previously (Fig. 1F) and plated the two populations either in ALVO-EM or AT1/2-M (Fig. 3H,I). We saw a steep increase in AT1 markers (AGER and CAV1) and a decrease in SFTPC, SCGB3A1, and SCGB3A2 in both populations when cultured in AT1/2-M. Expression of the AT2 marker LAMP3 remained stable. This data indicates that both secAT2 and AT2 cells can differentiate towards an AT1-phenotype.

## ScRNA-Seq comparison to lung tissue highlights regenerative phenotype of ALVOs

To further characterize the organoid-derived cells and compare them to their tissue counterparts, we took the scRNA-Seq data from the organoids grown in ALVO-EM, AT2-MM, and AT1/2-M (Fig. 4A) and integrated the dataset with the epithelial cell cluster from a previously published lung tissue dataset (Sikkema et al, 2023) (Figs. 4B and EV4A). Using the annotations of the tissue dataset, we observed that the organoid-derived cells overlapped with alveolar cell populations, including "AT0", "AT1", "AT2", and "AT2 proliferating". Additionally, these cells appeared in close proximity to "Neuroendocrine" (NE), "SMG mucous" (SMG: submucosal gland), and "pre-TB secretory". Unsupervised clustering identified 31 clusters in total. For subsequent analysis, we selected clusters that included organoid-derived cells (Fig. EV4A).

The resulting subset was re-clustered and annotated based on known marker genes and prior tissue annotations (Fig. 4C–E; Dataset EV2; see Methods for more details). All major clusters contained contributions from all three organoid donors (Fig. EV4B). Among the clusters, the "cycling" cluster was highly enriched for proliferation-associated genes (TOP2A, MKI67, and CDK1) and comprised predominantly organoid-derived cells from all three conditions, underscoring the pro-proliferative

environment of organoid cultures relative to homeostatic lung tissue (Fig. 4E). The "SMG mucous" cluster, containing mostly tissue-derived cells, expressed known SMG goblet cell markers (MUC5B, NKX3-1, and TFF3). The "NE + Ionocyte" cluster included tissue-derived ionocytes (FOXI1 and ASCL3) and NE cells (CALCA and ASCL1), and also featured contributions from organoids, particularly from the AT2-MM condition (Figs. 4E and EV4C). Interestingly, organoid-derived cells from the AT2-MM condition also formed the majority of an "NE-like" cluster (Figs. 4E and EV4C). Because the "NE-like" cluster was predominantly composed of cells from donor Lu39, we assessed by qPCR whether the other two donors (Lu37 and Lu38) also showed increased expression of the neuroendocrine transcription factor ASCL1 in AT2-MM. Although Lu39 exhibited the highest overall expression, all three donors showed increased ASCL1 expression in AT2-MM (Fig. EV4D). Further differential expression analysis of the NE-containing clusters revealed that while ionocyte markers were absent in organoids, NE markers and AT2 markers (SFTPA2 and SFTPC) were expressed at higher levels (Fig. EV4E). Instead of alveolar markers, tissue-derived cells in this cluster expressed SCGB1A1, a club cell marker reported in a subset of NE cells (Reynolds et al, 2000).

The "AT2", "AREGhi AT2", and "secretory AT2" clusters exhibited high expression of canonical AT2 markers (SFTPC, SFTPA1, SFTPD, NAPSA, ABCA3, LAMP3, and HHIP) (Fig. 4E). The "AT2" cluster, the largest in this subset, contained both tissue- and organoid-derived cells, with most organoid cells derived from ALVO-EM and AT2-MM conditions. Differential expression analysis between tissue- and organoid-derived AT2 cells revealed similar expression levels of canonical AT2 markers (Fig. EV4F; Dataset EV2). Notably, genes that differed between these two groups included histone-modifying proteins and ATPases. GO term analysis for biological processes (BP) revealed terms surrounding RNA and protein regulation, possibly reflecting differences between the in vitro versus in vivo environments (Fig. EV4G). Consistent with this interpretation, performing differential gene expression analysis on the entire tissue and organoid cell subset yielded similar findings (Fig. EV4H,I; Dataset EV2). The "AREGhi AT2" cluster consisted mostly of tissue-derived cells and likely represents an aberrant AT2 state, as AREG expression has been associated with fibrosis (Zhao et al, 2024) (Fig. 4E). "Secretory AT2" cells, compared to the "AT2" cluster, showed lower expression of canonical AT2 markers (Fig. 4F). Moreover, this cluster expressed higher levels of MUC5B, SCGB3A1, SCGB3A2, and CEACAM6 (Fig. 4F), likely corresponding to the secAT2s population described in Fig. 1. The "AT1" cluster, expressing canonical AT1 markers (CAV1, AGER, TIMP3, CLIC5, and PDPN),

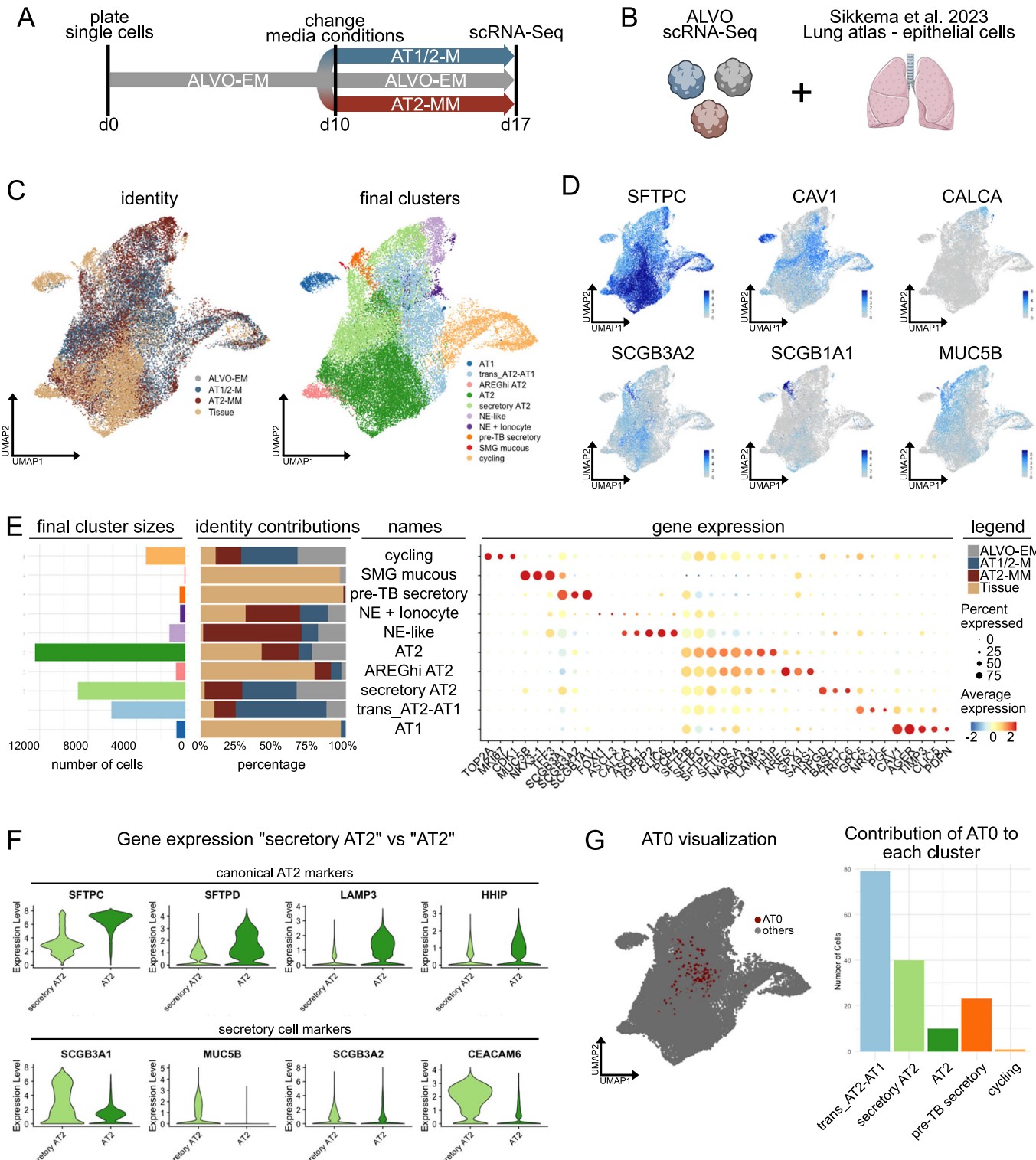

**F** Gene expression "secretory AT2" vs "AT2"

**G** AT0 visualization

Contribution of AT0 to each cluster

contained primarily tissue-derived cells and a small fraction of AT1/2-M organoid cells (Fig. 4E). Most AT1/2-M organoid cells fell into a "trans_AT2-AT1" cluster, representing a transitional state. This cluster co-expressed both AT1 and AT2 markers, albeit at lower levels than the canonical "AT1" and "AT2" clusters.

Because AT0 cells have also been described as an intermediate state between AT2s, AT1s, and secretory cells, we investigated their location and found that most tissue-derived AT0 cells were present in the "trans_AT2-AT1" cluster, followed by the "secretory AT2" cluster (Fig. 4G).

**Figure 4. ScRNA-Seq comparison to lung tissue highlights regenerative phenotype of ALVOs.**

(A) Timeline of media conditions used for p5 ALVO cultures that were processed for scRNA-Seq. (B) Schematic showing the integration of the scRNA-Seq datasets from the ALVOs and epithelial cells of the lung atlas from Sikkema et al, 2023. (C) UMAP showing integrated organoid/tissue subset, colored by organoid media condition and tissue origin (identity, left plot) and Seurat clusters (final clusters, right plot). (D) UMAP showing expression levels of the indicated genes. (E) Integrated organoid/tissue subset. Left: Bar plot showing absolute cell contributions to each final cluster. Center: Bar plot showing relative cell contributions of the identities to each cluster. Right: Dotplot showing gene expression of selected genes. (F) Violin plots showing gene expression levels of selected genes in "secretory AT2" and "AT2" clusters. $N = 3$ biological replicates. (G) UMAPs visualizing the location of AT0 cells (from tissue annotations) (left), and a bar plot showing the contribution of AT0 cells to final clusters (right).

In summary, while our organoids contain AT2 cells closely resembling their tissue counterparts, they also contain NE-like, secAT2 cells, and AT2-AT1-transitioning cells. The two latter populations are transcriptionally similar to the described multipotent progenitors, reflecting the heightened plasticity in organoid cultures compared to homeostatic lung tissue.

## IFN-γ selectively promotes alveolar progenitor growth while impairing AT1-like cells

To study alveolar epithelial regeneration in a pro-inflammatory environment as found in COPD and other lung diseases, we cultured organoids in ALVO-EM (containing secAT2 and AT2 progenitor populations) or AT1/2-M (containing AT2-like and AT1-transitioning cells) conditions in the presence of cytokines that are upregulated in lungs of COPD patients: IFN-γ, tumor necrosis factor-alpha (TNF-α), IL-6, IL-1α, and IL-1β (Bhowmik et al, 2000; Keatings et al, 1996; Sapey et al, 2009; Grumelli et al, 2004). We omitted the AT2-MM condition in these studies, as the presence of the inflammatory response suppressor dexamethasone could have confounded our results. All tested cytokines led to similarly sized or larger organoids in ALVO-EM conditions, while the organoids in AT1/2-M appeared significantly smaller in the presence of IFN-γ but not the other cytokines (Fig. EV5A).

Intrigued by the distinct effect of IFN-γ on the two organoid types, we quantified these changes using a cell viability assay. When IFN-γ was added at the start of organoid culture and maintained for 14 days, cell numbers increased under ALVO-EM conditions but decreased under AT1/2-M conditions (Fig. 5A). Corroborating these findings, IFN-γ added to the organoids on day 11 and incubating for 3 days did not result in cytotoxicity in ALVO-EM conditions, while the cytotoxic effect doubled in AT1/2/M compared to the control (Fig. 5B). When we examined expression levels of *SFTPC*, *SFTPA1* (AT2 markers), and *AGER* (AT1 marker) in AT1/2-M organoids, we found that exposure to IFN-γ led to an increase in *SFTPA1* and a decrease in *AGER*, indicating a selective loss of AT1-like cells but not alveolar progenitor cells in this condition (Fig. 5C). To determine if we could rescue the observed cytotoxic effect of IFN-γ in AT1/2-M organoids, we tested the Janus Kinase 1/2 (JAK1/2) inhibitor Ruxolitinib (RX). JAK1/2 acts downstream of the IFN-γ receptor to phosphorylate and activate STAT1, the key transcription factor in IFN-γ signaling. First, we assessed RX tolerance and found that concentrations up to 1 μM were well tolerated, whereas 10 μM was toxic (Fig. EV5B). At 1 μM, RX fully rescued AT1 cells from IFN-γ-induced cytotoxicity and restored expression of *AGER* and *CAV1* to levels comparable to the control (Figs. 5D,E and EV5C). To confirm that this represented on-target effects, we analyzed gene expression levels of the STAT1

target genes *IRF1* and *SOCS1* and found that RX suppressed the expression of these genes in a dose-dependent manner (Fig. EV5D).

Because the positive effect of IFN-γ in ALVO-EM showed variability among biological replicates (Fig. 5A), we performed a time course in ALVO-EM conditions to better understand the growth dynamics in the presence of IFN-γ and RX. Across three biological replicates, organoids exhibited accelerated growth during the first 10 days of IFN-γ exposure, followed by a reduction in cell numbers (Fig. EV5E). RX treatment prevented both the initial IFN-γ-driven growth acceleration and the subsequent cell number decline (Fig. EV5E). The reduction in cell numbers after day 10 was likely due to increased cell death observed in the organoid cultures exposed to IFN-γ, indicated by increased cell shedding and cell debris around the organoids (Fig. EV5F).

Our findings indicate that short-term IFN-γ treatment promotes the growth of alveolar progenitor cells, whereas prolonged exposure leads to cell death. To determine whether the distinct effects of IFN-γ on AT1-like and alveolar progenitor cells is dose-dependent, we treated AT1/2-M and ALVO-EM organoids with IFN-γ concentrations ranging from 0.1 to 100 ng/ml. In AT1/2-M cultures, we observed reduced cell numbers at both early (day 10) and late (day 16) time points starting at concentrations as low as 1 ng/ml, suggesting that even low levels of IFN-γ are toxic to AT1-like cells (Fig. 5F). In contrast, under ALVO-EM conditions, low IFN-γ concentrations promoted cell growth, while toxicity emerged only at higher doses (Fig. 5F). Although this pattern was consistent across all three biological replicates, the degree of variation indicates that different donors may exhibit varying sensitivities to IFN-γ.

To test whether IFN-γ promotes growth in both secAT2 and AT2 cells, we made use of the same sorting strategy as previously (Fig. 1F) and plated the populations separately in the absence or presence of IFN-γ. In both populations, the addition of IFN-γ enhanced the growth of the organoids without changing the organoid-forming efficiency (Fig. 5G). In summary, IFN-γ negatively affects AT1-like cells across a range of concentrations and exposure times, whereas lower concentrations and shorter treatments enhance the growth of secAT2 and AT2 progenitor cells.

To better understand the distinct effect of IFN-γ on alveolar progenitor cells and AT1-like cells at a molecular level, we treated 14-day-old organoids grown in ALVO-EM or AT1/2-M with IFN-γ and collected RNA at 0.5, 1, 4, and 8 h post-treatment. IFN-γ target genes *IRF1*, *SOCS1*, and *STAT1* were upregulated in both media conditions at 4 and 8 h of IFN-γ exposure but not at the earlier time points (Fig. EV5G). To further investigate IFN-γ induced changes in gene expression, we performed RNA-Seq on samples collected at the 4 h time point. After filtering, we found 142 genes that were upregulated upon IFN-γ treatment in both ALVO-EM and AT1/2-

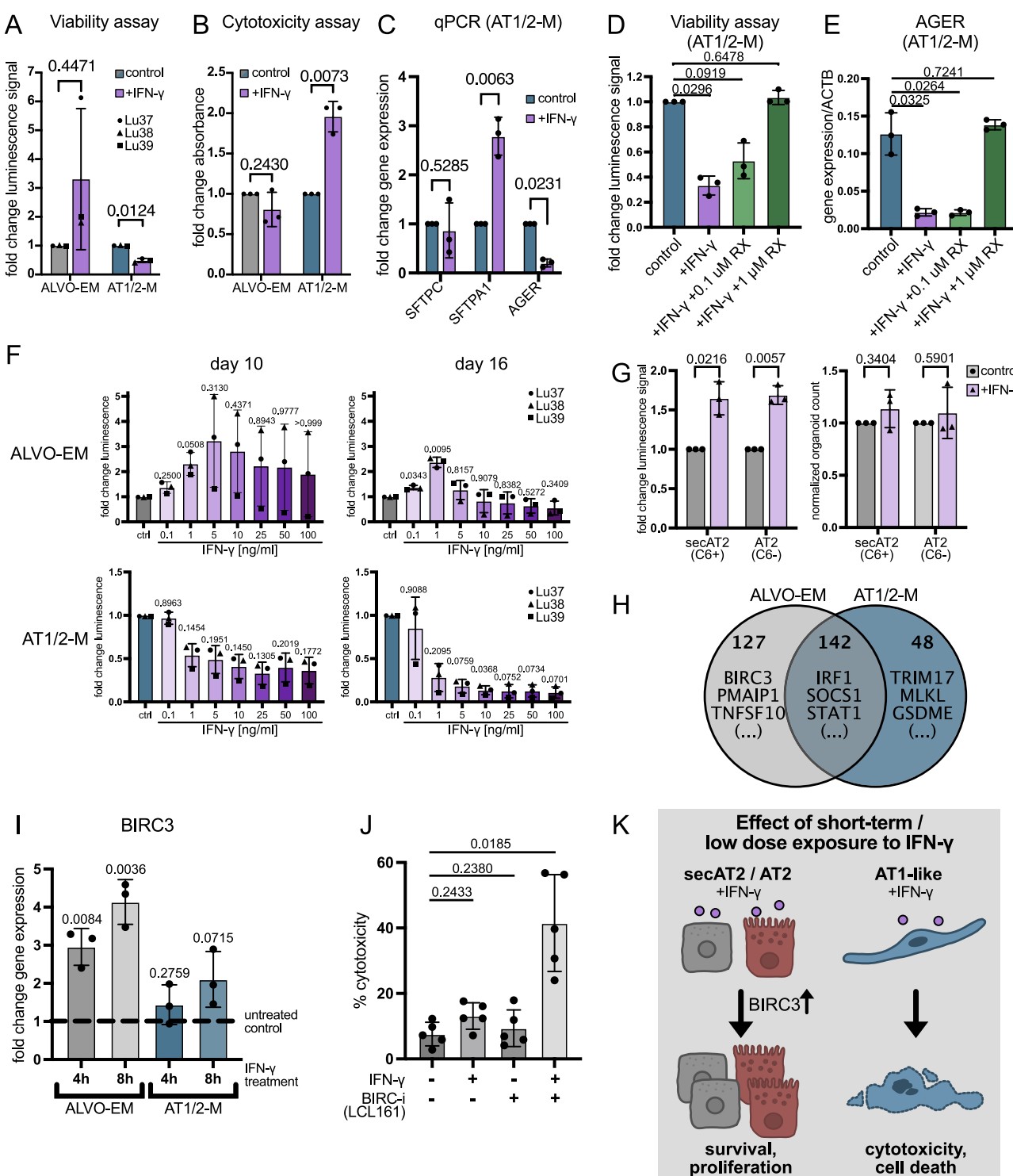

M, including known IFN-γ target genes such as *IRF1*, *SOCS1*, and *STAT1*. In addition, 48 genes were upregulated exclusively in AT1/2-M, including genes associated with cell death such as *TRIM17*, *MLKL*, *GSDME*, and *STING1* (Li et al, 2021; Basu-Shrivastava et al, 2021; Zhang et al, 2021; Jiang et al, 2020). 127 genes were uniquely upregulated in ALVO-EM, including genes that modulate cell death, such as *BIRC3*, *PMAIP1*, and *TNFSF10* (Liang et al, 2020;

Nikoletopoulou et al, 2013; Hao et al, 2023) (Fig. 5H; Dataset EV3). Although *PMAIP1* and *TNFSF10* are generally considered pro-apoptotic, cell death regulation depends on the balance between pro- and anti-apoptotic signals. Notably, only ALVO-EM organoids increased expression of the anti-apoptotic gene BIRC3 at 4 and 8 h post-IFN-γ exposure, hinting at a mechanism by which AT2 progenitor cells could evade the cytotoxic effects of IFN-γ

**Figure 5. IFN-γ selectively promotes alveolar progenitor growth while impairing AT1-like cells.**

(A) Viability assay of ALVOs cultured in indicated media conditions in the absence (control) and presence of IFN-γ (10 ng/ml) for 14 days. Two-sided paired *t*-tests were performed on log-transformed raw values. Data were shown as normalized mean ± SD. Exact *p* values are indicated. N = 3. (B) Cytotoxicity assay of ALVOs cultured in the indicated media conditions for 14 days. IFN-γ (10 ng/ml) was added on day 11 for 3 days. Two-sided paired *t*-tests were performed on log-transformed raw values. Data were shown as normalized mean ± SD. Exact *p* values are indicated. N = 3. (C) qPCR analysis of alveolar markers in ALVOs cultured in AT1/2-M in the absence (control) or presence of IFN-γ (10 ng/ml) for 14 days. Two-sided paired *t*-tests were performed on log-transformed raw values. Data were shown as normalized mean ± SD. Exact *p* values are indicated. N = 3. (D) Viability assay (cell titer glo) of ALVOs cultured in AT1/2-M in the absence (control) and presence of IFN-γ (10 ng/ml) and RX (0.1 and 1 uM) for 14 days. Matched one-way ANOVA with Geisser–Greenhouse, correction followed by Dunnett's multiple comparisons versus control was performed on log-transformed raw values. Data were shown as normalized mean ± SD. Exact adjusted *p* values are indicated. N = 3. (E) qPCR analysis of *AGER* in ALVOs cultured in AT1/2-M in the absence (control) and presence of IFN-γ (10 ng/ml) and RX (0.1 and 1 uM) for 14 days. Matched one-way ANOVA with Geisser–Greenhouse correction, followed by Dunnett's multiple comparisons versus control, was performed on log-transformed raw values. Data were shown as mean ± SD. Exact adjusted *p* values are indicated. N = 3. (F) Viability assay of ALVOs cultured in indicated media conditions for 10 or 16 days in the presence of different IFN-γ concentrations. Matched one-way ANOVA with Geisser–Greenhouse correction, followed by Dunnett's multiple comparisons versus control, was performed on log-transformed raw values. Data were shown as normalized mean ± SD. Exact adjusted *p* values are indicated. N = 3. (G) Viability assay and normalized organoid count of sorted secAT2s (C6+) and AT2s (C6−) cultured in ALVO-EM with and without IFN-γ (1 ng/ml) for 12 days. Two-sided paired *t*-tests were performed on log-transformed raw values. Data were shown as normalized mean ± SD. Exact *p* values are indicated. N = 3. (H) Venn diagram of genes upregulated in ALVO-EM and AT1/2-M upon treatment of IFN-γ (10 ng/ml) for 4 h determined by RNA-Seq. (I) qPCR analysis of *BIRC3* in ALVO-EM and AT1/2-M upon treatment of IFN-γ (10 ng/ml) for 4 and 8 h, normalized to their respective untreated controls (dotted line). Matched one-way ANOVA with Geisser–Greenhouse correction, followed by Dunnett's multiple comparisons versus control, was performed on log-transformed raw values. Data were shown as normalized mean ± SD. Exact adjusted p-values are indicated. N = 3. (J) Cytotoxicity assay of organoids in ALVO-EM with and without IFN-γ and the BIRC3 inhibitor LCL161 for 3 days. Matched one-way ANOVA with Geisser–Greenhouse correction, followed by Dunnett's multiple comparisons versus control, was performed on log-transformed raw values. Data were shown as mean ± SD. Exact adjusted *p* values are indicated. N = 5. (K) Schematic summarizing the effect of IFN-γ on the different alveolar epithelial cell types. Source data are available online for this figure.

(Fig. 5I). To test this, we treated ALVOs in ALVO-EM with IFN-γ in the absence or presence of the BIRC-inhibitor LCL161 for 3 days. While treatment with IFN-γ or LCL161 alone had no effect, the combination led to an increase in cytotoxicity (Fig. 5J), confirming that upregulation of BIRC3 protects the alveolar progenitor cells from the cytotoxic effect of IFN-γ (Fig. 5K). In AT1/2-M, both the addition of LCL161 alone and in combination with IFN-γ caused an increase in cytotoxicity (Fig. EV5H). This indicates that even though BIRC3 is not upregulated upon IFN-γ in AT1-like cells, it might still play a protective role in these cells in general. Lastly, we tested whether short-term (7 days) low-dose (1 ng/ml) IFN-γ exposure of organoids in ALVO-EM would impair the differentiation capacity of the cells. Both untreated and treated cells increased expression of AT1 markers *AGER* and *CAV1* to comparable levels, indicating that their differentiation capacity was not impaired by IFN-γ exposure (Fig. EV5I).

## ALVO-macrophage (MΦ) co-cultures recapitulate IFN-γ–mediated effects on alveolar epithelial cells

To determine whether the distinct effects of IFN-γ on alveolar progenitors and AT1-like cells hold up in a more complex setup, we developed an ALVO-MΦ co-culturing system that allowed us to test the effect of MΦ-produced cytokines on ALVOs. IFN-γ is produced during viral and bacterial infections by various immune cells, including MΦ, that are themselves stimulated by IFN-γ (Kak et al, 2018). MΦs are closely associated with alveolar epithelial cells and are found in greater numbers in COPD patients (Di Stefano et al, 1998; Finkelstein et al, 1995). To model the environment during an immune response to infection, we treated peripheral blood mononuclear cell (PBMC)-derived MΦs with lipopolysaccharides (LPS) and IFN-γ for 24 h before setting up a co-culture with ALVOs growing in BME domes in the upper compartment of a transwell (Fig. 6A). By transferring the transwells into plates containing fresh pretreated MΦ every 3–4 days, we maintained a consistent MΦs-derived cytokine supply for 14 days without having to disrupt the attached MΦs (Fig. 6B).

Under these conditions, we observed that MΦ-derived factors inhibited the growth of organoids cultured in AT1/2-M, while promoting the growth of organoids in ALVO-EM, mirroring the effect of the direct IFN-γ treatments (Fig. 6C,D). To confirm that these effects were IFN-γ-dependent, we repeated the co-culture in the presence of the IFN-γ signaling inhibitor RX. The addition of RX rescued the observed phenotype (Fig. 6E). In summary, these co-culture experiments show that MΦ exposed to a pro-inflammatory infection-like environment produce cytokines, including IFN-γ, in sufficient concentrations to affect alveolar progenitor and AT1-like cells, mediating a growth advantage and growth inhibition, respectively (Fig. 6F).

## Discussion

There is growing evidence that the notion of AT2 cells as the stem cells of the alveoli that give rise to AT1 cells over-simplifies the alveolar regeneration process. Lung epithelial cells demonstrate remarkable plasticity, with multiple populations, including airway cells, capable of repopulating the alveolar region following injury (Kadur Lakshminarasimha Murthy et al, 2022; Basil et al, 2022; Salwig et al, 2019; Liu et al, 2024a, 2020). To better capture this complexity, we developed a defined approach to modeling alveolar regeneration with human organoids. The sorting strategy and media conditions facilitate the expansion of cells co-expressing transitioning and secretory markers while maintaining baseline AT2 marker expression. This approach acknowledges the notion that regenerative alveolar progenitor cells are heterogeneous and not equivalent to homeostatic AT2 cells. Although we identified transcriptional differences between the AT2 and secAT2 populations and showed that they can be enriched using a C6-based sorting strategy, we did not observe clear functional differences between the two populations yet. Future studies should therefore assess their long-term self-renewal capacity, engraftment potential, and ability to transdifferentiate toward airway epithelial fates. Notably, the co-expression of multiple lineage markers may signify

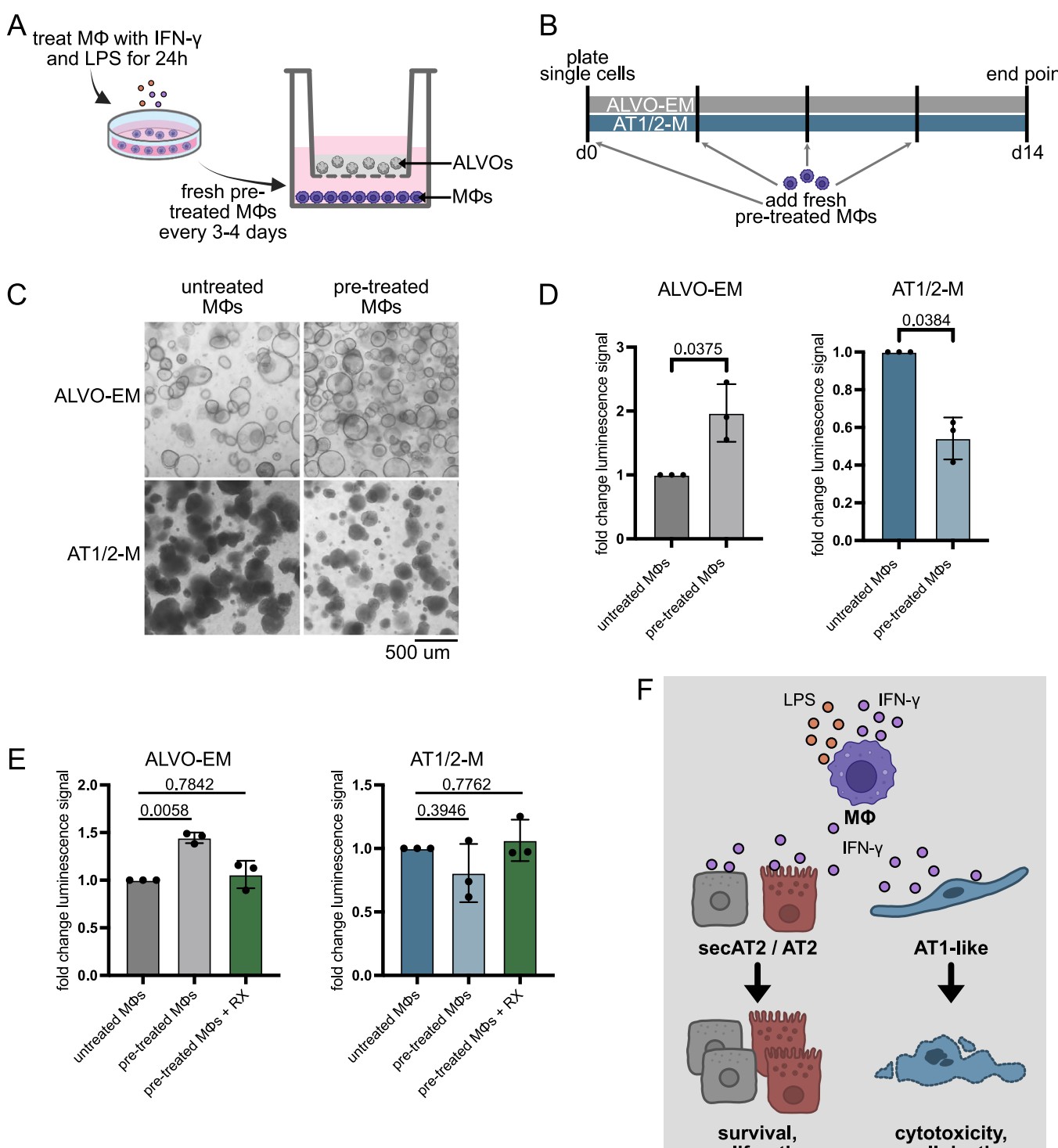

enhanced cellular plasticity and a greater potential to differentiate into diverse lung epithelial cell types.

Under the regeneration-mimicking organoid conditions, the addition of DCI induces AT2 maturation, as evidenced by increased pulmonary surfactant production and the presence of highly ordered tubular myelin in organoid lumens. To our knowledge, this is the first time tubular myelin has been observed in human alveolar in vitro cultures. While dexamethasone is known to accelerate AT2 maturation in fetal lungs, it is mostly used as a systemic anti-inflammatory drug in adults (Melani et al, 2023; Gonzales et al, 2002). Our findings suggest that dexamethasone may still enhance surfactant production in regenerating adult lungs, raising important considerations for its use as an immunosuppressant in COPD and other lung diseases. Moreover, we observe

**Figure 6. ALVO-MΦ recapitulate IFN-γ–mediated effects on alveolar epithelial cells.**

(A) Schematic of MΦs-ALVO co-culture setup. MΦs are treated with IFN-γ and LPS 24 h before they are put into co-culture with ALVOs growing on top of a transwell. Both compartments contain organoid media. (B) Timeline of MΦs-ALVO co-culture setup. Pretreated MΦs are added fresh every 3–4 days for a 14-day period. Co-cultures are either exposed to ALVO-EM or AT1/2-M. (C) Representative brightfield images of ALVOs in indicated media conditions in the presence of untreated or pretreated MΦs. (D) Viability assay of ALVOs cultured in indicated media conditions in the presence of untreated or pretreated MΦs. Two-sided paired *t*-tests were performed on log-transformed raw values. Data were shown as normalized mean ± SD. Exact *p* values are indicated. (E) Viability assay of ALVOs cultured in indicated media conditions in the presence of untreated MΦs, pretreated MΦ, or pretreated MΦ + RX (1 uM). Matched one-way ANOVA with Geisser–Greenhouse correction, followed by Dunnett's multiple comparisons versus control, was performed on log-transformed raw values. Data were shown as normalized mean ± SD. Exact adjusted *p* values are indicated. (F) Summary of macrophage co-culture findings: Macrophages in the presence of LPS and IFN-γ, representing an infection environment, secrete IFN-γ in concentrations sufficient to have a positive effect on regenerating secAT2/AT2 cells and a detrimental effect on AT1-like cells. Source data are available online for this figure.

evidence of an NE-like phenotype in the DCI-containing condition. This finding was unexpected, as no known connection exists between DCI and the induction of a neuroendocrine lineage. Recently, a human fetal alveolar organoid model also found evidence for NE-like cells in their DCI-containing cultures (Lim et al, 2025). Further experiments are needed to determine whether rare NE-like cells expand in response to DCI or if DCI can induce NE-like features.

YAP signaling has recently emerged as a key factor of AT1 differentiation, but current protocols only show this differentiation effect in vitro in combination with the withdrawal of Wnt-activating agents (Burgess et al, 2024; Lim et al, 2025). Our results show that YAP activation via LATS inhibition is sufficient to promote an AT1-like phenotype even in the presence of Wnt-activators. However, the majority of AT1-like cells have a gene expression profile resembling the AT2-to-AT1 transitional state rather than that of fully differentiated AT1 cells. It is possible that organoid-derived differentiated AT1 cells are underrepresented in the scRNA-Seq dataset due to technical reasons. The typical shape of AT1 cells makes this cell type especially sensitive to dissociation methods necessary to achieve single-cell suspensions. A biological explanation for the underrepresentation is that additional signals or prolonged culture periods might be required for terminal AT1 differentiation. Moreover, 2D cultures may be better suited to capture the characteristic morphology of AT1 cells. AT1 cells are tightly associated with the basement membrane, a relationship that is central to their function. We show that AT1 cell shape varies markedly depending on the ECM coating of the transwells. Although this was not the primary focus of the present study, further optimization of ECM composition to enhance AT1 attachment and differentiation may enable in vitro systems that better support functional testing of AT1 cells. Nevertheless, we posit that the presence of transitional states in our organoids better reflects regenerative conditions and provides a valuable platform for optimizing AT1 differentiation in therapeutic contexts.

MUC5B is expressed in SMGs and the distal airways of human lungs, typically restricted to club cells under homeostatic conditions (Okuda et al, 2019). However, MUC5B expression in AT2 cells has been observed in disease, particularly in idiopathic pulmonary fibrosis (Hancock et al, 2018; Yao et al, 2024). In ALVOs, we identified a subpopulation of cells co-expressing AT2 markers and *MUC5B*, along with elevated *SCGB3A1* and *CEA-CAM6* expression, markers linked to secretory cells (Basil et al, 2022; Kadur Lakshminarasimha Murthy et al, 2022). This suggests that MUC5B expression in these cells may represent an adaptive feature of alveolar regeneration, potentially amplified by organoid pro-proliferative and stemness-inducing organoid culture conditions. Whether this co-expression occurs physiologically during alveolar repair or represents a culture artifact requires further investigation.

Our findings on IFN-γ provide insights into the complex interplay between tissue repair and inflammation. IFN-γ, a key cytokine primarily released by T cells in response to viral and bacterial infections, is elevated in COPD patients compared to healthy controls (Grumelli et al, 2004; Saetta et al, 2002) and further increases during infection-triggered exacerbations (Singh et al, 2010; Makris et al, 2008). The literature on the direct effect of IFN-γ on alveolar epithelial cells is conflicting. While one study found that IFN-γ promoted both AT2 proliferation and differentiation to AT1 cells (Zhang et al, 2023), other studies found that the addition of IFN-γ was detrimental to alveolar organoid cultures (Katsura et al, 2020; Wang et al, 2023). This discrepancy may partly arise from differences in the IFN-γ concentrations used: in vitro studies commonly apply ng/ml concentrations, whereas physiological estimates for IFN-γ levels during acute and chronic lung inflammation fall within the pg/ml range (Lauw et al, 1999; Aliberti et al, 2016). In our study, IFN-γ exhibited a consistently negative impact on AT1-like cells across a variety of concentrations tested, while its effect on alveolar progenitors (AT2 and secAT2 cells) varied depending on dose and exposure duration, with lower concentrations promoting growth of the cells. Notably, we found that the alveolar progenitor cells are protected from the cytotoxic effect of IFN-γ through upregulation of BIRC3. Additionally, we demonstrated that IFN-γ secreted by co-cultured stimulated MΦ was sufficient to replicate the observed effects, adding physiological relevance to our findings. These results imply that blocking IFN-γ signaling to prevent tissue destruction in emphysema patients may inadvertently impair alveolar regeneration. A deeper understanding of the pathways that render AT1 cells vulnerable and protect alveolar progenitor cells in IFN-γ-rich environments could inform the development of therapeutic strategies to enhance alveolar repair while effectively managing inflammation.

It should be noted that we observed marked heterogeneity between organoid lines in both marker expression and the magnitude of responses to stimuli such as IFN-γ. This variability likely reflects differences between donors and is therefore expected when working with primary human material. Organoid models derived from adult primary cells, which represent a more mature cellular state than fetal or iPSC-derived lung cells, are particularly valuable for modeling age-related lung diseases and for identifying therapeutic strategies to enhance alveolar regeneration in adults. At the same time, the intrinsic heterogeneity between primary cells from different donors can pose challenges and represent a limitation of the system. Elucidating the sources of these

heterogeneous responses may provide insight into patient-to-patient variability and ultimately help inform patient stratification in the future.

In conclusion, our study presents a thoroughly characterized ALVO model for investigating alveolar regeneration under inflammatory conditions. A key strength of this model is its controlled design, where the sole variable between alveolar progenitor- and AT1-promoting conditions is the addition of LATS-i. This feature minimizes confounding factors and enables precise mechanistic studies. Our platform holds promise for advancing drug screening efforts to identify factors that promote alveolar regeneration and AT1 differentiation, particularly in the context of inflammatory diseases such as COPD.

# Methods

### Reagents and tools table

| Reagent/resource | Reference or source | Identifier or catalog number |
| --- | --- | --- |
| **Experimental models** | | |
| Lung tissue | Diakonessenhuis Utrecht | |
| PBMCs | Sanquin Amsterdam | |
| **Recombinant DNA** | | |
| N/A | | |
| **Antibodies** | | |
| anti-NGFR-PE | BioLegend | Cat#345106 |
| anti-EPCAM-APC | BioLegend | Cat#369810 |
| anti-CEACAM6-PE | Invitrogen | Cat#12-0667-42 |
| Mouse IgG1 kappa Isotype Control PE | Thermo Fisher Scientific | Cat#12-4714-82 |
| anti-AGER | R&D Systems | Cat#AF1145 |
| anti-SCGB1A1 | R&D Systems | Cat#MAB4218 |
| anti-GPRC5A | Abbexa | Cat#abx005719 |
| anti-HT2-280 | Terrace Biotech | Cat#TB-29AHT2-280 |
| anti-KRT5 | BioLegend | Cat#905501 |
| anti-SFTPC | Santa Cruz | Cat#sc-518029 |
| anti-CEACAM6 | Abcam | Cat#ab78029 |
| **Oligonucleotides and other sequence-based reagents** | | |
| PCR primers | This study | See table EV2 |
| **Chemicals, Enzymes and other reagents** | | |
| AdDMEM/F12 | Gibco | Cat#12634010 |
| Pen/Strep | Gibco | Cat#15140-122 |
| HEPES | Gibco | Cat#15630-056 |
| L-GlutaMAX | Gibco | Cat#35050-038 |
| Primocin | InvivoGen | Cat#Ant-pm-1 |
| B27 | Gibco | Cat#17504-044 |
| N-Acetylcysteine | Sigma-Aldrich | Cat#A9165 |
| Heparin | Sigma-Aldrich | Cat#H3149 |

| Reagent/resource | Reference or source | Identifier or catalog number |
| --- | --- | --- |
| CHIR9902 | Tocris | Cat#4423 |
| SB431542 | Selleckchem | Cat#S1067 |
| BIRB796 | MedChemExpress | Cat#HY-10320 |
| FGF-10 | R&D systems | Cat#345-FG/CF |
| HRG1-b1 | Peprotech | Cat#100-03 |
| Dexamethasone | Sigma-Aldrich | Cat#D4902 |
| cAMP | Sigma-Aldrich | Cat#B7880 |
| LATS-i | MedChemExpress | Cat#HY-138489 |
| EGF | Peprotech | Cat#AF-100-15 |
| Y-27632 | AbMole Bioscience | Cat#M1817 |
| IL-1b | BioLegend | Cat#579402 |
| Collagenase | Sigma-Aldrich | Cat#C9407 |
| DNase | Sigma-Aldrich | Cat#DN25 |
| Red blood cell (RBC) lysis buffer | Roche | Cat#11814389001 |
| Basement membrane extract (BME) | R&D Systems | Cat#3533-005-02 |
| TrypLE | Gibco | Cat#2605010 |
| FBS | Sigma-Aldrich | Cat#F7524 |
| DAPI | Sigma-Aldrich | Cat#10236276001 |
| LysoTracker Green | Fisher Scientific | Cat#11594976 |
| Nile Red | Sigma-Aldrich | Cat#72485 |
| IFN-γ | Peprotech | Cat#300-02 |
| Ruxolitinib (RX) | MedChemExpress | Cat#HY-50856 |
| LCL161 | Selleckchem | Cat#S7009 |
| DAPT | Sigma-Aldrich | Cat#D5942 |
| FGF7 | Peprotech | Cat#100-19 |
| Human serum (HS) | Sigma-Aldrich | Cat#H4522 |
| IL-1A | Peprotech | Cat#200-01 A |
| IL-6 | Peprotech | Cat#200-06 |
| IL-33 | Peprotech | Cat#200-33 |
| IL-4 | Peprotech | Cat#200-04 |
| IL-10 | Peprotech | Cat#200-10 |
| IL-13 | Peprotech | Cat#AF-200-13 |
| IL-22 | Peprotech | Cat#200-22 |
| Invasin | Wijnakker et al.[33] | |
| Collagen I | Advanced Biomatrix | Cat#5074 |
| Dispase | Gibco | Cat#17105041 |
| Phalloidin | Sigma-Aldrich | Cat#65906 |
| Lymphoprep | Stemcell Technologies | Cat#18061 |
| Cellbanker freezing media | Amsbio | Cat#11888 |
| RPMI 1640 medium | Gibco | Cat#61870010 |
| M-CSF | Peprotech | Cat#300-25 |
| *E.coli* LPS | Sigma-Aldrich | Cat#L4391 |

| Reagent/resource | Reference or source | Identifier or catalog number |
|---|---|---|
| *P. aeruginosa* LPS | Sigma-Aldrich | Cat#L9143 |
| 1,4-Dithiothreitol | Sigma-Aldrich | Cat#D9779 |
| PneumaCult™ Alveolar Organoid Expansion Media | Stemcell Technologies | Cat#100-0847 |
| PneumaCult™ Alveolar Organoid Differentiation Medium | Stemcell Technologies | Cat#100-0861 |
| DRAQ5 | Biostatus | Cat#DR05500 |
| **Software** | | |
| GraphPad Prism 10 | | |
| R (v.4.4.1) | | |
| CellRanger (v.7.2.0) | Zheng et al, 2017 | |
| SoupOrCell algorithm (v2.5) | Heaton et al, 2020 | |
| Seurat package (v.5.1.0) | Hao et al, 2024 | |
| DESeq2 package (v.1.44.0) | Love et al, 2014 | |
| **Other** | | |
| Cell Titer Glo (CTG) 3D Cell Viability Assay | Promega | Cat#G9681 |
| Cytotoxicity assay | Promega | Cat#G1780 |
| RNA extraction kits | Quiagen | Cat#74106, Cat#74536 |
| High-Capacity RNA-to-cDNA kit | Applied Biosystems | Cat#16271891 |
| iQ SYBRGreen supermix | Bio-Rad | Cat#1725006CUST |
| 500 μm filter | pluriSelect | Cat#43-50500-03 |
| 100 μm filter | Greiner Bio-One | Cat#542000 |
| Suspension culture plates | Greiner Bio-One | Cat#657185, Cat#665102, Cat#662102 |
| White-walled 96-well plates | Greiner Bio-One | Cat#655098 |
| Black-bottom, black-walled 96-well plate | Greiner Bio-One | Cat#655076 |
| Transwell inserts | Greiner Bio-One | Cat#662641, Cat#665641 |
| Cell scrapers | Greiner | Cat#541070 |

## Human samples

Distal human lung tissue samples were obtained as non-cancerous adjacent tissues from tumor resections at the Diakonessenhuis Hospital, Utrecht (Utrecht, The Netherlands). Samples were collected at least 5 cm from the tumor. Patients had no diagnosed lung diseases other than lung cancer and had received no cancer treatment before surgery. Informed consent was acquired from all donors. The study was approved by the ethical committee and was in accordance with the Declaration of Helsinki and Dutch law. This study is compliant with all relevant ethical regulations regarding research involving human participants. As patient samples were anonymized, sex, gender, age, race and other information were not recorded and are not available.

## Tissue processing for organoid cultures

Non-tumor lung tissue from lobectomies were kept in saline solution on ice or in washing media (Table EV1) at 4 °C until processing. Tissues were transferred to a sterile dish and minced using scissors and razor blades. The mechanically dissociated tissue was transferred to a 50 ml conical tube, and 10 ml prewarmed digestion buffer (washing media + 0.5 mg/ml collagenase (Sigma-Aldrich #C9407)) was added per gram of tissue, but not exceeding 25 ml per tube. Tissue was incubated in the digestion buffer on a shaker at 37 °C for 25 min. Following incubation, DNase (50 ug/ml; Sigma-Aldrich #DN25) was added, and tubes were placed on ice for 5 min. The digested tissue was filtered through a 500 μm filter (pluriSelect #43-50500-03) placed on a 50 ml tube, using the flat end of a syringe plunger to push the tissue through the filter. The filter was rinsed with 10 ml of cold washing media, and the process was repeated with a 100 μm filter (Greiner Bio-One #542000). Filtered suspensions were centrifuged at $400 \times g$ for 5 min, and supernatants were carefully aspirated without disturbing the cell pellet. Cell pellets were resuspended in ~1 ml of red blood cell (RBC) lysis solution (Roche #11814389001) per gram of tissue and incubated at room temperature (RT) for 5 min. The lysis reaction was quenched with 15 ml of washing buffer (Table EV1), the cell suspension was centrifuged at $400 \times g$ for 5 min, and the supernatant was aspirated. If the cell pellet was still red, the RBC lysis step was repeated no more than once. The resulting cell pellet typically has a color ranging from white to dark gray, depending on the amount of pollution present in the tissue. The pellet was resuspended thoroughly in 1 ml of washing buffer (Table EV1). If large clumps were observed, the suspension was filtered through a 100 μm filter (Greiner Bio-One #542000) and spun down again. The number of live cells was determined using Trypan Blue, and cells were resuspended in a mixture of 20% washing buffer (Table EV1) and 80% BME (R + D Systems #3533-005-02) at a concentration of ~3000 cells/μl. Drops of 20–40 μL were plated on pre-warmed suspension culture plates (Greiner Bio-One #657185, #665102, #662102). After the BME solidified (10–15 min), pre-warmed organoid media supplemented with Y-27632 (10 uM; AbMole Bioscience #M1817) and optionally IL-1β (10 ng/ml; BioLegend #579402) was added for the first few days. The media was changed every 2–4 days. Organoid cultures were tested for mycoplasma contamination regularly, and tests were always negative.

## Making single-cell suspensions and passaging organoids

Organoids were passaged approximately every 14 days. After aspirating media, ~1 ml TrypLE (Gibco 2605010) was added to one well of a six-well plate (scale volume down for smaller formats), and BME domes were disrupted mechanically by pipetting up and down, and organoid suspension was collected into a 15 ml conical tube and incubated at 37 °C for 2–4 min. For difficult-to-dissociate organoid cultures (in the presence of LATS-i), the incubation time was increased in 4 min increments, up to 30 min, with periodical pipetting. Organoids were disrupted by pipetting up and down rigorously, without introducing bubbles, using a 1000 μl pipette tip or a Pasteur glass pipette. The dissociation process was monitored under a brightfield microscope periodically, and the reaction was quenched by adding 10 ml cold washing media (Table EV1) when the suspension was mostly single cells. The number of live cells was determined using Trypan Blue. Cells were spun down at $400 \times g$ for 5 min, and the media was aspirated. If a BME layer was visible above the cell pellet, the pellet

was resuspended in cold washing media and spun again. When no BME was visible above the cell pellet anymore, the cells were resuspended in a mixture of 20% washing buffer (Table EV1) and 80% BME (R&D Systems #3533-005-02) at a concentration of 300–400 cells/μl. Drops of 10–25 μl were plated on pre-warmed suspension culture plates (Greiner bio-one #657185, #662102, #665102). For six-well plates, ~400 ul cell/BME mix was plated per well, for 12-well plates ~120 ul, and for 24-well plates ~60 ul. After the BME solidified (10–15 min), pre-warmed organoid media supplemented with Y-27632 (10 uM; AbMole Bioscience #M1817) was added for the first few days. The media was changed every 2–4 days.

## FACS of organoids

Organoids were made into single-cell suspensions as outlined in "Making single-cell suspensions and passaging organoids" and resuspended in FACS buffer [PBS + 2% FBS (Sigma-Aldrich #F7524) + 2 mM EDTA] at 10 million cells/ml. Cells were divided into several tubes for fluorophore-minus-one (FMO) or isotype staining controls. DAPI (Sigma-Aldrich #10236276001), anti-NGFR-PE (1:100; BioLegend #345106), anti-EPCAM-APC (1:100; BioLegend #369810), anti-CEACAM6-PE (1:100; Invitrogen #12-0667-42), or isotype control-PE (1:100; Thermo Fisher Scientific #12-4714-82) were added, and cells were incubated on ice for 15–30 min. Cells were spun down at $400 \times g$ for 5 min, supernatant was discarded, the pellets were resuspended in FACS buffer, and cell suspensions were strained into FACS tubes. Cells were gated on single, DAPI-cells, and then either sorted for EPCAM+, NGFR-cells, or CEACAM6+/- cells into eppendorf tubes containing ALVO-EM (Table EV1) using a BD FACSAria Fusion. Cells were plated in a mixture of 20% washing buffer (Table EV1) and 80% BME (R&D Systems #3533-005-02) at 300–400 cells/ul as outlined in "Making single-cell suspensions and passaging organoids".

## Flow cytometry analysis of LysoTracker and Nile red stained organoid cells

Organoids that were cultured in ALVO-EM (Table EV1) for 10 days followed by AT2-MM (Table EV1) for 7 days were made into single-cell suspensions as outlined in "Making single-cell suspensions and passaging organoids" and resuspended in washing media to a concentration of 1 million cells/ml. DAPI (Sigma-Aldrich) and either LysoTracker Green (100 nM; Fisher Scientific #11594976) or Nile Red (500 ng/ml; Sigma-Aldrich #72485) were added, and cells were incubated while rotating at 37 °C for 45 or 30 min, respectively. After the incubation time, cells were spun down at $400 \times g$ for 5 min, supernatant was removed, and the cells were washed with FACS buffer [PBS + 2% FBS (Sigma-Aldrich #F7524) + 2 mM EDTA]. After another spinning step, the cell pellet was resuspended in FACS buffer and strained into FACS tubes. Cells were gated on single, DAPI- cells, and the median fluorescent signal of lysotracker ([488] 530/30) and Nile Red ([561] 585/15) was measured on a BD LSRFortessa flow cytometer.

## Viability assay: Cell Titer Glo (CTG) 3D

Single cells were plated at a density of 300–400 cells/μl in a mixture of 20% washing buffer (Table EV1) and 80% BME (R&D Systems #3533-005-02), with a total volume of 5 μl (Greiner bio-one #655098) or 10 ul per well (STEMCELL Technologies #200-0562) in 96-well plates, or 60 ul per well in 24-well plates. The BME was allowed to solidify for 10–15 min at 37 °C before adding 100 μl or 400 ul of prewarmed media, respectively. The media conditions containing varying concentrations of IFN-γ (Peprotech #300-02) and RX (MedChemExpress #HY-50856) are outlined in the figures and were added from day 0 on, with Y-27632 (10 uM; AbMole Bioscience #M1817) present for the first few days. For each condition, 2–4 wells were plated as technical replicates. Media was replaced every 2–4 days. On the days of analysis, the manufacturer's protocol for the CTG assay (Promega #G9681) was followed. Briefly, CTG solution was added to the media containing wells at a 1:1 ratio, and the plate was protected from light and shaken for 10 min before incubating for 10–20 min. A 100 μl sample of the media/CTG mix was then transferred from each well into a black-bottom, black-walled 96-well plate (Greiner Bio-One #655076). Luminescence signals were recorded using a standard plate reader. Measurements on day 0 immediately after plating were done to account for differences in plating between organoid lines. Subsequent measurements were normalized to day 0 values, and normalized again to the control group for each organoid line.

## Cytotoxicity assay of IFN-γ-treated organoids

Cells were plated for the cytotoxicity assay the same way as described for the viability (CTG) assay. IFN-γ (10 ng/ml; Peprotech #300-02), and LCL161 (10 uM; Selleckchem #S7009) where applicable, were added to the media on day 11 of culture, and the cytotoxicity assay (Promega #G1780) was performed on day 14 according to the manufacturer's protocol. Briefly, 50 ul of media was transferred from each well into a new clear bottom 96-well plate, an equal volume of CytoTox reagent was added, and the plate was shaken for 5 min. The mixture was incubated at RT for 30 min. Subsequently, 50 ul stop solution was added, the plate was shaken for 5 min, and absorbance was measured at 490 nm using a standard plate reader.

## Small molecule, growth factors, human serum, cytokine, and SCT media treatments

Small molecules LATS-i (10 mM; MedChemExpress #HY-138489), DAPT (1 uM; Sigma-Aldrich #D5942), and RX (100 nM, 1 uM, 10 uM; MedChemExpress #HY-50856), growth factors EGF (50 ng/ml; Peprotech #AF-100-15) and FGF7 (10 ng/ml; Peprotech #100-19), and HS (10%; Sigma-Aldrich #H4522) were added to organoid culture according to the timelines stated in the figures. Cytokines IL-1A (Peprotech #200-01 A), IL-1B (BioLegend #579402), IL-6 (Peprotech #200-06), IL-33 (Peprotech #200-33), IL-4 (Peprotech #200-04), IL-10 (Peprotech #200-10), IL-13 (Peprotech #AF-200-13), IL-22 (Peprotech #200-22), and IFN-γ (Peprotech #300-02), were added to organoid cultures at a concentration of 10 ng/ml unless stated otherwise. PneumaCult™ Alveolar Organoid Expansion Media (STEMCELL Technologies #100-0847) and PneumaCult™ Alveolar Organoid Differentiation Medium (STEMCELL Technologies #100-0861) were used as outlined in the results section.

## Transwell cultures

Transwell inserts (Greiner bio-one #662641) were coated with 10 μg/ml Invasin (Wijnakker et al, 2025) in PBS or 2% Collagen I

(Advanced Biomatrix #5074) in washing media (Table EV1) at 4 °C overnight, or with 5% BME (R&D Systems #3533-005-02) in washing media (Table EV1) at 37 °C for 30 min. Day 14 organoids cultures were processed to single-cell suspensions as outlined in "Making single-cell suspensions and passaging organoids" and 100k–200k cells were plated per transwell. Media was added to the bottom and top of the transwell until cell layer was confluent, usually after 1 week. At this point, media was removed from the upper compartment to establish air–liquid interface (ALI) conditions. Transwells were fixed after approximately one week of ALI culture.

## Immunohistochemistry for organoid slides, transwells, and whole-mount samples

Dispase (1 mg/ml; Gibco #17105041) was added to the media of the submerged organoids and the plate was incubated at 37 °C for 45–60 min to dissolve the BME. The organoid suspension was carefully collected into a 15 ml conical tube using a 1 ml pipette tip with cut off tip to prevent shearing of the larger organoids. The tube and tip were pre-coated with FBS (Sigma #F7524) to prevent sticking of the organoids to the plastic. Cold washing media (Table EV1) was added to the tube, and the organoids were spun down at $50 \times g$ for 3 min. The supernatant was removed and the organoids were fixed with 4% paraformaldehyde for 30–60 min at RT. For transwell cultures, the cells were fixed inside of the transwells without the addition of dispase, and the membrane with cells was removed from the transwell for downstream processing.

To prepare slides, the organoids and transwell samples were dehydrated, paraffin embedded, and sectioned. Standard H&E stainings and immunohistochemistry was performed using antibodies against AGER (1:500; R&D Systems #AF1145), SCGB1A1 (1:250; R&D Systems #MAB4218), GPRC5A (1:1000; Abbexa #abx005719), HT2-280 (1:500; Terrace Biotech #TB-29AHT2-280), KRT5 (1:2000; Biolegend #905501), CEACAM6 (1:100, Abcam #ab78029), and SFTPC (1:250; Santa Cruz #sc-518029). Phalloidin (1:1000; Sigma-Aldrich #65906) was used to visualize cell boundaries.

Organoids were prepared for whole-mount staining as described in detail elsewhere (Pleguezuelos-Manzano et al, 2020). HT2-280 (1:250; Terrace Biotech #TB-29AHT2-280) was added overnight at 4 °C, and Nile red (500 ng/ml; Sigma-Aldrich #72485) was added for 30 min at RT.

Whole-mount-stained organoids and sections were imaged using a Leica Sp8 confocal microscope. Fluorescent images were processed using LAS X software.

## MΦ-organoid co-cultures and viability assay (CTG)

Human PBMCs were obtained from buffy coats (Sanquin Amsterdam) via density gradient using Lymphoprep (STEMCELL Technologies #18061) as described by the manufacturer. PBMCs were then frozen in Cellbanker freezing media (Amsbio #11888) at −80 °C until further use. To obtain MΦs, the PBMCs were seeded in an RPMI 1640 medium (Gibco #61870010) supplemented with 10% FBS (Sigma #F7524) and 5% PenStrep (Gibco #15140-122) atop a standard 20 cm plastic tissue culture dish and left to adhere for at least 2 h at 4 °C and 37% $CO_2$. The non-adherent cells were then thoroughly washed away, and the remaining monocyte-

enriched adherent cells were gently detached using cell scrapers (Greiner #541070). For the co-culture experiments, the monocytes were either seeded at a density of $3 \times 10^5$ or $2 \times 10^5$ cells in 12- or 24-well cell culture plates, respectively. To differentiate monocytes into MΦs, the media was supplemented with 40 ng/ml recombinant human M-CSF (Peprotech #300-25) for 7 days, with media refresh every 2–3 days. To polarize them towards their pro-inflammatory phenotype, they were stimulated for 24 h with a combination of 100 ng/ml LPS [50 ng/ml *E.coli* LPS (Sigma-Aldrich #L4391) and 50 ng/ml *P. aeruginosa* LPS (Sigma-Aldrich #L9143)] and 10 ng/ml IFN-γ (Peprotech #300-02). On the day of co-culture, the media was aspirated, and the MΦs were washed with PBS, before the organoid media and the organoids in transwells were added to the wells.

To grow organoids in transwells, single-cell suspensions of organoids as outlined in "Making single-cell suspensions and passaging organoids" were plated in a mixture of 20% washing buffer (Table EV1) and 80% BME (R&D Systems #3533-005-02) at 300–400 cells/ul into a transwell insert (Greiner Bio-One #662641 and #665641) in a total volume of 30 or 20 ul for 12 or 24 well inserts, respectively. After the BME had solidified, the transwells were transferred to the plate containing the pretreated MΦs, and organoid culture media [ALVO-EM or AT1/2-M (Table EV1); with or without 1 uM RX (MedChemExpress #HY-50856)] was added on top of the transwell and to the MΦs containing bottom compartment. The organoids in transwells were transferred to a fresh plate containing pretreated MΦs every 3–4 days, and the media was refreshed in both compartments at the same time.

On day 14, the transwells with organoids were transferred to a fresh plate, the media was aspirated from the top and replaced with fresh media. The manufacturer's protocol for the CTG assay (Promega #G9681) was followed. Briefly, an equal volume of CTG solution was added to each transwell, and the plate was protected from light and shaken for 10 min before incubating for 5 min and additional mixing by pipetting up and down. For each transwell, 3–4 100 μl samples of the media/CTG mix were transferred into a black-bottom, black-walled 96-well plate (Greiner Bio-One #655076) as technical replicates. Luminescence signals were recorded using a standard plate reader. Measurements on day 0 immediately after plating were done to account for differences in plating between organoid lines. Subsequent measurements were normalized to day 0 values, and normalized again to the control group for each organoid line.

## Transmission electron microscopy

Medium was removed from organoid-containing wells, and cold washing media with 10% FBS was added to the wells. The BME was carefully disrupted using a 1 ml pipette tip, and the organoid suspension was transferred to Eppendorf tubes. Tubes were incubated on ice for 30 min, followed by centrifugation at $300 \times g$ for 5 min. The supernatant was discarded. For fixation, the organoid pellet was washed once in fixative buffer (1.5% glutaraldehyde/0.067 M cacodylate buffered to pH 7.4/1% sucrose) and then incubated in fresh fixative buffer at RT for 3 h. The organoids were washed with washing buffer 2 (0.1 M cacodylate (pH 7.4)/1% sucrose) and finally resuspended in fresh washing buffer 2 and kept at 4 °C until further processing.

Then, organoids were incubated in 1% osmium tetroxide and 1.5% K4Fe(CN)6 in 0.1 M sodium cacodylate (pH 7.4) for 1 h at 4 °C. Next, organoids were dehydrated in ethanol (70%, 90%, up to 100%) infiltrated with Epon resin for 2 days, and finally embedded in the same resin and polymerized at 60 °C for 48 h. Ultrathin sections were cut on a Leica Ultracut UCT ultramicrotome (Leica Microsystems) using a diamond knife (Diatome) and mounted on Formvar-coated copper grids. Subsequent staining of the sections with 2% uranyl acetate in 50% ethanol and lead citrate allowed visualization under a Tecnai T12 Electron Microscope equipped with an Eagle 4k × 4k CCD camera (Thermo Fisher). Images were stitched, uploaded, shared and annotated using Omero and PathViewer.

## RNA extraction

Media was aspirated from the organoid cultures, and RNA lysis buffer with 1,4-dithiothreitol (40 mM; Sigma-Aldrich #D9779) was added directly to the organoid-containing BME domes. RNA extraction was performed using the RNA kits (Qiagen #74106 and #74536) according to the manufacturer's protocol.

## cDNA preparation and RT-qPCR analysis

cDNA was prepared using the High-Capacity RNA-to-cDNA (Applied Biosystems #16271891) kit according to the manufacturer's protocol. cDNA was diluted 1:10 with Milli-Q water, and RT-qPCR was performed using the iQ SYBRGreen supermix (Bio-Rad #1725006CUST) and gene-specific primers (Table EV2) in a CFX Connect Real-Time PCR machine (Bio-Rad). The results were analyzed by normalization to the housekeeping gene ACTB ($2^{-\Delta CT}$) and, depending on the experiment, to a reference sample ($2^{-\Delta\Delta CT}$).

## Bulk mRNA sequencing

Bulk RNA-Seq was performed on three organoid lines (Lu37, Lu38, and Lu39) cultured in ALVO-EM or AT1/2-M (Table EV1) and treated with IFN-γ (10 ng/ml; Peprotech #300-02) for 4 h or untreated. RNA was extracted as described above. For library preparation, a minimum of 100 ng of total RNA was used per condition. The Utrecht Sequencing Facility (USEQ, Utrecht, The Netherlands) carried out the library preparation using the TruSeq Stranded mRNA polyA kit, followed by sequencing on the Illumina NextSeq2000 platform. Reads were mapped to the Human genome (GRCh38) using STAR mapping software (v. 2.7.5c). Read counts were assigned using featureCounts (v2.03.) from the subreads package. Raw count data were imported into R using the read.csv function. The DESeq2 (Love et al, 2014) package (v.1.44.0) was used to create a DESeq dataset. The three lines were treated as replicates, and differentially expressed genes were determined for untreated vs. IFN-γ-treated organoids for each of the two media conditions. Genes with ≤10 reads across all conditions, a p-adjusted value ≥0.05, or a log2 fold change ≤1 were filtered out.

## 10x Genomics scRNA-Seq

Organoids from three donors (Lu37, Lu38, and Lu39) at p5, cultured in the three media conditions ALVO-EM, AT2-MM, and AT1/2-M, were made into single-cell suspensions as outlined in "Making single-cell suspensions and passaging organoids" and stained with DAPI (Sigma-Aldrich #10236276001) and DRAQ5 (Biostatus #DR05500) for 10 min in FACS buffer [PBS + 2% FBS (Sigma #F7524) + 2 mM EDTA]. After the incubation time, cells were spun down, resuspended in FACS buffer, and strained into FACS tubes. Live cells (DAPI-/DRAQ5+) were sorted into PBS using a BD FACSAria Fusion. Cells from the same condition but different donors were mixed in equal parts, and 25,000 cells per condition were subjected to droplet-based scRNA-seq using the 10x Genomics platform. Libraries were prepared using the 10x Genomics Chromium 3' Gene Expression solution v3 on a GEMX machine and sequenced on a NovaSeq6000 (Illumina). Reads were mapped to the Human genome (CRCh38) using CellRanger (Zheng et al, 2017) software (v.7.2.0). Reads were demultiplexed based on genotype using the SoupOrCell algorithm (v2.5) (Heaton et al, 2020).

## Single-nucleotide polymorphism (SNP) analysis and scRNA-Seq genotype matching

To match the sequenced cells to the three donors (Lu37, Lu38, and Lu39) we performed SNP analysis. Oligo ligation assay (OLA) SNP typing (Luminex) was performed following manufacturers recommendations. Probes were designed to target population-wide SNPs with an incidence of around 50%, covering all major chromosomes and with coverage in whole genome and exome sequencing as well as in RNA sequencing approaches such as polyA-enriched RNA sequencing or total RNA sequencing. Detailed target information and genomic coordinates can be found at: https://github.com/Hubrecht-Clevers/SNP_genotyping_airway/design.txt. DNA was isolated from the three organoid lines (Lu37, Lu38, and Lu39) that were used for scRNA sequencing. OLA probes were used to amplify a genomic target before hybridization to MagPlex-TAG microspheres (Luminex). Signal intensities were measured using a Luminex-MagPix (Luminex). Genotype profiles were generated using an in-house generated script https://github.com/Hubrecht-Clevers/SNP_genotyping_airway/SNP_profiles_processing.R. Genotypes from single-cell samples were extracted from SoupOrCell vcf-files generated during genotype demultiplexing. These genotypes were compared with OLA SNP typing results. Sample IDs were assigned based on the Euclidean distance between an OLA SNP typing result and SoupOrCell genotype.

## scRNA-Seq data preprocessing

CellRanger and Souporcell output files were processed using the Seurat (Hao et al, 2024) package (v.5.1.0) in R (v.4.4.1). Firstly, cells identified as doublets or with an unassigned identity by Souporcell were removed from further analyses. Transcripts found in less than three cells were removed. High-quality cells were subsequently obtained by filtering out cells expressing less than 200 or more than 7500 transcripts, and a mitochondrial gene percentage higher than 10%.

## scRNA-Seq data integration with a healthy tissue dataset

To enable identification of cell types in the organoids and comparison of their transcriptome to in vivo lung tissue, we integrated all organoid datasets with the reference tissue scRNA-seq

dataset published by Sikkema et al (2023), which contains curated cell type annotation (Sikkema et al, 2023). Prior to integration, only epithelial cells were subsetted from the object based on the authors' annotation (ann_level_1). Thereafter, the object was downsampled to contain the same number of cells as the combined organoid datasets (29,064 cells). Integration of the three organoid datasets [ALVO-EM, AT2-MM, and AT1/2-M) (Table EV1)] with the downsampled tissue dataset was performed using Reciprocal PCA (RPCA) with the default parameters. After computing PCA dimensions, uniform manifold approximation and projections (UMAPs) were rendered using dims = 1:35. The data has been deposited at Gene Expression Omnibus (GSE287209) and are publicly available as of the date of publication.

### scRNA-Seq clustering, subsetting, and differential expression analysis

Using Seurat's FindClusters function with a resolution of 0.8, we obtained 31 cell clusters. Given the available information on the cell identities assigned by Sikkema et al (2023), tissue-derived clusters could be easily distinguished. For a deeper characterization of organoid-derived cells, we subsequently subsetted all clusters containing organoid cells (i.e., clusters 0, 1, 3, 4, 5, 7, 8, 9, 11, 16, 17, 21, 23, 24, 26, and 30) (Fig. EV4A). Cluster 28 was exclusively tissue-derived and constituted an impure population of AT2 cells combined with alveolar macrophages (MARCO+), it was excluded. Subsetting and re-scaling of the data yielded 20 clusters. Clusters 0, 1, 2, and 3 contained the highest numbers of tissue-derived cells annotated as "AT2". Therefore, these were merged and labeled as "AT2". Since cells in clusters 4, 5, and 16 expressed both AT2 (e.g., *SFTPC*) and AT1 markers (e.g., *CAV1*), they were annotated as "trans_AT2-AT1". Clusters 6, 7, 8, and 9 showed the highest expression of secretory markers such as MUC5B and HPGD, hence why they were assigned the identity "secretory AT2". Clusters 10, 12 and 15 were annotated as "cycling" cells. Cluster 11 was annotated as "NE-like", given the high expression of common neuroendocrine markers such as CALCA and ASCL1. Cluster 13 was defined as "AREGhigh AT2" cells, and cluster 14 was annotated as "AT1" cells. Finally, clusters 17, 18, and 19 were identified based on their original tissue annotation as "pre-TB secretory", "NE + Iono-cyte", and "SMG mucous" cells, respectively. Differential gene expression analyses between clusters and samples were performed with FindAllMarkers, using min.pct = 0.25, and logfc.threshold = 0.25.

### Statistical analysis

Data were presented as mean ± standard deviation, with individual data points shown to represent biological replicates (organoid lines from different donors). Statistical analyses were performed using GraphPad Prism 10. For comparisons between two groups, a two-sided paired *t*-test was performed on log-transformed raw values. For comparisons involving multiple groups against a control group, or among multiple groups, a matched one-way ANOVA with Geisser–Greenhouse correction followed by Dunnett's multiple comparisons versus control was performed on log-transformed raw values. Normalization of data to a control group is for visualization purposes only. Exactly *p* values or adjusted *p* values if a post hoc test was used are stated in the figures. No blinding was done. Each experiment was replicated at least once.

## Data availability

Single-cell RNA-seq data have been deposited at GEO at GSE287209 accession number (https://www.ncbi.nlm.nih.gov/geo/query/acc.cgi?acc=GSE287209) and are publicly available as of the date of publication.

The source data of this paper are collected in the following database record: biostudies:S-SCDT-10_1038-S44318-026-00774-4.

## Peer review information

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

## Acknowledgements

We acknowledge the Utrecht Sequencing Facility (USEQ) for providing sequencing service and data. USEQ is subsidized by the University Medical Center Utrecht and the Netherlands X-omics Initiative (NWO project 184.034.019). We thank the Single Cell Genomics core of the Princess Maxima Center (Utrecht, The Netherlands) and the Flow Cytometry Core at the Hubrecht Institute (Utrecht, The Netherlands) for their technical assistance. AFMD has received funding from the European Respiratory Society (ERS) and the European Union's H2020 research and innovation program under the Marie Sklodowska-Curie grant agreement No 847462. The authors are solely responsible for its content; it does not represent the opinion of ERS and the European Commission, and ERS and the EU Commission are not responsible for any use that might be made of data appearing therein. This work was also supported by the Netherlands Organ-on-Chip Initiative, an NWO Gravitation project (024.003.001) funded by the Ministry of Education, Culture and Science of the government of the Netherlands (KB and HC), the Oncode Institute (partly financed by the Dutch Cancer Society) (HC) and the Accelerate Lung Regeneration Consortium BREATH (12.0.18.002) of the Lung Foundation Netherlands (LMvR and HC). CPC is financially supported by the Gravitation Program "Materials Driven Regeneration", funded by the Netherlands Organization for Scientific Research (024.003.013). The authors confirm that this manuscript adheres to the recommendations of the International Committee of Medical Journal Editors (ICMJE).

AI-assisted technologies in the writing process: During the preparation of this work, the authors used ChatGPT-4o to spell- and language-check written text.

## Author contributions

**Antonella F M Dost**: Conceptualization; Formal analysis; Supervision; Funding acquisition; Validation; Investigation; Visualization; Writing—original draft; Writing—review and editing. **Katarína Balážová**: Investigation; Writing—review and editing. **Carla Pou Casellas**: Data curation; Formal analysis; Writing—review and editing. **Lisanne M van Rooijen**: Investigation. **Wisse Epskamp**: Investigation. **Gijs J F van Son**: Data curation. **Willine J van de Wetering**: Investigation. **Carmen Lopez-Iglesias**: Investigation. **Harry Begthel**: Investigation. **Peter J Peters**: Supervision. **Niels Smakman**: Resources. **Johan H van Es**: Funding acquisition; Project administration. **Hans C Clevers**: Supervision; Funding acquisition; Project administration; Writing—review and editing.

Source data underlying figure panels in this paper may have individual authorship assigned. Where available, figure panel/source data authorship is listed in the following database record: biostudies:S-SCDT-10_1038-S44318-026-00774-4.

## Disclosure and competing interests statement

HC is an inventor on patents held by the Royal Netherlands Academy of Arts and Sciences that cover organoid technology. He is currently Head of Pharma Research and Early Development (pRED) at Roche, Basel, Switzerland. HC's full disclosure is given at https://www.uu.nl/staff/JCClevers/. HC is a member of the EMBO Journal editorial advisory board.

# Expanded View Figures

**Figure EV1.  Optimized ALVO conditions allow for long-term expansion of alveolar progenitor populations. Related to Fig. 1.**

(**A**) IF images of AO organoids raised in AO media as a staining control, p0 organoids generated with our method, and p1 organoids after EPCAM+/NGFR- FACS sorting as described in Fig. 1A. Scale bar = 50 um. The right column shows a close-up of the area indicated by boxes in the left column. DAPI = nuclei; SFTPC = AT2 markers; KRT5 = basal cell marker; SCGB1A1 =  club cell marker. (**B**) FACS plot showing sorting strategy of p0 d14 organoid-derived cells gated for DAPI-/EPCAM+ single cells. NGFR-/HT2-280+ (HT2+) and NGFR-/HT2-280- (HT2-) populations were sorted, plated in organoid cultures, and further analyzed in Fig. EV1C+E. (**C**) Brightfield images of HT2+ and HT2- sorted cells (see Fig. EV1B) cultured with HRG1 or EGF. (**D**) Dotplot showing gene expression of selected genes in different lung cell types. Expression data and cell type annotations were taken from the lung atlas Sikkema et al, 2023. (**E**) qPCR analysis of HT2+ and HT2- cells sorted from p0 organoids and cultured with HRG1 or EGF in p1 (see also EV1B+C). AOs are included for comparison. A two-sided paired *t*-test was performed on log-transformed raw values. Data were shown as mean ± SD. Exact *p* values are indicated. (**F**) FACS gating strategy for EPCAM+/NGFR- cell sorting of p0 organoids (left panel). Percentage of NGFR- cells within the EPCAM+ fraction (right panel). Data were represented as mean ± SD. (**G**) qPCR analysis of AT2 (*SFTPC*) and AT1 (*CAV1, AGER*) markers in ALVOs cultured in indicated media conditions for 14 days. A two-sided paired *t*-test was performed on log-transformed raw values. Data were shown as mean ± SD. Exact *p* values are indicated. (**H**) qPCR time course of AT2 (*SFTPA1* and *SFTPC*) and AT1 (*AGER*) markers in day 14-old ALVOs at indicated passages with whole lung lysate as reference (dotted line). (**I**) Dotplot of the top 15 differentially expressed genes (DEGs) for Seurat clusters 0, 1, and 2. Plot showing the clusters can be found in Fig. 1D.

▶

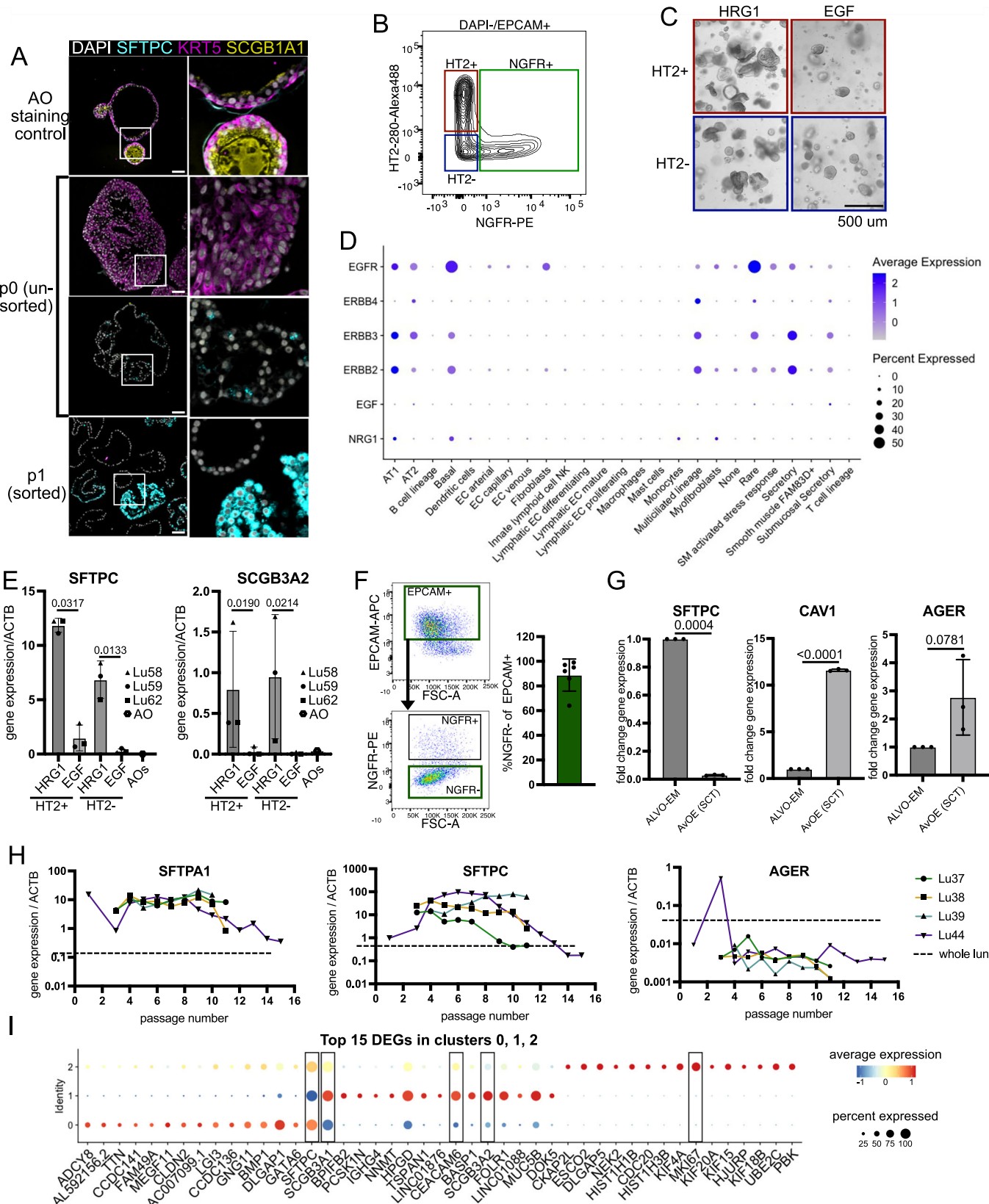

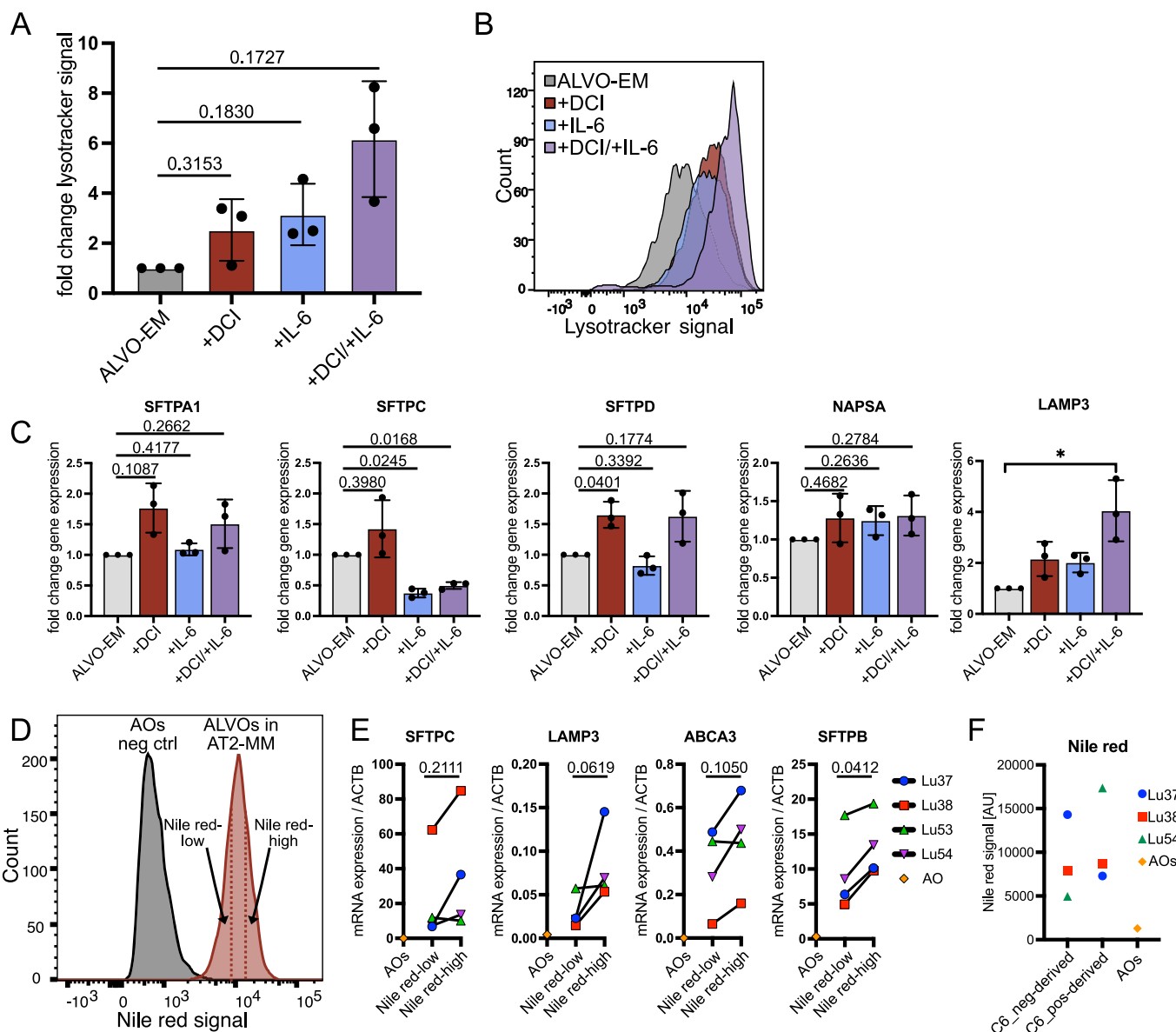

**Figure EV2. Maturation media supports AT2 surfactant secretion with tubular myelin formation. Related to Fig. 2.**

(A) Flow cytometry analysis of lysotracker signal in ALVOs cultured in the indicated conditions. Matched one-way ANOVA with Geisser–Greenhouse correction, followed by Dunnett's multiple comparisons versus control, was performed on log-transformed raw values. Data were shown as normalized mean ± SD. Exact adjusted *p* values are indicated. (B) Representative flow cytometry histogram showing Lyostracker signal distribution in indicated conditions. (C) qPCR analysis of AT2 markers in ALVOs cultured in the indicated conditions as outlined in Fig. 2A. Matched one-way ANOVA with Geisser–Greenhouse correction, followed by Dunnett's multiple comparisons versus control, was performed on log-transformed raw values. Data were shown as normalized mean ± SD. Exact adjusted *p* values are indicated. (D) Representative flow cytometry histogram showing Nile red signal distribution in AOs (negative control) and ALVOs cultured in AT2-MM. Nile red-low and -high cells are indicated by the dotted lines. (E) qPCR analysis of sorted Nile red-low and -high cells from ALVOs cultured in AT2-MM (also see panel D) and AOs (negative control). Two-sided paired *t*-tests were performed on log-transformed raw values. Exact *p* values are indicated. (F) Flow cytometry Nile red signal of C6- (neg) and C6+ (pos) derived cells matured in AT2-MM as outlined in Fig. 2A compared to AO-derived cells.

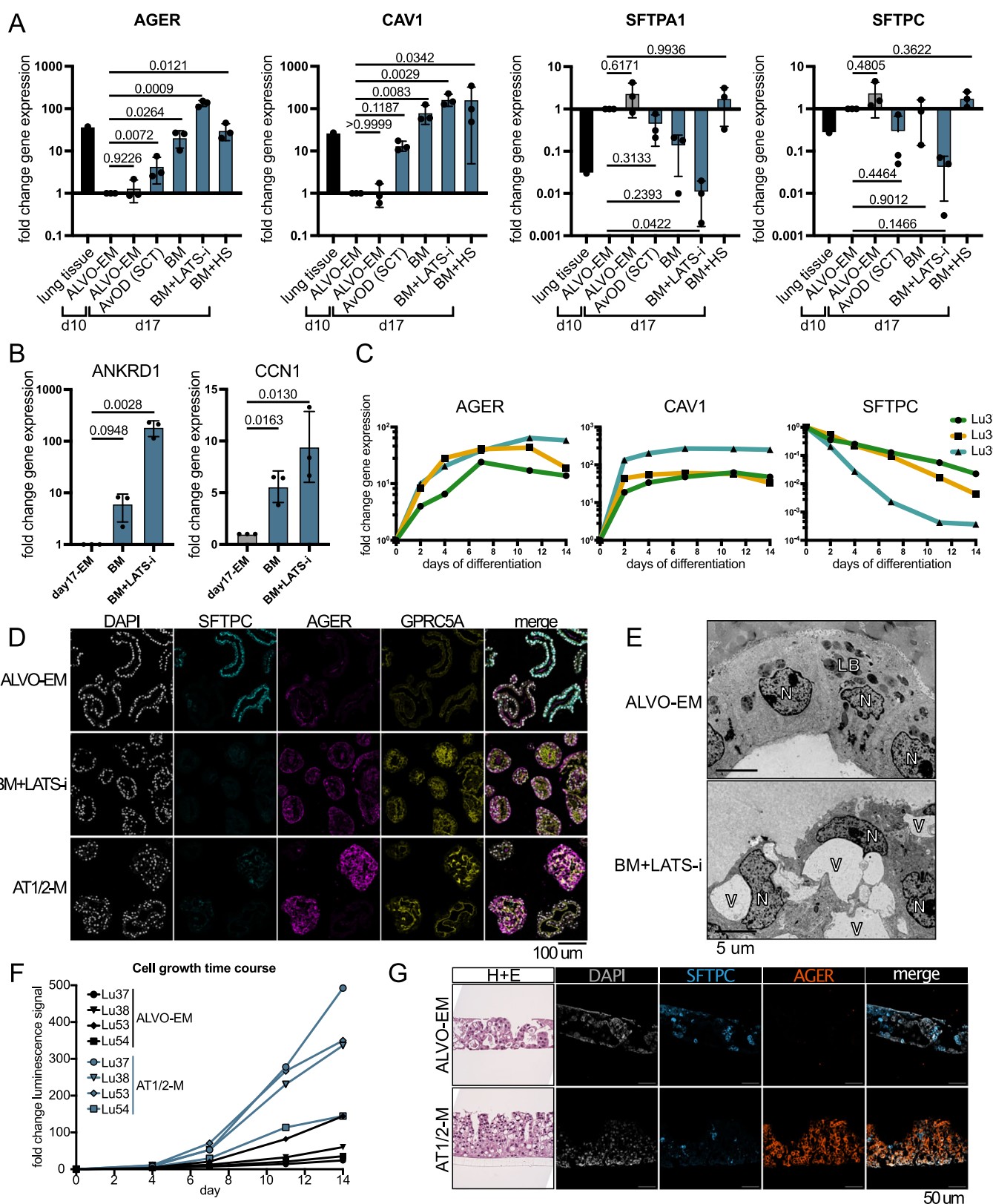

**Figure EV3.  Removal of proliferation factors and inhibition of LATS drives AT1 differentiation. Related to Fig. 3.**

(A) qPCR analysis of AT1 (*AGER* and *CAV1*) and AT2 markers (*SFTPA1* and *SFTPC*) in ALVOs cultured in indicated conditions as outlined in Fig. 3A with whole lung tissue as reference. Matched one-way ANOVA with Geisser–Greenhouse correction followed by Dunnett's multiple comparisons versus control (ALVO-EM day 10) was performed on log-transformed raw values. Data were shown as normalized mean ± SD (normalized to ALVO-EM day 10). Exact adjusted *p* values are indicated. (B) qPCR analysis of YAP target genes in ALVOs cultured in the indicated conditions as outlined in Fig. 3A. Matched one-way ANOVA with Geisser–Greenhouse correction, followed by Dunnett's multiple comparisons versus control, was performed on log-transformed raw values. Data were shown as normalized mean ± SD. Exact adjusted *p* values are indicated. (C) qPCR time course of AT1 (*AGER* and *CAV1*) and AT2 markers (*SFTPC*) in the three indicated ALVO lines after switching to BM + LATS-i media following a 10-day expansion phase in ALVO-EM. (D) IF images of ALVOs cultured in the indicated conditions as outlined in Fig. 3A. DAPI = nuclei; SFTPC = AT2 marker; AGER and GPRC5A = AT1 marker. (E) Electron microscopy image of ALVOs cultured in ALVO-EM or BM + LATS-i as outlined in Fig. 3A. N nucleus, V vacuole, LB lamellar bodies. (F) Cell growth time course (cell titer glo) in four ALVO lines grown in ALVO-EM or AT1/2-M. Data were normalized to day 0. (G) Side-view brightfield (H + E staining) and IF images of cells in 2D ALI transwells cultured in the indicated media conditions. DAPI = nuclei; SFTPC = AT2 marker; AGER = AT1 marker.

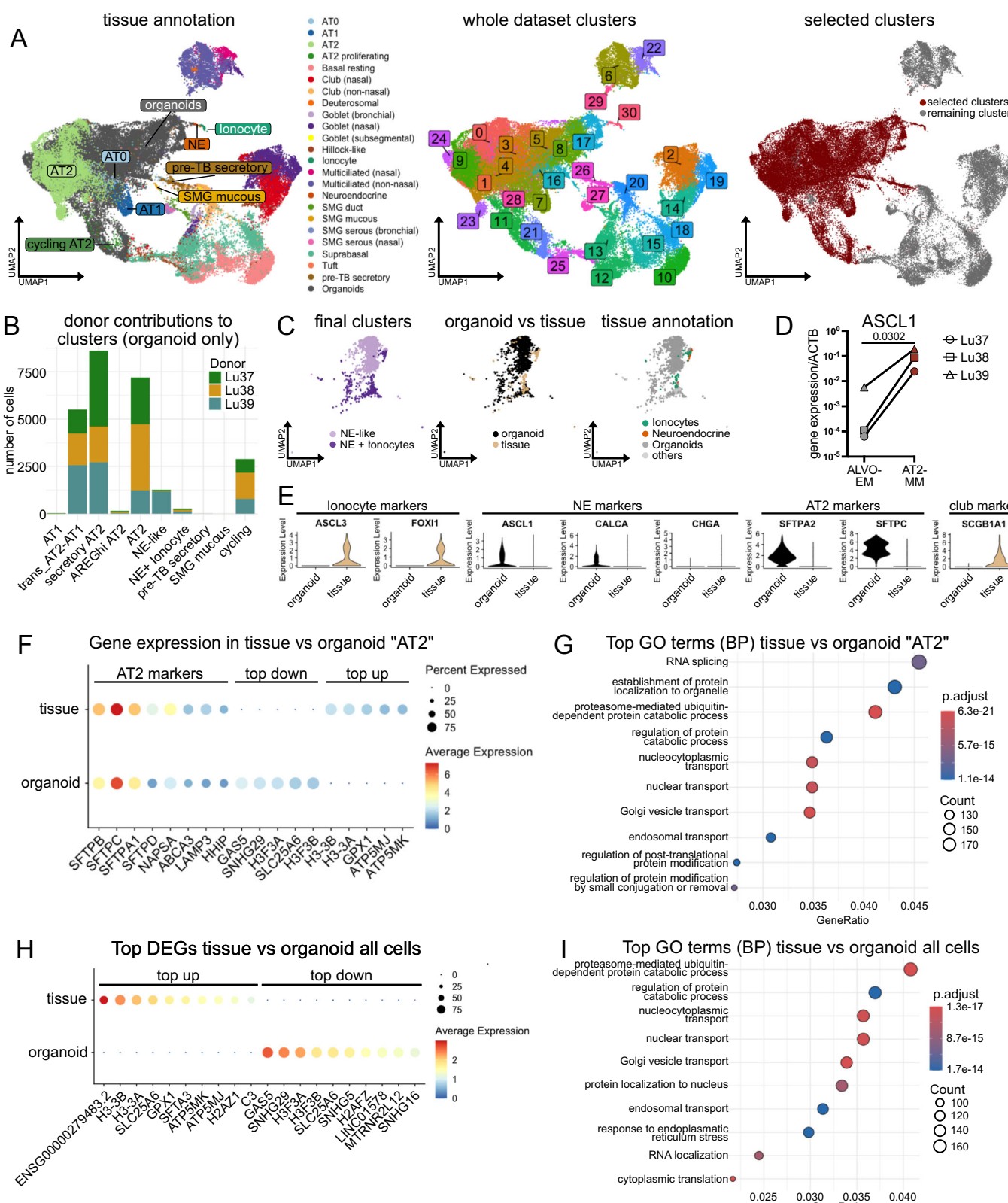

**Figure EV4. ScRNA-Seq comparison to lung tissue highlights regenerative phenotype of ALVOs. Related to Fig. 4.**

(A) UMAPs showing integrated organoid/tissue data, colored by tissue annotations and organoids (left), seurat clusters (center), and selected clusters for subsetting (right). (B) Bar plot showing the contributions of each organoid line (donor) to the organoid fraction of each final cluster in the organoid/tissue subset. (C) UMAP subset of the "NE-like" and "NE + Ionocyte" clusters of the organoid/tissue subset, colored by final cluster (left), organoid vs tissue origin (center), and tissue annotation (right). (D) qPCR analysis of NE-fate transcription factor ASCL1 in ALVO-EM and AT1/2-M. A two-sided paired *t*-test was performed on log-transformed raw values. Exact *p* value is indicated. (E) Violin plots showing gene expression levels of selected genes in "NE-like" and "NE + Ionocyte" clusters. (F) Dotplot showing AT2 markers and top down- and up-regulated genes comparing organoid and tissue origin within the "AT2" cluster of the organoid/tissue subset. (G) Top ten GO terms (biological processes) of differentially expressed genes between tissue and organoid cells from the "AT2" cluster. (H) Dotplot showing top up- and down-regulated genes comparing organoid and tissue origin within the whole organoid/tissue subset. (I) Top ten GO terms (biological processes) of differentially expressed genes between tissue and organoid cells from the whole organoid/tissue subset.

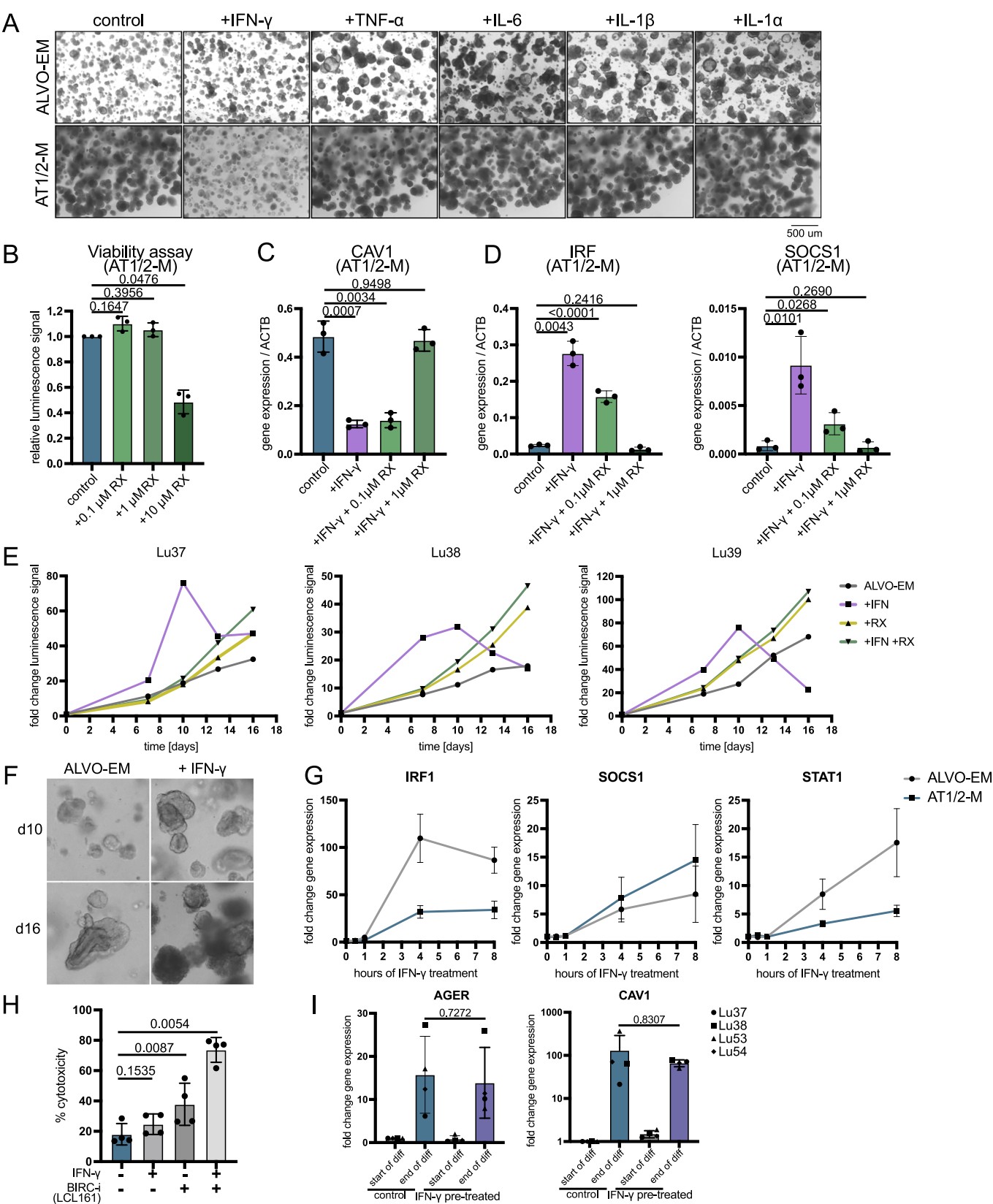

◄ **Figure EV5. IFN-γ selectively promotes alveolar progenitor growth while impairing AT1-like cells. Related to Fig. 5.**

(A) Brightfield images of ALVOs grown in indicated media conditions in the absence (control) or presence of indicated cytokines for 14 days. (B) Viability assay of ALVOs cultured in AT1/2-M in the absence (control) and presence of indicated concentrations of RX for 14 days. Matched one-way ANOVA with Geisser–Greenhouse correction, followed by Dunnett's multiple comparisons versus control, was performed on log-transformed raw values. Data were shown as normalized mean ± SD. Exact adjusted *p* values are indicated. (C) qPCR analysis of *CAV1* in ALVOs cultured in AT1/2-M in the absence (control) or presence of IFN-γ (10 ng/ml) and RX (0.1 and 1 uM) for 14 days. Matched one-way ANOVA with Geisser–Greenhouse correction, followed by Dunnett's multiple comparisons versus control, was performed on log-transformed raw values. Data were shown as mean ± SD. Exact adjusted *p* values are indicated. (D) qPCR analysis of IFN-γ target genes in ALVOs cultured in AT1/2-M in the absence (control) or presence of IFN-γ (10 ng/ml) and RX (0.1 and 1 uM) for 14 days. Matched one-way ANOVA with Geisser–Greenhouse correction, followed by Dunnett's multiple comparisons versus control, was performed on log-transformed raw values. Data were shown as mean ± SD. Exact adjusted *p* values are indicated. (E) Viability assay time course of three ALVO lines cultured in ALVO-EM in the absence and presence of IFN-γ (10 ng/ml) and RX (1 uM). (F) Brightfield images of ALVOs cultured in ALVO-EM in the absence and presence of IFN-γ (10 ng/ml) at the indicated time points. (G) qPCR time course of IFN-γ target genes in ALVOs cultured in indicated media conditions in the presence of IFN-γ (10 ng/ml) normalized to their respective untreated controls. (H) Cytotoxicity assay of organoids in AT1/2-M with and without IFN-γ and the BIRC3 inhibitor LCL161 for 3 days. Matched one-way ANOVA with Geisser–Greenhouse correction, followed by Dunnett's multiple comparisons versus control, was performed on log-transformed raw values. Data were shown as normalized mean ± SD. Exact adjusted *p* values are indicated. (I) qPCR analysis of ALVOs that were not treated (control) or pretreated with IFN-γ from days 0–7, and differentiated in AT1/2-M subsequently from days 10–17. Two-sided paired *t*-tests were performed on log-transformed raw values between the indicated groups. Data were shown as normalized mean ± SD. Exact *p* values are indicated.

