## [Peer Review File · The EMBO Journal]

Interferon γ selectively promotes survival of alveolar progenitor cells in a human lung organoid model

Antonella Dost, Katarína Balážová, Carla Pou Casellas, Lisanne van Rooijen, Wisse Epskamp, Gijs van Son, Willine van de Wetering, Carmen Lopez-Iglesias, Harry Begthel, Peter Peters, Niels Smakman, Johan van Es, and Hans Clevers

Corresponding authors: Hans Clevers (h.clevers@hubrecht.eu) , Antonella Dost (a.dost@hubrecht.eu)

Review Timeline:

Submission Date:	2nd Dec 25
Editorial Decision:	4th Dec 25
Revision Received:	21st Jan 26
Editorial Decision:	24th Feb 26
Revision Received:	11th Mar 26
Accepted:	26th Mar 26

Editor: Daniel Klimmeck

Transaction Report:

This manuscript was previously reviewed at another journal. As EMBO Press has a transfer agreement (including the identities of the referees) with that journal, revision was invited based on the reports from that previous external submission.

**Reviewer #1 (Remarks to the Author):**

This manuscript described a modified protocol of generating and maintaining alveolar organoids from
isolated epithelial cells derived from primary human lobectomy material. The authors use this model (that
was improved with regard to a differentiation step (LATS-i) and with regard to prior expansion in ALVO-EM)
to demonstrate a role of recombinant IFN γ on organoid cell population behaviour, and extrapolate a role of
IFN γ in human COPD. While some aspects of the presented data are interesting, I have concerns regarding
data interpretation and conclusions drawn.

The authors use lobectomy material derived from lung cancer surgeries. No information is provided as to
patient history regarding age, gender, type of lung cancer, smoking history, COPD stage, or steroid
medication history, which might impact the composition and cellular responses of the organoids, particularly
in light of the findings on IFN γ and a putative role in COPD.

Given that the authors identify quite some heterogeneity in the different cultures (e.g., Fig 1C, S1H), and that
such criteria might significantly affect alveolar stem cell behaviour, such information should be provided to
better interpret the results.

On a minor note, might the inflammatory tumor microenvironment/signature within the lobectomy material
additionally impact the cells of origin used for organoids (how distant from tumors was the tissue used for
preps)?

For ethical and practical reasons, we work with surgical "rest material" that is fully anonymized, so patient-
specific data (age, sex, smoking history, etc.) are unavailable. However, we do know that none of the patients
were diagnosed with COPD or other lung diseases beyond lung cancer, for which the surgery was performed.
Furthermore, they did not receive any form of cancer-therapy before the surgery. We will add this
information to the methods section.

Regarding heterogeneity: variation between human donors is expected and intrinsic to primary organoid
systems. It is not possible to get tissue from a "normal, healthy" lung. Material either comes from lungs that
were rejected for transplantation, or from lobectomies, like in our case. We consistently use at least three
biological replicates to ensure reproducibility of observed responses. We also choose to show donor
identifiers for full transparency. The distal lung tissue that we receive is at least 5cm from the tumor and we
will add this information to the methods section.

The data on establishment of the new protocol seem to lack consistency, and clarity regarding information
on experimental setup, which makes interpretation difficult, and it remains unclear what message the
authors want to convey about their new protocol.

We tried to be as clear and transparent with the protocol as possible. We will make some clarifications
regarding which media was used to establish the organoid lines, and add information regarding passage
numbers, as suggested by other reviewers. Current approaches depend on HTII-280, a widely used AT2-
specific membrane antigen whose corresponding gene remains unknown. It is known that HTII-280-negative
cells can generate SFTPC+ organoids (Basil, Nature, 2022; figure 3) and that not all AT2 cells stain positive for
HTII-280 (PMID: 32272001). This suggests that progenitors capable of forming AT2-like cells are excluded
when HTII-280-based sorting is used. We therefore see a conceptual advantage in our more inclusive
approach. Moreover, we achieve long-term expansion of AT2 progenitor cells AND secretory AT2 progenitor
cells, which has not been demonstrated before. Another novel aspect of our system is that the sole variable
between our progenitor expansion media and our differentiation media is LATS-inhibitor, making precise
mechanistic studies possible. We have extensive characterization of our organoids, including scRNA-Seq,

which is not the case for existing protocols (Katsura, CSC, 2020, Salahudeen, CSC, 2020). Together, we think
that our system presents an advancement to currently available protocols.

For example, Fig 1C right panel lacks Lu44 that was referred to before to be the most long-lasting line (21
passages). Fig 1C and 1F again show different donors. Figure 1H shows data from donor 37, 53 and 54, and
Figure 1I only from 53 and 54.

Due to practical reasons, it is not feasible to use the exact same three donors for every experiment. We view
the diversity of donors as a strength rather than a weakness, and we are fully transparent which lines were
used for which experiment. We did not continue using Lu44 because it represented an outlier when it came
to long-term expansion, and we chose to continue with more representative lines.

The scRNASeq experiments (from donors 37-39) lack information about passage numbers and how long
these organoids have been in culture. Can the authors explain why there is such low amount of proliferating
cells, within the cells with progenitor phenotype that should reveal higher proliferation rates compared to
AT2 cells (“secAT2”, C6+; “RAS-like”)? The scRNA-Seq was done at passage 5 with each passage being 14 days
apart; we will add this information to the manuscript. We would like to emphasize that our goal is not to
model the alveolus at homeostasis, where epithelial cells are quiescent, but rather a regenerative state. This
is why organoid-derived cells form the majority of proliferating cells when integrated with lung atlas data
(Figure 4A). Under regenerative conditions, we do not expect many mature, and therefore quiescent, AT2
cells in expansion media. As Reviewer 2 pointed out, the field increasingly recognizes that alveolar
progenitors are heterogeneous. Our organoid conditions promote stemness and proliferation, enriching two
AT2-like progenitor subpopulations, one of which resembles the secretory alveolar phenotype described in
recent Nature papers (Murthy, Nature, 2022 and Basil, Nature, 2022). These “secAT2” cells share features
with AT0, TRB-SC, and AT2-3A2 populations, cells thought to lie along a continuum between secretory and
alveolar lineages.

In Figure 2D, there are organoids that are HT2-280 negative after AT2-MM culture, but Nile Red positive.
What are these organoids, and can the authors comment on the heterogeneity of their cultures? To answer
this question, we will do FACS sorting on Nile Red high / low populations in AT2-MM with subsequent gene
expression analysis. Regarding HT2-280, it has been noted that “HT2-280 is a less useful marker for
subculture of AT2 cells due to loss of expression during culture.” (PMID: 32272001). This is why we show
maturation with functional dyes such as LysoTracker and Nile Red instead.

It is difficult to assess the ultimate advantage of the presented over previously published models (e.g., PMID:
33128895), also in light of the results of Fig 3F and 3I indicating separation between C6+ and C6- cells seems
dispensable. – Please refer to page lines 37+ for the advantage of our system to the published protocols.
Moreover, figure 3F and 3I show that both AT2 and secAT2 populations can give rise to AT1-like cells, but
that does not make their separation dispensable.

Also, it remains open how well the ALVOs are suited to model the alveolus containing AT1 cells. Although
AT1 markers are increasingly expressed (particularly in AT1/AT2-M, no matter if used directly or after ALVO-
EM expansion), there is little contribution from organoids compared to tissue to the AT1 pool when looking
at identity contribution (Fig 4E). Given that only very few AT1 cells are contained in the organoids, the
abstract statement that “Unlike previous models, this system supports long-term expansion of newly
identified human-specific alveolar progenitor cells and serum-free differentiation into alveolar type 1 (AT1)

cells.” seems not to be supported by data. – The cells in our organoids have high expression of AT1 markers
such as AGER on protein level, which is generally accepted by the lung field to indicate AT1 identity. We have
acknowledged and discussed the low contribution of fully differentiated AT1 cells in the scRNA-Seq data in
the discussion as follows: *“However, the majority of AT1-like cells have a gene expression profile resembling
the AT2-to-AT1 transitional state rather than that of fully differentiated AT1 cells. It is possible that organoid-
derived differentiated AT1 cells are underrepresented in the scRNA-Seq dataset due to technical reasons. The
typical shape of AT1 cells makes this cell type especially sensitive to dissociation methods necessary to
achieve single-cell suspensions. A biological explanation for the underrepresentation is that additional signals
or prolonged culture periods might be required for terminal AT1 differentiation. Nevertheless, we posit that
the presence of transitional states in our organoids better reflects regenerative conditions and provides a
valuable platform for optimizing AT1 differentiation in therapeutic contexts.”*

The culture type-specific effects of IFN γ are an interesting finding (ALVO-EM vs AT1/2 medium). The
conclusion that in particular, AT1-like cells are “negatively” affected would benefit from a more detailed
analysis (i.e. type/pathways of cell death induced by the cytokine, etc). Are the IFN γ effects observed on
“AT1-like” cells, or trans_AT2-AT1 cells (Fig 4), similarly observed in truly differentiated AT1 cells? –We
clearly demonstrate a reduction in cell numbers (5A), an increase in cytotoxicity (5B), and an upregulation of
genes associated with cell death (5H). We also demonstrate reduced AT1 marker expression (5C), indicating a
selective loss of AT1 marker-expressing cells. To further address this question, we will perform a more
detailed analysis of the cells that survive after IFN-gamma exposure, including stainings of the organoids. The
focus of this publication is not to achieve terminal AT1 differentiation in vivo but to model alveolar
regeneration.

In light of the different actions of IFN γ on secAT2/AT2 vs “AT1-like” cells, what is the expected “net outcome”
of IFN γ signaling in COPD according to the authors data, and how can this be interpreted with regard to
COPD (emphysema) lung phenotype, given that such lungs do not contain more or more proliferating
AT2/progenitor cells? – There is in fact an accumulation of secAT2s in COPD and smoker lungs (Basil et al,
Nature), and we do not know the reason for this yet. However, the inflammatory landscape of COPD lungs is
too complex to make a prediction regarding a net outcome of the effect of IFN-gamma.

How much “COPD phenotype” is imprinted in the organoids (given that likely, at least some of the donors
were smokers/COPD patients, see above)? – As mentioned above, none of the donors had a lung disease
diagnosis apart from lung cancer. The question of an imprinted smoking or COPD phenotype – even after
33 months in organoid culture conditions – is very interesting but beyond the scope of this manuscript.

On a minor note, why do the authors use PBMC-derived macrophage-like cells for co-cultures, given that T
cells are major sources of IFN γ ? – This is for practical reasons: T cells need to be patient matched, which is
not the case for macrophages. While T cells are a major source for IFN- γ , macrophages also secrete the
cytokine.

All figure legends should contain information on n numbers, number of independent experiments, p values
and statistics applied. - We will add this information to the figure legends.

Other:

• Fig 1A: which protocol (“published methods”) do the authors exactly refer to? – The protocols
corresponding to ref 20 Katsura et al., 2020, and 21 Youk et al., 2020. We will add this information to the

figure.

• P4: what do the authors mean by “appeared healthier”? (line 114). – the organoids appear bigger and with
less shedding cells. We will clarify this in the manuscript.

• How were the stats calculated in Fig 3I and 5G (for datapoints normalized to 1)? – this is described in the
methods: “For comparisons between two groups, an unpaired t-test was used, or a paired t-test if the data
were normalized to the control group. For comparisons involving multiple groups against a control group, or
among multiple groups, a two-way ANOVA followed by Dunnett’s post-hoc test was employed. When data
were normalized to the control group, a matched two-way ANOVA with Dunnett’s post-hoc test was
performed on log-transformed values to account for baseline differences”

• Fig S5F does not show cell death (line 348) – These images show excessive shedding of cells and cell debris
from the organoids, which is an indication of dying cells. We will rephrase this in the manuscript.

**Reviewer #2 (Remarks to the Author):**

In this manuscript entitled “Interferon- γ selectively promotes survival of alveolar progenitor 1 cells in a
human 2 organoid model” the authors describe the development of a novel approach to adult human
alveolar epithelial culture modelling with particular emphasis on an inclusive approach to cell isolation to
ensure the capture of all potential alveolar progenitors. They then define media components required for
AT2 maturation and AT2-AT1 differentiation from their mixed population. Having undertaken detailed
characterisation, they then use this model to identify the differing role of interferon- γ on the proliferation
and survival of different alveolar epithelial population, finding that interferon- γ is cytotoxic to AT1 cells but
promotes survival of AT2 cells.

The manuscript is well written and easy to follow. The figures are carefully prepared and they present their
data clearly with appropriate analysis. Their transparency in presenting negative and variable data is
appreciated.

Their novel organoid model is potentially of great interest to the field. By taking a more unbiased approach
to sorting lung epithelial cells from fresh adult lung tissue than current protocols, they find different AT2
subtypes (AT2, secretory AT2) and then push them toward different fates (AT2-like or AT1-like) by
manipulation of media components. As we increasingly recognise that alveolar progenitors are not simply
one entity, the ability to generate a manipulable model containing several AT2- and AT1-like subtypes is of
real significance both in disease modelling and understanding AT2 transdifferentiation in health and disease.

To convince the reader of their conclusions about the true originality of the model and the ability to use it for
the disease modelling described in the second part of the manuscript, I would suggest the following:

Major comments

1. The authors state in the introduction that using HTII-280 sorting strategies may exclude important alveolar
progenitors; indeed, they then use this argument for creating a novel culture strategy which does not rely on
this poorly characterised transmembrane protein. Though I agree with their concerns about HTII-280, the
absence of any data to support their statement is a concern. I would suggest:

• They should provide evidence that HTII-280 is absent from non-standard AT2 cells (e.g. by citing existing
scRNAseq datasets) to support the logic of their experimental approach.
Because the corresponding gene to HT2-280 is not known, it cannot be looked up in scRNA-Seq data.
However, it is known that HTII-280-negative cells can generate AT2 organoids (Basil, Nature, 2022,
figure 3). This suggests that progenitors capable of forming AT2 cells are excluded when HTII-280-
based sorting is used. Furthermore, it has been observed that “(...) although the majority of AT2 cells
express HTII-280, HTII-280- AT2 cells are observed even in healthy human lungs” (PMID: 32272001).
• They should show whether HTII-280-based AT2 isolation generates a similarly heterogeneous AT2-like
population, and whether the same manipulations of the culture media (i.e. use of AT2-MM / AT1/2-M) allow
maturation of AT2 and AT1-like cells from HTII-280-isolated organoids. This will likely highlight the key
advantages of this new model.
- To satisfy multiple reviewer requests, we will sort p0 organoids at day 14 for EPCAM+/NGFR-/HT2-280+ and
EPCAM+/NGFR-/HT2-280- populations and plate them in media containing either HRG1 or EGF. This
experiment will tell us whether the observed alveolar progenitor heterogeneity comes from the sorting
strategy and therefore from the starting cell populations, or whether it comes from the media conditions.
2. Figures 5 and 6 describe the differential response of AT2-like and AT1-like cells to interferon- γ . By
necessity, the culture media for each cell type is different with AT1-like cells maintained in LATS-i. Bearing in
mind the crosstalk between LATS and IFN pathways, how are the authors sure that the observed effects on
cell proliferation and survival due to cell type and not the effect of LATS-i? – We have experiments ongoing
to address this question. We will differentiate the organoids first, withdraw LATS-i, then add IFN to see if you
still see increased cytotoxicity.
The following would be reassuring:
• To demonstrate that LATS-i does itself not affect cell proliferation and survival.
- LATS-i in fact increases the proliferation rate of our organoids, data we can add to the manuscript.
Additionally, in figure 5F we compare the IFN-treated cultures to their own controls and therefore normalize
for difference in cell growth between the ALVO-EM and AT1/2-M conditions.
• To check that the combination of LATS-i and interferon- γ does not affect the phenotype of the AT1-like
cells grown in AT1/2-M.
- We will add better characterization (RNA and stainings) of the cells that survive long-term IFN-g exposure in
the AT1/2-condition apart from the RNA analysis we did (figure 5C). However, our cytotoxicity assays are
conducted in already grown and differentiated cultures that were exposed to IFN-g for only 3 days (5B) and
we do not expect drastic changes in cell identity in this time frame.
Other comments
1. The introduction opens with a brief discussion of COPD and the need for better models of the effect of
inflammatory insults in the lung. While I agree, this really isn't the focus of the manuscript which has far
broader reach in modelling the alveolar epithelium in health and disease. Indeed, I would suggest the models
they describe would be more immediately relevant to modelling diseases of the alveolus e.g. ILD. To this end,
I would suggest the introduction (and discussion) are reworded to downplay the emphasis on COPD.
- We agree with this comment and will rephrase the introduction.

2. Fig S1E&F: AGER is used in S1E then CAV1 in S1F as AT1 markers. Subsequently both are used. For
transparency and continuity, I would suggest including both in Fig S1.
- We will add this data.

3. Fig 2B/Fig S2A&B: There is an assumption here that higher lysotracker = more AT2-like. It would be nice to
have a) freshly isolated AT2 cells (e.g by HTII-280 sorting) as a control to define physiological levels of
lysotracker, and b) a control line which does not have LBs as a readout for physiological lysosome-resident
lysotracker staining.
- We will try to add this data pending availability of fresh patient material. However, the correlation between
lysotracker staining and AT2 maturity has been described in the literature ([https://doi.org/10.1186/1465-
9921-14-123](https://doi.org/10.1186/1465-9921-14-123)).

4. Fig 2C: The EM of secreted surfactant is beautiful – thank you for including this. The contrast and zoom in
the middle panel could be improved to make the LBs more convincing, however.
- We are also very excited about these images! We will improve the quality as suggested.

5. Line 196: It has been shown that LATS manipulation alone can drive an AT1 phenotype in organoids
derived from precursors (e.g. Burgess et al PMID 38642558; Lim et al PMID 39815007). This should be
mentioned.
- Both publications are already cited, but we will additionally cite them at relevant parts in the manuscript.
However, in both publications, LATS-i is used in combination with Wnt withdrawal, and not by itself. We do
show in our own manuscript that Wnt withdrawal can cause differentiation, even without LATS-I addition
(Figure S3A, base media “BM”).

6. In Fig 5, ruxolitinib is used at concentrations of 100nM-1uM then at 1uM in Fig 6. Can the authors justify
this choice? The IC50 appears to vary widely according to cell type, but these seem to be high
concentrations. Might there be off-target effects?
- We tested the drug at 0.1uM, 1uM, and 10uM (figure S5B) and saw no cytotoxicity at 0.1uM and 1uM. We
therefore tested 0.1uM and 1uM in our rescue studies (figure 5D, 5E, S5, S5C, S5D) and saw a dose-
dependent effect, with a partial rescue of viability, AT1 gene expression, and suppression of IFN target genes
at 0.1uM and a full rescue at 1uM. Because we wanted a full rescue, we chose 1uM in our macrophage
studies in figure 6. Ruxolitinib is used widely, and the dose-dependent repression of IFN target genes
indicates that the drug is on-target.

7. In Fig 5A&F, different symbols for each donor would help understanding of the individual responses to
interferon- γ .
- We will add this to the figure.

8. In Fig 5C, SFTPA is used as an AT2 marker rather than SFTPC (which is used everywhere else). Why is this?
- We will add the data for SFTPC as well.

9. Figure 5J demonstrates a cytotoxic effect of BIRC-i in interferon- γ cells grown in ALVO-EM. This experiment
should also be undertaken in in AT1/2-M-cultured cells, despite there being no upregulation of BIRC3 in this
population.

- We will do this experiment and add it to the manuscript.

10. In figure 6, activated macrophages are used as a source of interferon- γ , but there are no data to confirm
that this is the case. Data should be included to confirm interferon- γ production, and ideally that (as stated in
line 394-395) that a consistent supply of interferon- γ is indeed being provided when cells are replaced every
3-4 days.

- There is a lot of literature showing that LPS and IFN-g stimulation of macrophages leads to a pro-
inflammatory (“M1-like”) phenotype and IFN-g secretion (doi: 10.3389/fimmu.2019.01084, PMID: 29921905,
<https://doi.org/10.1093/intimm/5.11.1383>, PMID: 24680476). Furthermore, the fact that we get a rescue of
the phenotype (figure 6E) with the JAK/STAT inhibitor ruxolitinib indicates that it is indeed IFN-g causing the
phenotype.

11. In figure 3G, the authors show a significant difference in AT-1 like phenotypes emerging dependent on
the ALI transwell culture coating. This is not explored further in this manuscript, but is important. I would
suggest this is expanded on in the discussion.

- We will expand on this in the discussion

Minor points

Fig S1A: The SFTPC signal is very hard to see in row 3, column 2 (I assume this is supposed to represent
positive staining) – we will improve the image

Fig 2C legend: Spelling (DIC) – we will correct this

Fig 4E (legend on right), Fig S4A,B,F,H: The text is too small to read – we will improve this

Line 102: A reference, and ideally a brief justification, is required to justify negative sorting of NGFR – NGFR is
a well-established surface molecule used to sort lung basal cells, the stem cells of the airway epithelium. We
will add a reference for this (<https://doi.org/10.1073/pnas.0906850106> and
<https://doi.org/10.1165/rcmb.2020-0464OC>)

Line 563: I assume this should read 10 million cells/ml – indeed, we will change this

Line 710: Spelling (Quiagen) – we will correct this

Reviewer #2 (Remarks on Protocol(s)):

Very well written and comprehensive

Reviewer #3 (Remarks to the Author):

In this manuscript, Dost et al. present a new set of protocols for deriving human distal lung organoids with
varying differentiation statuses. Using these culture systems, the authors examine how IFN- γ treatment
differentially affects organoids depending on the epithelial cell types present. While the study provides
improved methods for expanding organoids from primary human lung epithelial cells and includes detailed
characterization, several critical concerns remain unresolved, as outlined below.

Major comments:

1. Overall, although the sorting-free approach for the initial culture of lung single-cell suspensions and the
inclusion of potential distal progenitor cells capable of generating alveolar lineages may enhance cell survival
and expansion, the manuscript does not fully substantiate its conclusions, partly due to the heterogeneity of
the starting cell population that gives rise to organoids. Specifically,
a. The authors should contextualize better the rationale for using NGFR to enrich “progenitor cells” for
alveolar organoid generation. What does NGFR mark within P0 organoids? The authors should demonstrate
what populations exist in EpCAM+NGFR- cells. Flow cytometry for EpCAM, NGFR and HTII-280 require to
show cellular profiling. scRNA-seq of EpCAM+NGFR- vs EpCAM+HTII-280+ cell comparison is required to
determine the initial populations in their P0 organoids. Interpretation is further hindered by the lack of
scRNA-seq or other profiling data at the critical timepoint between P0 and the subsequent EpCAM+NGFR-
sorting, when cellular heterogeneity would be manifested. The authors should include such data to clarify
which cell types are present and potentially contribute to the observed progenitor diversity.
NGFR is a well-established surface molecule used to sort lung basal cells, the stem cells of the airway
epithelium (<https://doi.org/10.1073/pnas.0906850106> and <https://doi.org/10.1165/rcmb.2020-0464OC>).
Our rational is to exclude basal cells from our alveolar organoid cultures. To satisfy multiple reviewer
requests, we will sort p0 organoids at day 14 for EPCAM+/NGFR-/HT2-280+ and EPCAM+/NGFR-/HT2-280-
populations and plate them in media containing either HRG1 or EGF. This experiment will tell us whether the
observed alveolar progenitor heterogeneity comes from the sorting strategy and therefore from the starting
cell populations, or whether it comes from the media conditions.
b. What does CEACAM6 specifically mark in this culture? Although the authors state that CEACAM6 is a
marker for “RAS” cells based on Basil et al. (2022), CEACAM6 expression is known to span a broad range of
distal epithelial populations, including AT0, TRB-SC/RAS, club, goblet, and AT1 cells. Figure 1D also showed
heterogeneity of CEACAM6-expressing cells in organoids.
- As shown in figure 1D and 1G, CEACAM6 expression correlates with reduced, albeit still high, expression of
SFTPC and increased expression of the secretory markers SCGB3A1 and SCGB3A2. Figure 1E shows varying
degrees of co-expression of CEACAM6 and SFTPC on protein level. The club and RAS cell marker SCGB1A1 is
not expressed. Moreover, our CEACAM6+ cells in ALVO-EM do not express higher levels of AT1 markers,
which is the case for AT0 cells. The expression pattern of the organoid secAT2s most closely aligns with the
expression profile of TRB-SCs (Murthy et al, Nature, 2022) or AT2/3A2 cells (Basil et al, Nature, 2022).
The mentioned populations (AT0, TRB-SC, RAS, AT2/3A2) are very recent cell type/cell state discoveries that
are not well described or understood yet. Because the populations were described by two separate labs
independently, there are different nomenclatures that likely describe overlapping populations. It is likely that
these cell types lie on a spectrum from secretory to alveolar cells, with varying levels of secretory and
alveolar gene co-expression. We do not claim that our organoids contain one specific cell population/state
that corresponds to the populations described in vivo. Instead, we chose the term secAT2 cells, because they
co-express secretory and alveolar genes, one of the common themes of the newly described populations.
c. What are secAT2 cells? Based on the presented data, these cells appear to resemble AT0-like population,
reported by Murthy et al. (2022). However, Murthy et al. demonstrated that AT0 cells exhibit markedly
reduced SFTPC expression compared to AT2 cells, whereas the so-called secAT2 cells in this study show
SFTPC and LAMP3 levels comparable to AT2 cells (Figure 1G, J,). Does this imply that secAT2 cells are an in
vitro artifact, a culture-specific intermediate state, or a transitional population bridging AT2 and AT0 states?
Critically, Murthy et al. further demonstrated that AT2 cells transition into AT0 cells and subsequently into
TRB-SC (SFTPB+SCGB3A2+SFTPC-) cells upon EGF withdrawal. In contrast, in the current study (Figure S1E),

EGF withdrawal appears to increase both SFTPC and SCGB3A2 expression. The authors also claim that ALVO-
EM culture promotes CEACAM6 and SCGB3A2 expression in “AT2” cells (CEACAM6-neg AT2) – is this effect
simply attributable to the absence of EGF in ALVO-EM? Moreover, Murthy et al. reported unidirectional
differentiation of AT2 toward AT1 or TRB-SC lineages through AT0 states. Given this, the rationale for
including secAT2 cells to enrich AT2 cells or to improve alveolar organoid formation requires clearer
justification. Are secAT2 and AT2 cells interconvertible under this culture condition? To substantiate these
findings, it would be important to determine whether HTII-280+AT2 cells (EpCAM+NGFR-HTII-280+ vs
EpCAM+NGFR-HTII-280-) can give rise to secAT2-like or AT0-like cells – or vice versa – when cultured with
EGF or HRG1.

- As mentioned in the above comment, we do not claim that our organoids contain one specific cell
population/state that corresponds to the populations described in vivo. Our rationale for including secAT2 cell
is that such a cell type/state likely contributes to alveolar regeneration. In mice, club cells and
bronchioalveolar stem cells (BASCs, co-expressing the secretory club cell marker SCGB1A1 and the AT2
marker SFTPC) contribute to alveolar repair. In humans, the respiratory bronchioles contain multiple
populations that likewise have secretory-alveolar gene co-expression (AT0, TRB-SC, RAS, AT2/3A2). If and
how these cells contribute to alveolar regeneration in humans is not known yet, but it has been
demonstrated that iPSC-derived RAS cells can give rise to AT2 cells (Basil, 2022, Nature and doi:
10.1038/s41587-025-02569-0.). This data suggests a trajectory from secretory to alveolar cells.
While it has been shown that withdrawal of EGF can cause transdifferentiation of AT2s to TRB-SC through an
AT0 state, our organoid cultures grow without EGF long-term and still contain an AT2 population, indicating
that we have two stable progenitor populations in our organoid cultures.
To address the effect of HRG1/EGF in our cultures, we will sort for EpCAM+NGFR-HTII-280+ and
EpCAM+NGFR-HTII-280- as suggested, and grow these two populations in media containing either EGF or
HRG1 subsequently.

2. Although substituting EGF with the alternative EGFR family ligand HRG1 in alveolar organoid cultures is an
interesting modification, it appears to only modestly improve organoid propagation. Previous study using
SFFF medium containing EGF (Katsura et al. 2020) already demonstrated long-term subculture beyond 10
passages. In the current study, while the authors report somewhat improved expansion, SFTPC and SFTPA1
expression levels decline after passage 10 (Figure S1G), restricting organoid use to the first 10 passages.
Therefore, the practical benefit of this protocol appears limited. Furthermore, a similar sorting-free approach
was previously employed by Salahudeen et al. (2020) to generate human distal lung organoids from primary
lung tissues. They used EpCAM+Lysotracker+ cells from PO organoids, which likely contain AT2 and secAT2
(or AT0 cells) based on the current study. So, it raises an important question – would this strategy provide a
more effective means to enrich AT2 and/or potential progenitor-like populations than the current use of
EpCAM+NGFR- cells, given the non-specificity of NGFR and the resulting cellular heterogeneity?

- One important practical benefit is that the FACS/MACS sorting is not done on the day of the tissue
processing. In practicality, receiving tissue from the hospital, processing it to single cell suspensions, then
sorting and plating the cells is a long and harsh process that reduces cell viability. There are other practical
considerations rarely talked about in scientific publications, like the availability of a FACS machine on the day
the tissue is received, which can be unpredictable. For further explanations of advantages of the system,
please refer to page 1 line 37+.

Regarding the EpCAM+Lysotracker+ sorting strategy from Salahudeen et al. (2020): Lysotracker signal
corresponds to the maturity level of AT2 cells. However, it has been shown that AT2 “stem/signaling” cells
(PMID: 33208946), and also the newly described progenitor populations residing in the respiratory

bronchioles (AT0, TRB-SC, RAS) have lower levels of canonical AT2 markers. It makes sense that proliferating
AT2 progenitor cells would not have the same level of maturity as quiescent AT2 cells that produce high
levels of surfactant. AT2 progenitor cells would therefore likely have lower levels of lamellar bodies and
lysotracker staining, which is why we think that lysotracker sorting is a less inclusive approach than our
sorting strategy. We see the heterogeneity of our ALVO-EM cultures as an advantage because it gives us the
opportunity to start studying the secretory-alveolar populations in vitro.

3. Regarding the use of HRG1, the signaling aspects related to this alteration require further exploration.
Since HRG1 does not bind to EGFR but rather ERBB3 and ERBB4, what are the levels of expression of each of
these receptors in secAT2 and AT2 cells during organoid culture? Ideally, this should be assessed at the
timepoint following the unsorted culture of lung single cell suspensions at day 14 prior to sorting for further
organoid culture. Are there differences in downstream signaling between these populations when cultured
with EGF vs. HRG1? Does HRG1 specifically drive the generation of secAT2 cells from AT2 cells, or is this
effect simply due to the absence of EGF, as noted above? Finally, what are the cellular sources of HRG1 in
lung tissues, and what is the expression pattern of its receptors within these tissues?

- We have data showing that ERBB3 and ERBB4 is expressed in AT2 cells in lung tissue, while HRG1 (NRG1) is
expressed by multiple cell types, including AT1 cells. We can also show that ERBB3 and ERBB4 are expressed
in our organoid cultures, both in the AT2 and secAT2 populations.

To satisfy multiple reviewer requests, we will sort p0 organoids at day 14 for EPCAM+/NGFR-/HT2-280+ and
EPCAM+/NGFR-/HT2-280- populations and plate them in media containing either HRG1 or EGF. This
experiment will tell us whether the observed alveolar progenitor heterogeneity comes from the sorting
strategy and therefore from the starting cell populations, or whether it comes from the media conditions.

4. In Figure S1D and E, organoids cultured without HRG1 and EGF revealed poor organoid formation but
higher expression levels of SFTPC and AGER. The authors should provide an explanation for this observation.
– Removal of these factors decreases proliferation of the progenitor cells. In the absence of these pro-
proliferative and pro-stemness cues, the cells likely mature/differentiate to either AT2 or AT1 cells. However,
this effect is very small compared to our AT1 differentiation protocol where AGER is upregulated almost 100
29 fold (figure 3I).

5. Single cell RNA-seq data from three separate patients of organoids from initial single cell culture in ALVO-
EM (Figure 1D/S1H-I), and after switching to ALVO-EM, AT1/2-M, and AT2-MM (Figure 4C-E/S4A-C), clearly
show that one patient (Lu39) is responsible for the majority of 3 separate clusters in Fig. 1/S1, and
furthermore is responsible for the vast majority of neuroendocrine-like cells from the different culturing
conditions in Figure 4/S4 (see most pertinently Figure S4C). However, the authors do not refer to this clear
demarcation in the Results section related to each of these figures. The authors should update the Results
section to communicate to the readers that this is the case, probably attributable to heterogeneity in human
samples taken for organoid culture. Furthermore, the authors may want to soften their interpretation of
these results as representative of general principles of neuroendocrine phenotype emergence in DCI-
containing organoid cultures (see Discussion, line 425-430), given that 2/3 patient samples do not seem to
appreciably show this conversion. – We will add qPCR data showing that we also get neuroendocrine-gene
upregulation in the other organoid lines. While this is not the focus of the paper, we think that it is an
interesting observation to point out in the discussion, and this observation is not only based on our own data
but also data from the referenced publication (ref 53) and another recent publication that we will add to the
discussion (PMID: 41075787).

6. In Figure 2A, what was the initial cell population used for this experiment? Did the authors test whether
AT2 and secAT2 cells differ in their capacity to differentiate into mature AT2 cells? – *The initial cell
population are all cells present in our ALVO-EM conditions, so both AT2-like cells and secAT2 cells. To answer
this question, we will sort out these two populations and expose them to our AT2-MM separately.*
Given the heterogeneity of this culture, it is surprising that minimal differences were observed in response
to the screening factors. Furthermore, in Figure 2D, the majority of cells in the ALVO-EM condition do not
express HTII-280 despite of their high expression of AT2 markers shown in Figure 1. Additionally, in the AT2-
MM condition, one organoid contains Nile red+ lipids but is negative for HTII-280, whereas another organoid
shows lower lipid content but high HTII-280 expression. How do the authors explain this discrepancy? –*To
answer the question regarding the Nile red-high cells, we will do FACS and subsequent gene expression
analysis on the Nile red-high/low populations. Regarding HTII-280, it has been pointed out that this surface
molecule might be downregulated in in vitro cultures and might therefore not be a reliable marker:
“However, it is important to note that although the majority of AT2 cells express HTII-280, HTII-280– AT2
cells are observed even in healthy human lungs (Figure 2B). Furthermore, HTII-280 is a less useful marker for
subculture of AT2 cells due to loss of expression during culture.” (PMID: 32272001)*
Since DCI is known to promote alveolar maturation in both iPSC-derived and fetal lung organoids, a similar
effect is likely in primary alveolar organoids. Does HRG1 modulate this maturation process differently than
EGF? – *Both iPSC- and fetal-derived organoids are in a more developmental-like state compared to primary
adult cells, so it is not obvious that adult alveolar cells would respond the same. To answer the question
regarding HRG1/EGF, we will test the AT2-MM in the presence of either EGF or HRG1.*

7. LATS-i has previously been reported to induce AT1 differentiation in both iPSC- and primary lung tissue-
derived alveolar organoids (Burgess et al. 2024). In this current study (Figure 3), adding LATS-i to ALVO-EM
induces upregulation of AT1 markers, but the cells do not show AT1 phenotypes; some AGER⁺ or CAV1⁺ cells
remain proliferative, suggesting that AT1-like cells can divide – an event that does not occur in lung tissue.
Moreover, the majority of cells co-express AT2 markers and/or SCGB3A1 and SCGB3A2, indicative of
transitional or immature state. In Figure S3D, the claim that AT1/2-M organoids retain a mixture of AT1 and
AT2 cells is likely misleading; the data suggest that many cells co-express AT1 and AT2 markers, consistent
with immature or transitioning cells rather than a true mixture of two distinct cell types. Critically, in Figure
3I, LAMP3 expression remains stable under the AT1/2-M condition. Given these observations, the current
culture condition favors the maintenance of transitional or immature cells, despite LATS-i stimulation, and
still requires additional cues such as ECM components or mechanical signals, similar to previous methods. It
would be important to clarify what advantages and novel insights this protocol offers over prior approaches.
– *In Burgess et al (PMID: 38642558), the authors induce differentiation of primary AT2 cells with their L+DCI
medium and compare it to SFFF medium (PMID: 35355018). These two mediums are very different, most
notably the absence of CHIR in the differentiation medium. Other published strategies include the addition of
human serum to induce differentiation (PMID: 33128895). The strength of our approach is that the sole
variable between our expansion and our differentiation medium is LATS-inhibitor. This minimizes
confounding factors when comparing progenitor cells to differentiated cells and it enables precise
mechanistic studies. Furthermore, we do acknowledge the presence of immature or transitioning cells in the
manuscript, e.g. in the discussion “A biological explanation for the underrepresentation is that additional
signals or prolonged culture periods might be required for terminal AT1 differentiation. Nevertheless, we posit
that the presence of transitional states in our organoids better reflects regenerative conditions and provides a
valuable platform for optimizing AT1 differentiation in therapeutic contexts.”*

8. What is the rationale for adding IL-1 β to all organoid culture conditions during the first 0–3 days? Could
this early exposure precondition the cells and thereby confound their responsiveness to the additional
cytokines tested in Figure 5, making it difficult to interpret the screening results?
- We only add IL-1 β at p0, as it has been reported to increase organoid outgrowth (PMID: 33128895). The
screening of the cytokines was done at passage 6/7 after several months of in vitro expansion. We do not
think that the cells and their daughter cells have such a long-lived IL-1 β memory. Even if they did, coming
from human lungs, the cells would most certainly have been exposed to IL-1 β in vivo at some point due to
infections such as the common cold, which is something we cannot control for.
9. The findings related to differential effects of IFN- γ on cells from different organoid conditions are
intriguing, but overall analysis is limited. Given the presence of multiple cell types and states within these
organoids (Figure 4) and the lack of a cell-tracking system, it remains unclear which specific populations are
actually responding to IFN- γ , especially since the analysis relies on qPCR of only one or two genes and
viability assays on bulk organoids. Furthermore, AT1/2-M organoids seem to retain immature or transitional
cells instead of “AT1” cells.
- figure 5G shows that both AT2 and secAT2s respond with increased proliferation to IFN. Additionally, we
will further characterize the cells that survive IFN exposure in AT1/2-M conditions using stainings.
10. 10. In Major et al. (PMID: 32527928), although IFN- γ is not investigated (rather Type I and III interferons),
it is shown that interferon signaling is deleterious for lung epithelial regeneration in vivo during influenza A
models. In Lin et al. (PMID: 39352385), the authors show that IFN- γ signaling can lead to dysplastic lung
epithelial regeneration during an in vivo influenza A model, including differentiation of dysplastic-like cells
from AT2 cells in organoid culture. The authors show in Figure 5/S5 that IFN- γ exposure up to day 10 leads to
ALVO-EM expansion. Does this exposure to IFN- γ have consequences for later differentiation characteristics
of these populations? Experiments to address this may involve the short-term exposure to IFN- γ of ALVO-EM
organoids, (up to day 10, since longer exposure seems to have deleterious effects – see Figure S5F), with
subsequent passaging to AT1/2-M to see if their differentiation ability is altered compared to controls.
- We think that this is a good suggestion. We will differentiate organoids after IFN exposure to see if the
exposure impaired their differentiation capacity.
11. In Figure 6, how do macrophages influence the cellular phenotypes of ALVO-EM and AT1/2-M organoids
beyond their effects on organoid growth? Do they also impact differentiation? What specific evidence
supports the author’s conclusion that pre-treated macrophages selectively modulate secAT2/AT2
proliferation and AT1 cell death in these experiments?
- Our goal with these experiments is to show that we see the same effect on our progenitor cultures
(increased growth) and differentiated cultures (decreased growth) with macrophage-secreted IFN γ as we see
with the addition of recombinant IFN γ , adding more physiological relevance to the model. Since our focus is
not on macrophage-epithelial interactions, we feel that further characterization is beyond the scope of this
manuscript.
Minor comments:
1. Please update your diagram in Figure 1A and your methods to clarify your final culture conditions for initial
investigation of organoid growth characteristics in Figure 1/S1 – it seems that ultimately SFFF was used for

the initial unsorted 14 day culture, and ALVO-EM and other types were used upon passaging for p1 and
further, but this could use clarification. – This is indeed not clearly communicated and we will change the
phrasing and add explanations. In fact, we only used SFFF media (containing EGF) until we had optimized the
ALVO-EM conditions (replacing EGF with HRG1). To establish the organoid lines that we used in this
publication, we used both EGF and HRG1 in p0 and then switched to ALVO-EM from p1 onwards.

2. The UMAPs in Figure 1D combined with the text explanations are laborious to interpret with the cluster
assignments not given anywhere in Figure 1D. Furthermore, related to Major Point 4 above, it is critical for
the interpretation of this scRNA-seq data that certain clusters seem to be derived almost entirely from the
cells of one patient (Lu39). Thus, the data in Figure S1H would be better served alongside the UMAPs in
Figure 1D. – We will adjust the figures accordingly.

3. Related to the data in Figure 1 – for the secAT2 sorting and clustering, please clarify the details of the
passage number this was done at – for the scRNA-seq, presumably this was after passage 1 (~14 days after
sorting/plating of Epcam+ NGFR- cells and culture in ALVO-EM), and for the secAT2 organoids, these were
presumably sorted at p1 and the data displayed in Figure 1H-I are from p2 after this? These important details
are not currently included in the figure legends or Methods. – We will add passage information as requested.
The scRNA-Seq was done at p5. The data from figure 1H-I are at p4. Because we can expand, freeze, and
thaw our organoid lines as needed, there is no “linear time line”, but all experiments are done before
passage 10, as stated in the manuscript.

4. Line 119: what is the rationale saying a ‘more progenitor-like state’ based on the higher expression of
SCGB3A2? – we will change the phrasing to “more secretory-like state”, as SCGB3A2 is a marker for secretory
cells

5. Line 160: Should there be a reference to Figure 1H here? Figure 1H is seemingly not currently referenced
anywhere in the paper as of now. – indeed, thank you for pointing this out.

6. Line 546: “canonical tube” – the authors likely meant to say “conical tube” - indeed

7. In the Methods section “Making single cell suspensions and passaging organoids”, please clarify what
types of suspension culture plates were used and how many “drops” of cells/BME were included on each
plate (line 555), as these are important details for recapitulation of these methods. – we will add these
details

Reviewer #3 (Remarks on Protocol(s)):

Overall, the authors provide the detailed methods for this study. Please also see our comments to the
authors.

Dear Hans, dear Antonella,

Thank you for transferring your manuscript for consideration by the EMBO Journal, as well as sharing a preliminary point-by-point response on the concerns raised by experts during peer-review at another journal. I now went through your rebuttal response in detail and also discussed the matter with my editorial colleagues. I am pleased to share that we found your arguments sensible as such and the complementary experiments potentially suitable to address the major critique by the referees.

We can accordingly invite you to complete a revised version of your work for swift reconsideration by the existing referees for the EMBO Journal.

Related, I enclose formatting requirements for our venue at resubmission, Please let me know if there are any questions related.

Thank you for the opportunity to consider your work for publication. I look forward to receiving your revision.

Best regards,
Daniel

Daniel Klimmeck, PhD
Senior Editor
The EMBO Journal

Read our guidance for manuscript revisions and related editorial policies: <https://link.springer.com/journal/44318/submission-guidelines#cms-Revised-submissions>

<https://media.springernature.com/original/springer-cms/rest/v1/content/27825798/data/v1>

- a point-by-point response to the referees' comments, with a detailed description of the changes made (as a word file).
- a word file of the manuscript text.
- individual production quality figure files (one file per figure)
- a complete author checklist
- Expanded View files (replacing Supplementary Information)
- a Reagents and Tools Table as part of the Methods section

We realize that it is difficult to revise to a specific deadline. In the interest of protecting the conceptual advance provided by the work, we recommend a revision within 3 months (4th Mar 2026). Please discuss the revision progress ahead of this time with the editor if you require more time to complete the revisions.

Reviewer #1 (Remarks to the Author):

This manuscript described a modified protocol of generating and maintaining alveolar organoids from isolated
epithelial cells derived from primary human lobectomy material. The authors use this model (that was
improved with regard to a differentiation step (LATS-i) and with regard to prior expansion in ALVO-EM) to
demonstrate a role of recombinant IFN γ on organoid cell population behaviour, and extrapolate a role of IFN γ
in human COPD. While some aspects of the presented data are interesting, I have concerns regarding data
interpretation and conclusions drawn.

The authors use lobectomy material derived from lung cancer surgeries. No information is provided as to
patient history regarding age, gender, type of lung cancer, smoking history, COPD stage, or steroid medication
history, which might impact the composition and cellular responses of the organoids, particularly in light of the
findings on IFN γ and a putative role in COPD.

Given that the authors identify quite some heterogeneity in the different cultures (e.g., Fig 1C, S1H), and that
such criteria might significantly affect alveolar stem cell behaviour, such information should be provided to
better interpret the results.

On a minor note, might the inflammatory tumor microenvironment/signature within the lobectomy material
additionally impact the cells of origin used for organoids (how distant from tumors was the tissue used for
preps)?

*A: For ethical and practical reasons, we work with surgical "rest material" that is fully anonymized, so patient-*
*specific data (age, sex, smoking history, etc.) are unavailable. However, we do know that none of the patients*
*were diagnosed with COPD or other lung diseases beyond lung cancer, and that they did not receive cancer*
*treatment before surgery. We added this information to the material section "Samples were collected at least 5*
*cm from the tumor. Patients had no diagnosed lung diseases other than lung cancer and had received no*
*cancer treatment before surgery."*

*Regarding heterogeneity: variation between human donors is expected and intrinsic to primary organoid*
*systems. It is not possible to get tissue from a "normal, healthy" lung. Material either comes from lungs that*
*were rejected for transplantation, or from lobectomies, like in our case. We consistently use at least three*
*biological replicates to ensure reproducibility of observed responses and we show donor identifiers for full*
*transparency.*

The data on establishment of the new protocol seem to lack consistency, and clarity regarding information on
experimental setup, which makes interpretation difficult, and it remains unclear what message the authors
want to convey about their new protocol.

*A: We apologize for the confusion. We now reworked this whole part of the manuscript and added the final*
*media information to figure 1A and also adjusted table EV1.*

*As for the message, we see multiple advantages of our protocol over existing protocols:*

*1) Current approaches depend on HTII-280, a widely used AT2-specific membrane antigen whose*
*corresponding gene remains unknown. It is known that HTII-280-negative cells can generate SFTPC+*
*organoids (Basil, Nature, 2022) and that not all AT2 cells stain positive for HTII-280 (Evans et al., 2020, PMID:*
*32272001). This suggests that progenitors capable of forming AT2-like cells are excluded when HTII-280-*
*based sorting is used. We therefore see a conceptual advantage in our more inclusive approach.*

*2) We show that HRG1(replacing EGF) leads to better growth and higher expression of alveolar markers.*

*3) We achieve long-term expansion of AT2 progenitor cells AND secretory AT2 progenitor cells, which has not*
*been demonstrated before.*

*4) The sole variable between our progenitor expansion media and our differentiation media is LATS-inhibitor,*
*making precise mechanistic studies possible. This is in contrast to existing protocols, where differentiation is*
*achieved by the addition of human serum, or the addition of LATS-i in combination with Wnt-withdrawal.*

*5) We did extensive characterization of our organoids, including scRNA-Seq, which is not the case for existing*
*protocols (Katsura et al., CSC, 2020, Youk et al., CSC, 2020).*

*Together, we think that our system presents an advance to currently available protocols.*

For example, Fig 1C right panel lacks Lu44 that was referred to before to be the most long-lasting line (21

passages). Fig 1C and 1F again show different donors. Figure 1H shows data from donor 37, 53 and 54, and
Figure 1I only from 53 and 54.

*A: Due to practical reasons, it is not feasible to use the exact same three donors for every experiment. We
view the diversity of donors as a strength rather than a weakness, and we are fully transparent which lines
were used for which experiment. We did not continue using Lu44 because it represented an outlier when it
came to long-term expansion, and we chose to continue with more representative lines.*

The scRNASeq experiments (from donors 37-39) lack information about passage numbers and how long
these organoids have been in culture. Can the authors explain why there is such low amount of proliferating
cells, within the cells with progenitor phenotype that should reveal higher proliferation rates compared to AT2
cells (“secAT2”, C6+; “RAS-like”)?

*A: The scRNA-Seq was done at passage 5 with each passage being 14 days apart, as outlined in the methods
section “Making single-cell suspensions and passaging organoids”; we added the passage information to the
“10x Genomics scRNA-Seq” methods section of the manuscript and to the figure legend.*

*We would like to emphasize that our goal is not to model the alveolus at homeostasis, where epithelial cells
are quiescent, but rather a regenerative state. This is why organoid-derived cells form the majority of
proliferating cells when integrated with lung atlas data (Figure 4A). Under regenerative conditions (i.e. in
expansion media), we do not expect many mature, and therefore quiescent, AT2 cells. As Reviewer 2 pointed
out, the field increasingly recognizes that alveolar progenitors are heterogeneous. Our organoid conditions
promote stemness and proliferation, enriching two AT2-like progenitor subpopulations, one of which
resembles the secretory alveolar phenotype described in recent Nature papers (Murthy et al., Nature 2022,
and Basil et al., Nature 2022). These “secAT2” cells share features with AT0, TRB-SC, and AT2-3A2
populations, cells thought to lie along a continuum between secretory and alveolar lineages.*

In Figure 2D, there are organoids that are HT2-280 negative after AT2-MM culture, but Nile Red positive.
What are these organoids, and can the authors comment on the heterogeneity of their cultures?

*A: It has been stated elsewhere: “HTII-280 is a less useful marker for subculture of AT2 cells due to loss of
expression during culture.” (Evans et al., 2020, PMID: 32272001). This is why we show maturation with
functional dyes such as lysotracker and nile red instead. We agree that the presented image may be confusing
so we removed it from the figure. Instead, we added qPCR analysis of FACS-sorted nile red low and nile red
high cells in AT2-MM to show that nile red intensity indeed correlates with AT2-marker expression (figure
EV2D and S2E).*

It is difficult to assess the ultimate advantage of the presented over previously published models (e.g., PMID:
33128895), also in light of the results of Fig 3F and 3I indicating separation between C6+ and C6- cells seems
dispensable.

*A: Please see one of our previous answers (page 1; lines 37+) for the advantage of our system to the
published protocols. Figure 3F and 3I show that both AT2 and secAT2 populations can give rise to AT1-like
cells, but that does not make their separation dispensable. While we agree that in this manuscript we mostly
show transcriptional rather than functional differences between the two populations, we believe this system to
be useful to probe the differences in more detail in the future.*

Also, it remains open how well the ALVOs are suited to model the alveolus containing AT1 cells. Although AT1
markers are increasingly expressed (particularly in AT1/AT2-M, no matter if used directly or after ALVO-EM
expansion), there is little contribution from organoids compared to tissue to the AT1 pool when looking at
identity contribution (Fig 4E). Given that only very few AT1 cells are contained in the organoids, the abstract
statement that “Unlike previous models, this system supports long-term expansion of newly identified human-
specific alveolar progenitor cells and serum-free differentiation into alveolar type 1 (AT1) cells.” seems not to
be supported by data.

*A: The cells in our organoids show high expression of AT1 markers such as AGER at the protein level, which
is generally accepted by the lung field to indicate AT1 identity. We have acknowledged and discussed the
modest contribution of fully differentiated AT1 cells in the scRNA-Seq data in the discussion as follows:*

*“However, the majority of AT1-like cells have a gene expression profile resembling the AT2-to-AT1 transitional*

*state rather than that of fully differentiated AT1 cells. It is possible that organoid-derived differentiated AT1*
*cells are underrepresented in the scRNA-Seq dataset due to technical reasons. The typical shape of AT1 cells*
*makes this cell type especially sensitive to dissociation methods necessary to achieve single-cell suspensions.*
*A biological explanation for the underrepresentation is that additional signals or prolonged culture periods*
*might be required for terminal AT1 differentiation. Nevertheless, we posit that the presence of transitional*
*states in our organoids better reflects regenerative conditions and provides a valuable platform for optimizing*
*AT1 differentiation in therapeutic contexts.” Moreover, we changed the term AT1 to AT1-like cell in the*
*abstract and in the summary figures of figure 5 and 6.*

The culture type-specific effects of IFN γ are an interesting finding (ALVO-EM vs AT1/2 medium). The
conclusion that in particular, AT1-like cells are “negatively” affected would benefit from a more detailed
analysis (i.e. type/pathways of cell death induced by the cytokine, etc). Are the IFN γ effects observed on “AT1-
like” cells, or trans_AT2-AT1 cells (Fig 4), similarly observed in truly differentiated AT1 cells?

*A: We demonstrate a reduction in cell numbers (5A), an increase in cytotoxicity (5B), and an upregulation of*
*genes associated with cell death (5H). We also demonstrate reduced AT1 marker expression (5C), indicating*
*a selective loss of AT1 marker-expressing cells. It is difficult to define “truly differentiated AT1 cells”. Some*
*reports indicate that mechanical cues from breathing might be necessary to maintain “true” AT1 identity*
*(Shiraishi et al, 2024, doi: 10.1016/j.cell.2023.02.010). However, the goal of our organoid model is not to*
*achieve terminal AT1 differentiation in vitro but to model alveolar regeneration. We think that the observation*
*that IFN-g affects AT1-like or transitioning AT1 cells is highly relevant to acute injury, infection, and chronic*
*disease.*

In light of the different actions of IFN γ on secAT2/AT2 vs “AT1-like” cells, what is the expected “net outcome”
of IFN γ signaling in COPD according to the authors data, and how can this be interpreted with regard to
COPD (emphysema) lung phenotype, given that such lungs do not contain more or more proliferating
AT2/progenitor cells?

*A: An accumulation of secAT2s is reported in COPD lungs (Basil et al, Nature), the reason for which remains*
*unknown. However, the inflammatory landscape of COPD lungs is too complex to make a prediction regarding*
*a net outcome of the effect of IFN-g.*

How much “COPD phenotype” is imprinted in the organoids (given that likely, at least some of the donors were
smokers/COPD patients, see above)?

*A: As now added to the methods, none of the donors had a lung disease diagnosis apart from lung cancer.*
*The question of an imprinted smoking or COPD phenotype – even after months in organoid culture – is very*
*interesting but beyond the scope of this manuscript.*

On a minor note, why do the authors use PBMC-derived macrophage-like cells for co-cultures, given that T
cells are major sources of IFN γ ?

*A: This is for practical reasons: T cells need to be patient matched, which is not the case for macrophages.*
*While T cells are a major source for IFN- γ , macrophages also secrete the cytokine.*

All figure legends should contain information on n numbers, number of independent experiments, p values and
statistics applied.

*A: Our n numbers are always specified by the number of dots/symbols in the figures. Each symbol represents*
*a biological replicate, which is specified in the methods section. We added the information regarding the*
*statistical test to the figure legends and we now show the exact p values in the figures.*

Other:

• Fig 1A: which protocol (“published methods”) do the authors exactly refer to?

*A: We added this information to the figure. We specifically referred to Youk et al. and Katsura et al.*

• P4: what do the authors mean by “appeared healthier”? (line 114).

*A: We added the following sentence to the manuscript as a better explanation: "(...) the organoids grown in*
*HRG1-containing media appeared larger and more numerous compared to EGF (figure EV1D)."*

• How were the stats calculated in Fig 3I and 5G (for datapoints normalized to 1)?

*A: We now added all information regarding statistical testing to the figure legends and also updated the*
*methods section. All statistical testing was done on raw values and not on values that were normalized to 1.*
*The normalized data is only used for visualization purposes.*

• Fig S5F does not show cell death (line 348)

*A: The image of IFN-treated organoids at day 16 shows excessive shedding of cells and cell debris from the*
*organoids, which is an indication of dying cells. We increased the magnification of the images to show this*
*more clearly, and also added an explanation to the manuscript.*

13 **Reviewer #2 (Remarks to the Author):**

In this manuscript entitled "Interferon- γ selectively promotes survival of alveolar progenitor 1 cells in a human
2 organoid model" the authors describe the development of a novel approach to adult human alveolar
epithelial culture modelling with particular emphasis on an inclusive approach to cell isolation to ensure the
capture of all potential alveolar progenitors. They then define media components required for AT2 maturation
and AT2-AT1 differentiation from their mixed population. Having undertaken detailed characterisation, they
then use this model to identify the differing role of interferon- γ on the proliferation and survival of different
alveolar epithelial population, finding that interferon- γ is cytotoxic to AT1 cells but promotes survival of AT2
cells.

The manuscript is well written and easy to follow. The figures are carefully prepared and they present their
data clearly with appropriate analysis. Their transparency in presenting negative and variable data is
appreciated.

Their novel organoid model is potentially of great interest to the field. By taking a more unbiased approach to
sorting lung epithelial cells from fresh adult lung tissue than current protocols, they find different AT2 subtypes
(AT2, secretory AT2) and then push them toward different fates (AT2-like or AT1-like) by manipulation of
media components. As we increasingly recognise that alveolar progenitors are not simply one entity, the ability
to generate a manipulable model containing several AT2- and AT1-like subtypes is of real significance both in
disease modelling and understanding AT2 transdifferentiation in health and disease.

To convince the reader of their conclusions about the true originality of the model and the ability to use it for
the disease modelling described in the second part of the manuscript, I would suggest the following:

Major comments

1. The authors state in the introduction that using HTII-280 sorting strategies may exclude important alveolar
progenitors; indeed, they then use this argument for creating a novel culture strategy which does not rely on
this poorly characterised transmembrane protein. Though I agree with their concerns about HTII-280, the
absence of any data to support their statement is a concern. I would suggest:

• They should provide evidence that HTII-280 is absent from non-standard AT2 cells (e.g. by citing existing
scRNAseq datasets) to support the logic of their experimental approach.

*A: Because the corresponding gene to HT2-280 is not known, it cannot be looked up in scRNA-Seq data.*
*However, it is known that HTII-280-negative cells can generate AT2 organoids (Basil, Nature 2022, figure 3).*
*This suggests that progenitors capable of forming AT2 cells are excluded when HTII-280-based sorting is*
*used. Furthermore, it has been observed that "(...) although the majority of AT2 cells express HTII-280, HTII-*
*280- AT2 cells are observed even in healthy human lungs" (Evans et al., 2020, PMID: 32272001). We now*
*added this information and the additional references to the introduction.*

• They should show whether HTII-280-based AT2 isolation generates a similarly heterogeneous AT2-like

population, and whether the same manipulations of the culture media (i.e. use of AT2-MM / AT1/2-M) allow
maturation of AT2 and AT1-like cells from HTII-280-isolated organoids. This will likely highlight the key
advantages of this new model.

*A: To respond to this point (also raised elsewhere), we sorted p0 organoids at day 14 for EPCAM+/NGFR-
5 /HT2-280+ (HT2+) and EPCAM+/NGFR-/HT2-280- (HT2-) populations. Because this experiment required us
to get new lung tissue and the total cell numbers we obtain per tissue and for each of the populations varies
between donors, we could not do all requested experiments with three biological replicates within a
reasonable time frame. We were able to show that both HT2+ and HT2- cells lead to organoid outgrowth in
HRG1-containing media (figure EV1C) and that both populations lead to cultures that have high SFTPC and
SCGB3A2 expression (figure EV1E). SFTPC expression levels of HT2- derived cells were lower compared to
HT2+ derived cells, indicating that the starting cell population does make a difference in subsequent gene
expression of cells. Due to cell number restraints, we were only able to derive flow analysis of one lung at p1
13 day 14. We found that HT2+ derived organoids had more HTII-280 signal and less CEACAM6 signal than
14 HT2- derived organoids (in HRG1-containing cultures) (reviewer figure C). However, the presence of HTII-
15 280+ cells that were derived from HT2- cells indicates that indeed HT2- cells can give rise to HTII-280+ cells.
This supports our reasoning of not relying on HTII-280 sorting for our organoid cultures.*

2. Figures 5 and 6 describe the differential response of AT2-like and AT1-like cells to interferon- γ . By
necessity, the culture media for each cell type is different with AT1-like cells maintained in LATS-i. Bearing in
mind the crosstalk between LATS and IFN pathways, how are the authors sure that the observed effects on
cell proliferation and survival due to cell type and not the effect of LATS-i?

The following would be reassuring:

• To demonstrate that LATS-i does itself not affect cell proliferation and survival.

*A: LATS-i in fact increases the proliferation rate of our organoids. We added this data to the manuscript (figure
EV3F). Of note, in figure 5F we compare the IFN-treated cultures to their own controls and therefore normalize
for difference in cell growth between the ALVO-EM and AT1/2-M conditions.*

• To check that the combination of LATS-i and interferon- γ does not affect the phenotype of the AT1-like cells
grown in AT1/2-M.

*A: The cytotoxicity assays are conducted in differentiated cultures that were exposed to IFN- γ for only 3 days
(5B) and we do not expect drastic changes in cell identity within this time frame.*

Other comments

1. The introduction opens with a brief discussion of COPD and the need for better models of the effect of
inflammatory insults in the lung. While I agree, this really isn't the focus of the manuscript which has far
broader reach in modelling the alveolar epithelium in health and disease. Indeed, I would suggest the models
they describe would be more immediately relevant to modelling diseases of the alveolus e.g. ILD. To this end,
I would suggest the introduction (and discussion) are reworded to downplay the emphasis on COPD.

*A: Well taken. We rephrased the introduction to lessen the focus on COPD.*

2. Fig S1E&F: AGER is used in S1E then CAV1 in S1F as AT1 markers. Subsequently both are used. For
transparency and continuity, I would suggest including both in Fig S1.

*A: Thank you. We added the requested data.*

3. Fig 2B/Fig S2A&B: There is an assumption here that higher lysotracker = more AT2-like. It would be nice to
have a) freshly isolated AT2 cells (e.g by HTII-280 sorting) as a control to define physiological levels of
lysotracker, and b) a control line which does not have LBs as a readout for physiological lysosome-resident
lysotracker staining.

*A: We added this data for both lysotracker and Nile red staining, using airway organoid-derived cells as a
negative control and tissue sorted HT2-280+ cells as a positive control. Our data confirms that our organoid-
derived cells exhibit lysotracker- and Nile red-signals comparable to tissue AT2s (figure 2E).*

4. Fig 2C: The EM of secreted surfactant is beautiful – thank you for including this. The contrast and zoom in
the middle panel could be improved to make the LBs more convincing, however.
*A: We are also very excited about these images! We improved the zoom and contrast as suggested.*
5. Line 196: It has been shown that LATS manipulation alone can drive an AT1 phenotype in organoids
derived from precursors (e.g. Burgess et al PMID 38642558; Lim et al PMID 39815007). This should be
mentioned.
*A: In both mentioned publications, LATS-i is used in combination with Wnt withdrawal to induce AT1*
*differentiation, and not by itself. We do show in our own manuscript that Wnt withdrawal alone can cause AT1*
*differentiation, even without the addition of LATS-i (Figure EV3A, base media “BM”). Our observation that*
*LATS-i alone can drive AT1-differentiation even in the presence of Wnt-activating agents was to our*
*knowledge not described previously. We added this distinction now more specifically to our manuscript.*
6. In Fig 5, ruxolitinib is used at concentrations of 100nM-1uM then at 1uM in Fig 6. Can the authors justify this
choice? The IC50 appears to vary widely according to cell type, but these seem to be high concentrations.
Might there be off-target effects?
*A: We tested the drug at 0.1uM, 1uM, and 10uM (figure EV5B) and saw no cytotoxicity at 0.1uM and 1uM. We*
*therefore tested 0.1uM and 1uM in our rescue studies (figure 5D, 5E, S5, S5C, S5D) and saw a dose-*
*dependent effect, with a partial rescue of viability, AT1 gene expression, and suppression of IFN target genes*
*at 0.1uM and a full rescue at 1uM. Because we aimed for a full rescue, we chose 1uM in our macrophage*
*studies in figure 6. Ruxolitinib is used widely, and the dose-dependent repression of IFN target genes*
*indicates that the drug is on-target. We found several studies that use Ruxolitinib at concentrations between 1-*
*10uM in organoids (Katsura et al, CSC, 2020 doi: 10.1016/j.stem.2020.10.005; Kolawole et al., 2019, doi:*
*10.1371/journal.ppat.1008057)*
7. In Fig 5A&F, different symbols for each donor would help understanding of the individual responses to
interferon-γ.
*A: We added the different symbols to the figure.*
8. In Fig 5C, SFTPA is used as an AT2 marker rather than SFTPC (which is used everywhere else). Why is
this?
*A: We apologize for the inconsistency. We now also did the analysis for SFTPC and added this to the figure.*
9. Figure 5J demonstrates a cytotoxic effect of BIRC-i in interferon-γ cells grown in ALVO-EM. This experiment
should also be undertaken in in AT1/2-M-cultured cells, despite there being no upregulation of BIRC3 in this
population.
*A: We undertook this experiment and added it to the manuscript. “In AT1/2-M, both the addition of LCL161*
*alone and in combination with IFN-γ caused an increase in cytotoxicity (S5H). This indicates that even though*
*BIRC3 is not upregulated upon IFN-γ in AT1-like cells, it might still play a protective role in these cells in*
*general.”*
10. In figure 6, activated macrophages are used as a source of interferon-γ, but there are no data to confirm
that this is the case. Data should be included to confirm interferon-γ production, and ideally that (as stated in
line 394-395) that a consistent supply of interferon-γ is indeed being provided when cells are replaced every 3-
4 days.
*A: Previous studies show that LPS and IFN-g stimulation of macrophages leads to a pro-inflammatory (“M1-*
*like”) phenotype and IFN-g secretion (doi: 10.3389/fimmu.2019.01084, PMID: 29921905,*
*<https://doi.org/10.1093/intimm/5.11.1383>, PMID: 24680476). Furthermore, the fact that we get a rescue of the*
*phenotype (figure 6E) with the JAK/STAT inhibitor ruxolitinib indicates that it is indeed IFN-g causing the*
*phenotype.*
11. In figure 3G, the authors show a significant difference in AT-1 like phenotypes emerging dependent on the

ALI transwell culture coating. This is not explored further in this manuscript, but is important. I would suggest
this is expanded on in the discussion.

*A: We now expanded on this in the discussion: "Moreover, 2D cultures may be better suited to capture the*
*characteristic morphology of AT1 cells. AT1 cells are tightly associated with the basement membrane, a*
*relationship that is central to their function. We show that AT1 cell shape varies markedly depending on the*
*ECM coating of the transwells. Although this was not the primary focus of the present study, further*
*optimization of ECM composition to enhance AT1 attachment and differentiation may enable in vitro systems*
*that better support functional testing of AT1 cells."*

Minor points

Fig S1A: The SFTPC signal is very hard to see in row 3, column 2 (I assume this is supposed to represent
positive staining)

*A: We added magnifications to the figure EVo that the stainings are easier to see.*

Fig 2C legend: Spelling (DIC)

*A: We corrected this.*

Fig 4E (legend on right), Fig S4A,B,F,H: The text is too small to read

*A: We improved the readability of the mentioned figures.*

Line 102: A reference, and ideally a brief justification, is required to justify negative sorting of NGFR

*A: NGFR is a well-established surface molecule used to sort lung basal cells, the stem cells of the airway*
*epithelium. We added references for this to the manuscript (<https://doi.org/10.1073/pnas.0906850106> and*
*<https://doi.org/10.1165/rcmb.2020-0464OC>)*

Line 563: I assume this should read 10 million cells/ml

*A: Good catch!*

Line 710: Spelling (Quiagen)

*A: Thanks*

Reviewer #2 (Remarks on Protocol(s)):

Very well written and comprehensive

Reviewer #3 (Remarks to the Author):

In this manuscript, Dost et al. present a new set of protocols for deriving human distal lung organoids with
varying differentiation statuses. Using these culture systems, the authors examine how IFN- γ treatment
differentially affects organoids depending on the epithelial cell types present. While the study provides
improved methods for expanding organoids from primary human lung epithelial cells and includes detailed
characterization, several critical concerns remain unresolved, as outlined below.

Major comments:

1. Overall, although the sorting-free approach for the initial culture of lung single-cell suspensions and the
inclusion of potential distal progenitor cells capable of generating alveolar lineages may enhance cell survival
and expansion, the manuscript does not fully substantiate its conclusions, partly due to the heterogeneity of
the starting cell population that gives rise to organoids. Specifically,
a. The authors should contextualize better the rationale for using NGFR to enrich "progenitor cells" for alveolar
organoid generation. What does NGFR mark within P0 organoids? The authors should demonstrate what

populations exist in EpCAM+NGFR- cells. Flow cytometry for EpCAM, NGFR and HTII-280 require to show
cellular profiling. scRNA-seq of EpCAM+NGFR- vs EpCAM+HTII-280+ cell comparison is required to
determine the initial populations in their P0 organoids. Interpretation is further hindered by the lack of scRNA-
seq or other profiling data at the critical timepoint between P0 and the subsequent EpCAM+NGFR- sorting,
when cellular heterogeneity would be manifested. The authors should include such data to clarify which cell
types are present and potentially contribute to the observed progenitor diversity.

*A: NGFR is a well-established surface molecule used to sort lung basal cells, the stem cells of the airway
epithelium (<https://doi.org/10.1073/pnas.0906850106> and <https://doi.org/10.1165/rcmb.2020-0464OC>). We
added these references to the manuscript. Our rationale is to exclude basal cells from our alveolar organoid
cultures. To satisfy this request, we obtained new lung tissues and sorted p0 organoids at day 14 for
EPCAM+/ a) NGFR-/HT2-280+ (HT2+), b) NGFR-/HT2-280- (HT2-), and c) NGFR+ populations for
subsequent qPCR profiling (reviewer figure A+B). Because total cell numbers and population sizes vary widely
between human lung samples, we were only able to obtain this profiling for 2 lungs. As expected, NGFR+ cells
have high expression of basal cell markers KRT5 and TP63 and low expression of Club (SCGB1A1) and AT2
(SFTPC, LAMP3) markers. NGFR-/HT2-280+ cells have high expression of AT2 markers and low expression
of basal cell markers. NGFR-/HT2-280- cells fall in between. They express lower levels of AT2 markers than
HT2+ cells but higher than NGFR+ cells. They express lower levels of basal cell markers than NGFR+ cells
but higher than HT2+ cells. They also express some of the secretoglobins (SCGB3A1/3A2/1A1) but the
expression varies between samples and no clear trend is visible. In summary, NGFR-/HT2-280- co-express
alveolar and airway (basal and secretory) markers, similar to what can be found in the respiratory bronchioles.*

b. What does CEACAM6 specifically mark in this culture? Although the authors state that CEACAM6 is a
marker for “RAS” cells based on Basil et al. (2022), CEACAM6 expression is known to span a broad range of
distal epithelial populations, including AT0, TRB-SC/RAS, club, goblet, and AT1 cells. Figure 1D also showed
heterogeneity of CEACAM6-expressing cells in organoids.

*A: As shown in figure 1D and 1G, CEACAM6 expression correlates with reduced, albeit still high, expression
of SFTPC and increased expression of the secretory markers SCGB3A1 and SCGB3A2. Figure 1E shows
varying degrees of co-expression of CEACAM6 and SFTPC on protein level. The club cell marker SCGB1A1
is not expressed. Moreover, our CEACAM6+ cells in ALVO-EM do not express higher levels of AT1 markers,
which is the case for AT0 cells. The expression pattern of the organoid secAT2s most closely aligns with the
expression profile of TRB-SCs (Murthy et al, Nature, 2022) or AT2/3A2 cells (Basil et al, Nature, 2022).
The mentioned populations (AT0, TRB-SC, RAS, AT2/3A2) are very recent cell type/cell state discoveries that
are not well described or understood yet. Because the populations were described by two separate labs
independently, there are different nomenclatures that likely describe overlapping populations. It is likely that
these cell types represent a spectrum from secretory to alveolar cells, with varying levels of secretory and
alveolar gene co-expression. We do not claim that our organoids contain one specific cell population/state that
corresponds to the populations described in vivo. Instead, we chose the term secAT2 cells, because they co-
express secretory and alveolar genes, one of the common themes of the newly described populations.*

c. What are secAT2 cells? Based on the presented data, these cells appear to resemble AT0-like population,
reported by Murthy et al. (2022). However, Murthy et al. demonstrated that AT0 cells exhibit markedly reduced
SFTPC expression compared to AT2 cells, whereas the so-called secAT2 cells in this study show SFTPC and
LAMP3 levels comparable to AT2 cells (Figure 1G, J.). Does this imply that secAT2 cells are an in vitro
artifact, a culture-specific intermediate state, or a transitional population bridging AT2 and AT0 states?
Critically, Murthy et al. further demonstrated that AT2 cells transition into AT0 cells and subsequently into
TRB-SC (SFTPB+SCGB3A2+SFTPC-) cells upon EGF withdrawal. In contrast, in the current study (Figure
EV1E), EGF withdrawal appears to increase both SFTPC and SCGB3A2 expression. The authors also claim
that ALVO-EM culture promotes CEACAM6 and SCGB3A2 expression in “AT2” cells (CEACAM6-neg AT2) –
is this effect simply attributable to the absence of EGF in ALVO-EM? Moreover, Murthy et al. reported
unidirectional differentiation of AT2 toward AT1 or TRB-SC lineages through AT0 states. Given this, the
rationale for including secAT2 cells to enrich AT2 cells or to improve alveolar organoid formation requires
clearer justification. Are secAT2 and AT2 cells interconvertible under this culture condition? To substantiate
these findings, it would be important to determine whether HTII-280+AT2 cells (EpCAM+NGFR-HTII-280+ vs

EpCAM+NGFR-HTII-280-) can give rise to secAT2-like or AT0-like cells – or vice versa – when cultured with
EGF or HRG1.

*A: We do not claim that our organoids contain one specific cell population/state that corresponds to the*
*populations described in vivo. Our rationale for including secAT2 cell is that such a cell type/state likely*
*contributes to alveolar regeneration. In mice, club cells and bronchioalveolar stem cells (BASCs, co-*
*expressing the secretory club cell marker SCGB1A1 and the AT2 marker SFTPC) contribute to alveolar repair.*
*In humans, the respiratory bronchioles contain multiple populations that likewise show secretory-alveolar gene*
*co-expression (AT0, TRB-SC, RAS, AT2/3A2). If and how these cells contribute to alveolar regeneration in*
*humans is not known, but it has been demonstrated that iPSC-derived RAS and primary human RAS cells*
*(HT2-280-/CEACAM6+) can give rise to AT2 cells (Pezet et al, 2025 (doi: 10.1038/s41587-025-02569-0), and*
*Basil, 2022, Nature). This data implies a trajectory from secretory to alveolar cells.*

*While it has been shown that withdrawal of EGF can cause transdifferentiation of AT2s to TRB-SC through an*
*AT0 state, our organoids are cultured without EGF over a long time period and still contain an AT2 population,*
*indicating that we have two stable progenitor populations in our organoid cultures.*

*To address the effect of HRG1/EGF in our cultures, we sorted for EpCAM+/NGFR-/HTII-280+ and*
*EpCAM+/NGFR-/HTII-280- as suggested and cultured these two populations in media containing either EGF*
*or HRG1. We show that organoids grown in HRG1 grow better and express higher levels of the AT2 marker*
*SFTPC and the RAS/progenitor marker SCGB3A2 (new figure EV1C+E). We also show that HRG1 (NRG1)*
*and its receptors ERBB3/4 are expressed in alveolar cells (figure EV1D), adding physiological relevance to the*
*culture conditions.*

2. Although substituting EGF with the alternative EGFR family ligand HRG1 in alveolar organoid cultures is an
interesting modification, it appears to only modestly improve organoid propagation. Previous study using SFFF
medium containing EGF (Katsura et al. 2020) already demonstrated long-term subculture beyond 10
passages. In the current study, while the authors report somewhat improved expansion, SFTPC and SFTPA1
expression levels decline after passage 10 (Figure EV1G), restricting organoid use to the first 10 passages.
Therefore, the practical benefit of this protocol appears limited. Furthermore, a similar sorting-free approach
was previously employed by Salahudeen et al. (2020) to generate human distal lung organoids from primary
lung tissues. They used EpCAM+Lysotracker+ cells from P0 organoids, which likely contain AT2 and secAT2
(or AT0 cells) based on the current study. So, it raises an important question – would this strategy provide a
more effective means to enrich AT2 and/or potential progenitor-like populations than the current use of
EpCAM+NGFR- cells, given the non-specificity of NGFR and the resulting cellular heterogeneity?

*A: We refer the referee to page 1 line 37+ of this rebuttal letter for a detailed explanation regarding the*
*advantages of our system. Another practical benefit is that the FACS/MACS sorting is not done on the day of*
*the tissue processing. In practicality, receiving tissue from the hospital, processing it to single cell*
*suspensions, then sorting and plating the cells is a long and harsh process that reduces cell viability. There*
*are other practical considerations rarely talked about in scientific publications, like the availability of a FACS*
*machine on the day the tissue is received, which can be unpredictable.*

*Regarding the EpCAM+/Lysotracker+ sorting strategy from Salahudeen et al. (2020): Lysotracker signal*
*corresponds to the maturity level of AT2 cells. However, it has been shown that AT2 “stem/signaling” cells*
*(PMID: 33208946), and also the newly described progenitor populations residing in the respiratory bronchioles*
*(AT0, TRB-SC, RAS) show lower levels of canonical AT2 markers. It makes sense that proliferating AT2*
*progenitor cells would not be at the same level of maturity as quiescent AT2 cells that produce high levels of*
*surfactant. AT2 progenitor cells would therefore likely have lower levels of lamellar bodies and lysotracker*
*staining, which is why we think that lysotracker sorting is a less inclusive approach than our sorting strategy.*
*We see the heterogeneity of our ALVO-EM cultures as an advantage because it gives us the opportunity to*
*start studying the secretory-alveolar populations in vitro.*

3. Regarding the use of HRG1, the signaling aspects related to this alteration require further exploration. Since
HRG1 does not bind to EGFR but rather ERBB3 and ERBB4, what are the levels of expression of each of
these receptors in secAT2 and AT2 cells during organoid culture? Ideally, this should be assessed at the
timepoint following the unsorted culture of lung single cell suspensions at day 14 prior to sorting for further
organoid culture. Are there differences in downstream signaling between these populations when cultured with

EGF vs. HRG1? Does HRG1 specifically drive the generation of secAT2 cells from AT2 cells, or is this effect
simply due to the absence of EGF, as noted above? Finally, what are the cellular sources of HRG1 in lung
tissues, and what is the expression pattern of its receptors within these tissues?

*A: We added lung tissue data showing that HRG1 (NRG1) and its receptors ERBB3/4 are expressed in*
*alveolar cells (figure EV1D), adding physiological relevance to the culture conditions. Moreover, ERBB3 and*
*ERBB4 are expressed in our organoid cultures, both in the AT2 and secAT2 populations (reviewer figure D).*
*To address the effect of HRG1/EGF in our cultures, we sorted for EpCAM+/NGFR-/HTII-280+ and*
*EpCAM+/NGFR-/HTII-280- and cultured these two populations in media containing either EGF or HRG1. We*
*show that organoids cultured in HRG1 grow better and express higher levels of the AT2 marker SFTPC and*
*the RAS/progenitor marker SCGB3A2 (new figure EV1C+E). We also show that HT2- derived organoid*
*cultures still express high levels of SFTPC, but lower than HT2+ derived organoids. This indicates that both*
*the initial population (HT2+/-) and the culture conditions (HRG1/EGF) affect the identify of outgrowing cells.*

4. In Figure EV1D and E, organoids cultured without HRG1 and EGF revealed poor organoid formation but
higher expression levels of SFTPC and AGER. The authors should provide an explanation for this
observation.

*A: Removal of these factors decreases proliferation of the progenitor cells. In the absence of these pro-*
*proliferative and pro-stemness cues, the cells likely mature/differentiate to either AT2 or AT1 cells. However,*
*this effect is very small compared to our AT1 differentiation protocol where AGER is upregulated almost 100*
*fold (figure 3I). We now restructured figure 1, following reviewers' requests, and removed the data referred to*
*in this comment. In the end, the viability and growth of cultures without EGF or HRG1 was so poor that gene*
*expression analysis on these cells did not add any valuable information to the manuscript.*

5. Single cell RNA-seq data from three separate patients of organoids from initial single cell culture in ALVO-
EM (Figure 1D/S1H-I), and after switching to ALVO-EM, AT1/2-M, and AT2-MM (Figure 4C-E/S4A-C), clearly
show that one patient (Lu39) is responsible for the majority of 3 separate clusters in Fig. 1/S1, and furthermore
is responsible for the vast majority of neuroendocrine-like cells from the different culturing conditions in Figure
4/S4 (see most pertinently Figure EV4C). However, the authors do not refer to this clear demarcation in the
Results section related to each of these figures. The authors should update the Results section to
communicate to the readers that this is the case, probably attributable to heterogeneity in human samples
taken for organoid culture. Furthermore, the authors may want to soften their interpretation of these results as
representative of general principles of neuroendocrine phenotype emergence in DCI-containing organoid
cultures (see Discussion, line 425-430), given that 2/3 patient samples do not seem to appreciably show this
conversion.

*A: We added qPCR data showing that we also observe upregulation of ASCL1, an important transcription*
*factor of neuroendocrine identity, in Lu37 and Lu38 (figure EV4D). The qPCR shows that the baseline*
*expression of ASCL1 is highest in Lu39, which might be the reason why the scRNA-Seq mainly picked it up in*
*this line. While the neuroendocrine phenotype is not the focus of the paper, we think that it is an interesting*
*observation to point out in the discussion, and this observation is not only based on our own data but also data*
*from the referenced publication (Lim et al, 2025).*

6. In Figure 2A, what was the initial cell population used for this experiment? Did the authors test whether AT2
and secAT2 cells differ in their capacity to differentiate into mature AT2 cells?

*A: Unless stated otherwise, the initial cell population is ALVO-EM cultured organoid-derived cells. To answer*
*the question whether AT2 (C6+) and secAT2 (C6-) cells differ in their AT2 maturation capacity, we sorted*
*these two populations and exposed them to our AT2-MM conditions. After the maturation period, both*
*populations showed high levels of Nile red staining compared to AO-derived cells, with no clear trend among*
*the three tested lines (reviewer figure E).*

Given the heterogeneity of this culture, it is surprising that minimal differences were observed in response to
the screening factors. Furthermore, in Figure 2D, the majority of cells in the ALVO-EM condition do not
express HTII-280 despite of their high expression of AT2 markers shown in Figure 1. Additionally, in the AT2-

MM condition, one organoid contains Nile red+ lipids but is negative for HTII-280, whereas another organoid
shows lower lipid content but high HTII-280 expression. How do the authors explain this discrepancy?

*A: It has been observed that “HTII-280 is a less useful marker for subculture of AT2 cells due to loss of*
*expression during culture.” (PMID: 32272001). This is why we show maturation with functional dyes such as*
*lysotracker and Nile red instead. We agree that the presented image was not optimal so we removed it from*
*the figure. Instead, we added a flow histogram and qPCR analysis of FACS-sorted Nile red-low and Nile red-*
*high cells in AT2-MM to show that Nile red intensity indeed correlated with AT2-marker expression (figure*
*EV2D and S2E). It is also noteworthy that by flow analysis, all AT2-MM-derived cells have high levels of Nile*
*red staining, comparable to primary tissue AT2 cells and much higher than airway organoid (AO)-derived cells*
*(figure EV2D). We therefore do not think that there is noteworthy heterogeneity in the AT2-MM organoids.*

Since DCI is known to promote alveolar maturation in both iPSC-derived and fetal lung organoids, a similar
effect is likely in primary alveolar organoids. Does HRG1 modulate this maturation process differently than
EGF?

*A: Both iPSC- and fetal-derived organoids are in a much earlier developmental state compared to primary*
*adult cells, and increased maturation with DCI has to our knowledge not been described in adult AT2s. To*
*answer the question regarding HRG1/EGF, we compared the maturation of C6- and C6+ cells in AT2-MM*
*(which contains HRG1) and AT2-MM without HRG1 with the addition of EGF. We found that HRG1-containing*
*media led to slightly higher Nile red signal in the C6- cells compared to EGF-containing media, while there was*
*no difference in C6+ cells (reviewer figure F). Therefore, HRG1 has a slight advantage over EGF when it*
*comes to AT2 maturation.*

7. LATS-i has previously been reported to induce AT1 differentiation in both iPSC- and primary lung tissue-
derived alveolar organoids (Burgess et al. 2024). In this current study (Figure 3), adding LATS-i to ALVO-EM
induces upregulation of AT1 markers, but the cells do not show AT1 phenotypes; some AGER⁺ or CAV1⁺ cells
remain proliferative, suggesting that AT1-like cells can divide – an event that does not occur in lung tissue.
Moreover, the majority of cells co-express AT2 markers and/or SCGB3A1 and SCGB3A2, indicative of
transitional or immature state. In Figure EV3D, the claim that AT1/2-M organoids retain a mixture of AT1 and
AT2 cells is likely misleading; the data suggest that many cells co-express AT1 and AT2 markers, consistent
with immature or transitioning cells rather than a true mixture of two distinct cell types. Critically, in Figure 3I,
LAMP3 expression remains stable under the AT1/2-M condition. Given these observations, the current culture
condition favors the maintenance of transitional or immature cells, despite LATS-i stimulation, and still requires
additional cues such as ECM components or mechanical signals, similar to previous methods. It would be
important to clarify what advantages and novel insights this protocol offers over prior approaches.

*A: In Burgess et al (PMID: 38642558), the authors induce differentiation of primary AT2 cells with their L+DCI*
*medium and compare it to SFFF medium (PMID: 35355018). These two mediums are very different, most*
*notably in the absence of CHIR in the differentiation medium. Other published strategies include the addition*
*of human serum to induce differentiation (PMID: 33128895). The strength of our approach is that the sole*
*variable between our expansion and our differentiation medium is LATS-inhibitor. This minimizes confounding*
*factors when comparing progenitor cells to differentiated cells and it enables precise mechanistic studies.*
*Furthermore, we do acknowledge the presence of immature or transitioning cells in the manuscript, e.g. in the*
*discussion “A biological explanation for the underrepresentation is that additional signals or prolonged culture*
*periods might be required for terminal AT1 differentiation. Nevertheless, we posit that the presence of*
*transitional states in our organoids better reflects regenerative conditions and provides a valuable platform for*
*optimizing AT1 differentiation in therapeutic contexts.”*

8. What is the rationale for adding IL-1 β to all organoid culture conditions during the first 0–3 days? Could this
early exposure precondition the cells and thereby confound their responsiveness to the additional cytokines
tested in Figure 5, making it difficult to interpret the screening results?

*A: We only add IL-1 β at p0, as it has been reported to increase organoid outgrowth (PMID: 33128895). The*
*screening of the cytokines was done at passage 6/7 after several months of in vitro expansion. We do not*
*think that the cells and their daughter cells have such a long-lived IL-1 β memory. Even if they did, coming from*
*human lungs, the cells would most certainly have been exposed to IL-1 β in vivo at some point due to*

*infections such as the common cold, which is something we cannot control for.*

9. The findings related to differential effects of IFN- γ on cells from different organoid conditions are intriguing,
but overall analysis is limited. Given the presence of multiple cell types and states within these organoids
(Figure 4) and the lack of a cell-tracking system, it remains unclear which specific populations are actually
responding to IFN- γ , especially since the analysis relies on qPCR of only one or two genes and viability
assays on bulk organoids. Furthermore, AT1/2-M organoids seem to retain immature or transitional cells
instead of “AT1” cells.

*A: Figure 5G shows that both AT2 and secAT2s respond with increased proliferation to IFN. Additionally, we*
*changed “AT1” to “AT1-like” throughout the text and in our summary figure 5K.*

10. 10. In Major et al. (PMID: 32527928), although IFN- γ is not investigated (rather Type I and III interferons),
it is shown that interferon signaling is deleterious for lung epithelial regeneration in vivo during influenza A
models. In Lin et al. (PMID: 39352385), the authors show that IFN- γ signaling can lead to dysplastic lung
epithelial regeneration during an in vivo influenza A model, including differentiation of dysplastic-like cells from
AT2 cells in organoid culture. The authors show in Figure 5/S5 that IFN- γ exposure up to day 10 leads to
ALVO-EM expansion. Does this exposure to IFN- γ have consequences for later differentiation characteristics
of these populations? Experiments to address this may involve the short-term exposure to IFN- γ of ALVO-EM
organoids, (up to day 10, since longer exposure seems to have deleterious effects – see Figure EV5F), with
subsequent passaging to AT1/2-M to see if their differentiation ability is altered compared to controls.

*A: To answer this question, we performed the following experiment: We exposed organoids in ALVO-EM to 1*
*ng/ml IFN-g for 7 days, removed the cytokine for 3 days, then switched the organoids to AT1/2-M*
*differentiation condition for an additional 7 days. We performed gene expression analysis on day 10 (start of*
*differentiation) and day 17 (end of differentiation). We found that the exposure to IFN-g did not influence*
*subsequent AT1 differentiation capacity. The previously exposed cells showed upregulation of AT1 markers*
*CAV1 and AGER to the same level as the control cells (figure EV5I).*

11. In Figure 6, how do macrophages influence the cellular phenotypes of ALVO-EM and AT1/2-M organoids
beyond their effects on organoid growth? Do they also impact differentiation? What specific evidence supports
the author’s conclusion that pre-treated macrophages selectively modulate secAT2/AT2 proliferation and AT1
cell death in these experiments?

*A: Our goal with these experiments is to show that we see the same effect on our progenitor cultures*
*(increased growth) and differentiated cultures (decreased growth) with macrophage-secreted IFNg as we see*
*with the addition of recombinant IFNg, adding more physiological relevance to the model. Since our focus is*
*not on macrophage-epithelial interactions, we feel that further characterization is beyond the scope of this*
*manuscript.*

Minor comments:

1. Please update your diagram in Figure 1A and your methods to clarify your final culture conditions for initial
investigation of organoid growth characteristics in Figure 1/S1 – it seems that ultimately SFFF was used for
the initial unsorted 14 day culture, and ALVO-EM and other types were used upon passaging for p1 and
further, but this could use clarification.

*A: We apologize that this was not clearly presented. We have now rewritten this part of the manuscript and*
*added the final media information to figure 1A and also adjusted table EV1. To establish the organoid lines*
*that we used in this publication, we used both EGF and HRG1 in p0 (which would correspond to SFFF media*
*with the addition of HRG1). For clarification, we now termed this media ALVO-p0. From p1 onwards, we*
*switched to ALVO-EM.*

2. The UMAPs in Figure 1D combined with the text explanations are laborious to interpret with the cluster
assignments not given anywhere in Figure 1D. Furthermore, related to Major Point 4 above, it is critical for the
interpretation of this scRNA-seq data that certain clusters seem to be derived almost entirely from the cells of
one patient (Lu39). Thus, the data in Figure EV1H would be better served alongside the UMAPs in Figure 1D.

*A: We added the cluster and donor ID annotations to the main figure 1D for easier interpretation. For further*
*clarification, we focus our DEG dotplot on the three large main clusters that are made up of all three donors*
*(figure EV1I). We also added a rescaling step that was previously missing, which is the reason why the UMAP*
*shape looks different than before. This does not change the data or the interpretation thereof.*

3. Related to the data in Figure 1 – for the secAT2 sorting and clustering, please clarify the details of the
passage number this was done at – for the scRNA-seq, presumably this was after passage 1 (~14 days after
sorting/plating of Epcam+ NGFR- cells and culture in ALVO-EM), and for the secAT2 organoids, these were
presumably sorted at p1 and the data displayed in Figure 1H-I are from p2 after this? These important details
are not currently included in the figure legends or Methods.

*A: We added the passage information to the figure legends as requested. The scRNA-Seq was done at p5.*
*The data from figure 1H-I are at p4. Because we can expand, freeze, and thaw our organoid lines as needed,*
*there is no “linear time line”, but all experiments are done before passage 10, as stated in the manuscript.*

4. Line 119: what is the rationale saying a ‘more progenitor-like state’ based on the higher expression of
SCGB3A2?

*A: We change the phrasing to “(...) the organoids expressed higher levels of the (...)secretory gene SCGB3A2*
*(figure EV1E), recently described as a marker for multiple alveolar progenitor populations”*

5. Line 160: Should there be a reference to Figure 1H here? Figure 1H is seemingly not currently referenced
anywhere in the paper as of now.

*A: Indeed, thank you for pointing this out. This is now corrected*

6. Line 546: “canonical tube” – the authors likely meant to say “conical tube”

*A: Good catch! Corrected*

7. In the Methods section “Making single cell suspensions and passaging organoids”, please clarify what types
of suspension culture plates were used and how many “drops” of cells/BME were included on each plate (line
555), as these are important details for recapitulation of these methods.

*A: We added these details to the methods section.*

Reviewer #3 (Remarks on Protocol(s)):

Overall, the authors provide the detailed methods for this study. Please also see our comments to the authors.

Dear Hans, dear Antonella,

Thank you again for transferring your amended manuscript (EMBOJ-2025-123201-T) to The EMBO Journal, together with a point-by-point response to the issues raised during peer-review at the previous venue. Please accept my apologies for the unusual protraction in getting back to you due to delayed expert input and detailed discussion in the editorial team. Your revised study was sent back to the referees for their scientific reassessment, and we have received detailed re-reports from all of them, which I enclose below. As you will see, the reviewers state that the work has been reasonably enhanced by the revisions and they can now support publication at the EMBO Journal, pending minor revision.

Thus, we are pleased to inform you that your manuscript has been accepted in principle for publication in The EMBO Journal.

Please carefully consider the remaining minor points raised by referees #2 and #3 by adding complementary results and/or amending the manuscript text - discussion of the results and related literature - where appropriate.

Also, we now need you to take care of a number of minor issues related to formatting and data annotation, as detailed below.

Please submit a revised version of the manuscript using the link enclosed below.

As you might remember from previous experience, every paper at the EMBO Journal includes a 'Synopsis', displayed on the html and freely accessible to all readers. The synopsis includes a 'model' figure as well as 2-5 one-short-sentence bullet points that summarize the article. I would appreciate if you could provide this figure and the bullet points.

Finally, we have noted that the submitted version of your article is also posted on the preprint platform bioRxiv. We would appreciate if you could alert bioRxiv on the acceptance of this manuscript at The EMBO Journal in order to allow for an update of the entry status. Thank you in advance!

Thank you again for giving us the chance to consider your manuscript for The EMBO Journal, I look forward to hearing from you and receiving your final revised version of the manuscript.

Best regards,

Daniel

>> Please add up to five keywords to your study.

>> Author Contributions: Remove the author contributions information from the manuscript text. Note that CRediT has replaced the traditional author contributions section as of now because it offers a systematic machine-readable author contributions format that allows for more effective research assessment. and use the free text boxes beneath each contributing author's name to add specific details on the author's contribution.

More information is available in our guide to authors.
<https://www.embopress.org/page/journal/14602075/authorguide>

>> Please correct the order of the manuscript sections to: Abstract / Introduction / Results / Discussion / Methods / Data Availability / Acknowledgements / Disclosure and Competing Interests Statement / References / Figure Legends

>> Adjust the title of the 'Summary' section to 'Abstract'.

>> Figure captions are to be included in main manuscript file (a separate file is not needed).

>> References: please adjust reference format to EMBO Journal format, 10 authors et al. DOIs should only be used for preprints and datasets that have not been published yet

>> Funding: check the following duplicate grant entries in our online system: Dutch Research Counsel (NWO) and Longfonds (Lung Foundation Netherlands).

>> Figure callouts: currently missing for Figure 3I; there are callouts for Fig. S2D and S2E, S4A/C, S5C, however no such figures. (supplementary figures are not allowed; only EV figures or Appendix figures).

>> Dataset EV legends: datasets are complex Excel tables that have multiple rows, columns and sheets but currently cannot be properly converted to PDF; the nomenclature should be adjusted to 'Dataset EV1', etc. (Tables EV1-EV4 to be uploaded as Datasets, callouts to be adjusted in manuscript. Check Table EV5 as well). Captions are to be included in the file itself as a separate sheet.

>> 'Disclosure and Competing Interests Statement'. Please add a statement: 'H.C. is a member of the EMBO Journal editorial advisory board.'

>> Data availability section: remove the referee token and make sure the GEO dataset is made publicly accessible.

>> Consider additional changes and comments from our production team as indicated below:

Figure Legends - Comments

- Please note that the exact p values are not provided in the legend of figure 3I"
- Please note that information related to n is missing in the legends of figures 1B, G, H, J; 2B, D, E, F; 3I, 4F, 5A-G, I, J
- Please note that the error bars are not defined in the legend of figure 2E "

Further information is available in our Guide For Authors: <https://link.springer.com/journal/44318/submission-guidelines>

Referee #1:

The manuscript highlights a modified protocol of generating and maintaining alveolar organoids from isolated epithelial cells that does not rely on previous Ats cell definition (HTII-280) which has likely excluded defined progenitor cell subsets, derived from primary human lobectomy material. The authors use this model to demonstrate a role of recombinant IFNy on organoid cell population behaviour, revealing different actions toward AT2 vs AT1, which might be relevant for lung regeneration.

The manuscript has now been improved by different aspects, some mentioned here:

In the revised version of the manuscript, the authors now add data to better highlight the novelty over previously published protocols that were based on HTII-280 AT2 cell selection. It is also stated now more clearly what the further improvements over published models are.

Also, the introduction is no longer focused on COPD, but in regeneration in general, which is appreciated as the organoid model is not a COPD model per se. In addition, information on donors (particularly regarding preexisting chronic lung disease/COPD) has been added, and information on statistics in the figure legends.

As to the heterogeneity of their cultures, the authors state that this is a strength of their model, however, when used to model defined aspects of lung regeneration, it might also pose difficulties. In a final version of their ms, the authors might at least state this limitation in comparison to iPSC-derived models.

Referee #2:

I was pleased to receive the resubmitted version of this manuscript entitled "Interferon- γ selectively promotes survival of alveolar progenitor cells in a human organoid model" which is now being considered for publication in EMBO J.

The authors describe the development of a novel approach to adult human alveolar epithelial culture modelling with particular emphasis on an inclusive approach to cell isolation to ensure the capture of all potential alveolar progenitors and defining media components required for AT2-AT1 differentiation. Having undertaken detailed characterisation, they then use this model to identify the differing role of Interferon- γ on AT1 and AT2 survival. Their findings provide a transcriptional analysis-focused dataset which will nonetheless provide a useful platform for further refinement and functional analysis of the cell types that are / can be derived using this approach.

The resubmitted version has been significantly amended to reflect the reviewers' comments and, in my view, is strengthened. This is particularly the case for the first part of the results section in which they provide further experimental evidence to support their stance that using HTII-280 sorting from primary tissue excludes a significant AT2 population which would otherwise be able to form viable alveolar organoids. The rewording of this section also improves readability for the non-expert.

I just have a few minor observations, then would support acceptance of the manuscript once addressed:

1. The ability to define AT2 (C6-) and secAT2 (C6+) transcriptionally and then sort them using a cell surface marker is a promising development. However, in this manuscript at least, these two populations behave in a very similar manner when placed in differentiation medium and/or challenged with IFN γ . The discussion would benefit from a comment on this to highlight to the reader the potential importance of being able to separate the populations despite there being no significant functional differences demonstrated in this work.
2. There are several places in which the formatting and font of the additional references need aligning with the rest of the manuscript.
3. On page 8, line 279 and 281 I think the figure reference should be EV3G, not EV3F.
4. Typo page 11 line 389 (figure EVV5E).

Referee #3:

Summary

The authors present an improved protocol for generating human primary distal lung organoids, along with detailed phenotypic analyses that add an additional layer to our understanding of heterogeneity within the human distal lung epithelium. They use negative selection for NGFR to exclude airway basal cells while retaining distal airway populations that may contribute to, or serve as progenitors for, alveolar epithelial cells. The authors then substitute EGF with HRG1, which enhances organoid expansion. Using this system, they show that IFN- γ is cytotoxic to AT1 cells while promoting the survival and proliferation of AT2 cells. Overall, the manuscript provides careful characterization and would be of interest to the field.

In general, the revised manuscript and the authors' responses address the major concerns raised previously, including issues related to the initial sorting strategy, starting cell populations, and organoid heterogeneity, particularly with respect to secAT2 cells. However, a few remaining points merit further consideration.

Major comments:

1. Reviewer Figures A-B are critical for readers to understand the initial culture inputs at the p0->p1 transition and should be included in the supplemental material, particularly in the absence of more comprehensive profiling (e.g., scRNA-seq of NGFR-cells from unsorted/plated distal lung versus HTII-280+ sorted populations after 14 days of culture under these conditions), which would provide a clearer assessment of starting cell heterogeneity but may not be available. In addition, the qPCR data show that in one of the two donors screened, KRT5 and TP63 are expressed at relatively high levels in the NGFR-HTII-280- population, suggesting that a subset of basal-like cells not marked by NGFR may persist in the starting population. This could potentially explain the presence of KRT5-SCGB1A1-SFTPC- organoids in these cultures (lines 137-138), possibly reflecting basal cell-derived outgrowths that have downregulated KRT5 and are restrained by the media conditions. While the optimized organoid system benefits from incorporating diverse cell types and states that may contribute to alveolar differentiation, the presence of heterogeneous airway populations in the initial cultures complicates interpretation. Without single-cell-level tracking or lineage-resolved analyses, some of the authors' conclusions are difficult to fully support. The authors should more clearly acknowledge this critical limitation in the Discussion and, where appropriate, temper their conclusions accordingly.

2. Further aiding reader understanding of this heterogeneity as it relates to SFTPC+ organoids compared to the KRT5-SCGB1A1-SFTPC- organoids (lines 137-138), it would be highly useful to include some quantification of the ratio of these types of organoids and its stability over early vs. late passages.

3. Fig. EV1D: The manuscript text describing the reanalyzed scRNA-seq data from Sikkema et al. 2023 (lines 121-123) states that "HRG/NRG1 and its receptors ERBB3 and ERBB4 are more highly expressed in lung alveolar cells than their family members EGF and EGFR (Fig. EV1D) and HRG1 has a positive effect on the growth and survival of AOs." However, unless the dot plot is mislabeled, this does not appear to be supported by the figure. In particular, ERBB4 shows low "Percent Expressed" in AT1 and AT2 cells (with possibly higher "Average Expression" in AT2 cells), and ERBB4 also appears low in the "Secretory" population that the authors may argue contains distal airway progenitors relevant to their culture strategy. In addition, the wording is confusing and potentially inaccurate in implying that EGF and EGFR are "family members" of ERBB3 and ERBB4; it would be important to clearly distinguish ligands (e.g., HRG/NRG1, EGF) from receptors (ERBB family members, including EGFR/ERBB1, ERBB2, ERBB3, ERBB4). The authors should revise this statement to accurately reflect the EV1D data, clarify which "lung alveolar cells" they are referring to (e.g., AT2 only vs. broader alveolar populations), and more precisely describe receptor versus ligand expression. It may also be preferable to emphasize the presence of ERBB3/ERBB4 (even if ERBB4 is low) as rationale for investigating HRG1, rather than overstating relative expression levels, ideally supported by ERBB3/ERBB4 expression in the authors' own organoid scRNA-seq data (e.g., Reviewer Fig. D).

4. The data presented in Reviewer Fig. E, concerning the differentiation potential of CEACAM6+ versus CEACAM6- organoids, are important for readers to fully appreciate the distinct behaviors of these cell populations in culture. These results should therefore be included in the Supplemental Figs.

5. Line 125, this appears to reference the wrong figure - should be Fig. EV1C.

A: We thank the reviewers for their comments and their approval of the revised version of the manuscript. We addressed any remaining concerns below.

Referee #1:

The manuscript highlights a modified protocol of generating and maintaining alveolar organoids from isolated epithelial cells that does not rely on previous Ats cell definition (HTII-280) which has likely excluded defined progenitor cell subsets, derived from primary human lobectomy material. The authors use this model to demonstrate a role of recombinant IFN γ on organoid cell population behaviour, revealing different actions toward AT2 vs AT1, which might be relevant for lung regeneration.

The manuscript has now been improved by different aspects, some mentioned here:

In the revised version of the manuscript, the authors now add data to better highlight the novelty over previously published protocols that were based on HTII-280 AT2 cell selection. It is also stated now more clearly what the further improvements over published models are.

Also, the introduction is no longer focused on COPD, but in regeneration in general, which is appreciated as the organoid model is not a COPD model per se. In addition, information on donors (particularly regarding preexisting chronic lung disease/COPD) has been added, and information on statistics in the figure legends.

As to the heterogeneity of their cultures, the authors state that this is a strength of their model, however, when used to model defined aspects of lung regeneration, it might also pose difficulties. In a final version of their ms, the authors might at least state this limitation in comparison to iPSC-derived models.

A: We thank the reviewer for their suggestions and are pleased to hear that we have improved the manuscript to be accepted for publication. To satisfy the remaining comment of the reviewer, we have added the following paragraph to the discussion: "It should be noted that we observed marked heterogeneity between organoid lines in both marker expression and the magnitude of responses to stimuli such as IFN- γ . This variability likely reflects differences between donors and is therefore expected when working with primary human material. Organoid models derived from adult primary cells, which represent a more mature cellular state than fetal or iPSC-derived lung cells, are particularly valuable for modeling age-related lung diseases and for identifying therapeutic strategies to enhance alveolar regeneration in adults. At the same time, the intrinsic heterogeneity between primary cells from different donors can pose challenges and represents a limitation of the system. Elucidating the sources of these heterogeneous responses may provide insight into patient-to-patient variability and ultimately help inform patient stratification in the future."

Referee #2:

I was pleased to receive the resubmitted version of this manuscript entitled "Interferon- γ selectively promotes survival of alveolar progenitor cells in a human organoid model" which is now being considered for publication in EMBO J.

The authors describe the development of a novel approach to adult human alveolar epithelial culture modelling with particular emphasis on an inclusive approach to cell isolation to ensure the capture of all

potential alveolar progenitors and defining media components required for AT2-AT1 differentiation. Having undertaken detailed characterisation, they then use this model to identify the differing role of Interferon- γ on AT1 and AT2 survival. Their findings provide a transcriptional analysis-focused dataset which will nonetheless provide a useful platform for further refinement and functional analysis of the cell types that are / can be derived using this approach.

The resubmitted version has been significantly amended to reflect the reviewers' comments and, in my view, is strengthened. This is particularly the case for the first part of the results section in which they provide further experimental evidence to support their stance that using HTII-280 sorting from primary tissue excludes a significant AT2 population which would otherwise be able to form viable alveolar organoids. The rewording of this section also improves readability for the non-expert.

A: We are grateful for the reviewer's suggestions which helped us improved the manuscript.

I just have a few minor observations, then would support acceptance of the manuscript once addressed:

1. The ability to define AT2 (C6-) and secAT2 (C6+) transcriptionally and then sort them using a cell surface marker is a promising development. However, in this manuscript at least, these two populations behave in a very similar manner when placed in differentiation medium and/or challenged with IFN γ . The discussion would benefit from a comment on this to highlight to the reader the potential importance of being able to separate the populations despite there being no significant functional differences demonstrated in this work.

A: We agree with this comment and added the following paragraph to the discussion: "Although we identified transcriptional differences between the AT2 and secAT2 populations and showed that they can be enriched using a C6-based sorting strategy, we have not yet observed clear functional differences between the two populations. Future studies should therefore assess their long-term self-renewal capacity, engraftment potential, and ability to transdifferentiate toward airway epithelial fates. Notably, the co-expression of multiple lineage markers may signify enhanced cellular plasticity and a greater potential to differentiate into diverse lung epithelial cell types."

2. There are several places in which the formatting and font of the additional references need aligning with the rest of the manuscript.

A: We have corrected the formatting.

3. On page 8, line 279 and 281 I think the figure reference should be EV3G, not EV3F.

A: We have made this correction.

4. Typo page 11 line 389 (figure EVV5E).

A: We fixed this typo.

Referee #3:

Summary

The authors present an improved protocol for generating human primary distal lung organoids, along with detailed phenotypic analyses that add an additional layer to our understanding of heterogeneity within the human distal lung epithelium. They use negative selection for NGFR to exclude airway basal cells while retaining distal airway populations that may contribute to, or serve as progenitors for, alveolar epithelial cells. The authors then substitute EGF with HRG1, which enhances organoid expansion. Using this system, they show that IFN- γ is cytotoxic to AT1 cells while promoting the survival and proliferation of AT2 cells. Overall, the manuscript provides careful characterization and would be of interest to the field.

In general, the revised manuscript and the authors' responses address the major concerns raised previously, including issues related to the initial sorting strategy, starting cell populations, and organoid heterogeneity, particularly with respect to secAT2 cells. However, a few remaining points merit further consideration.

Major comments:

1. Reviewer Figures A-B are critical for readers to understand the initial culture inputs at the p0->p1 transition and should be included in the supplemental material, particularly in the absence of more comprehensive profiling (e.g., scRNA-seq of NGFR⁻ cells from unsorted/plated distal lung versus HTII-280⁺ sorted populations after 14 days of culture under these conditions), which would provide a clearer assessment of starting cell heterogeneity but may not be available. In addition, the qPCR data show that in one of the two donors screened, KRT5 and TP63 are expressed at relatively high levels in the NGFR⁻ HTII-280⁻ population, suggesting that a subset of basal-like cells not marked by NGFR may persist in the starting population. This could potentially explain the presence of KRT5-SCGB1A1-SFTPC⁻ organoids in these cultures (lines 137-138), possibly reflecting basal cell-derived outgrowths that have downregulated KRT5 and are restrained by the media conditions. While the optimized organoid system benefits from incorporating diverse cell types and states that may contribute to alveolar differentiation, the presence of heterogeneous airway populations in the initial cultures complicates interpretation. Without single-cell-level tracking or lineage-resolved analyses, some of the authors' conclusions are difficult to fully support. The authors should more clearly acknowledge this critical limitation in the Discussion and, where appropriate, temper their conclusions accordingly.

A: We thank the reviewer for this comment. We do not feel comfortable adding reviewer figure B to the manuscript, as we only have data of 2 lungs for this analysis. Moreover, we do acknowledge that the starting population for organoids is heterogenous, but our scRNA Seq shows that there are no basal-like cells in our system. We do not think that we overstated any interpretations.

2. Further aiding reader understanding of this heterogeneity as it relates to SFTPC⁺ organoids compared to the KRT5-SCGB1A1-SFTPC⁻ organoids (lines 137-138), it would be highly useful to include some quantification of the ratio of these types of organoids and its stability over early vs. late passages.

A: We agree that this information could be useful for some research questions, but this data would not directly add to the conclusions presented in this manuscript.

3. Fig. EV1D: The manuscript text describing the reanalyzed scRNA-seq data from Sikkema et al. 2023 (lines 121-123) states that "HRG/NRG1 and its receptors ERBB3 and ERBB4 are more highly expressed in lung alveolar cells than their family members EGF and EGFR (Fig. EV1D) and HRG1 has a positive effect on the growth and survival of AOs." However, unless the dot plot is mislabeled, this does not

appear to be supported by the figure. In particular, ERBB4 shows low "Percent Expressed" in AT1 and AT2 cells (with possibly higher "Average Expression" in AT2 cells), and ERBB4 also appears low in the "Secretory" population that the authors may argue contains distal airway progenitors relevant to their culture strategy. In addition, the wording is confusing and potentially inaccurate in implying that EGF and EGFR are "family members" of ERBB3 and ERBB4; it would be important to clearly distinguish ligands (e.g., HRG/HRG1, EGF) from receptors (ERBB family members, including EGFR/ERBB1, ERBB2, ERBB3, ERBB4). The authors should revise this statement to accurately reflect the EV1D data, clarify which "lung alveolar cells" they are referring to (e.g., AT2 only vs. broader alveolar populations), and more precisely describe receptor versus ligand expression. It may also be preferable to emphasize the presence of ERBB3/ERBB4 (even if ERBB4 is low) as rationale for investigating HRG1, rather than overstating relative expression levels, ideally supported by ERBB3/ERBB4 expression in the authors' own organoid scRNA-seq data (e.g., Reviewer Fig. D).

A: We now rephrased this sections according to the reviewer's suggestions to: "HRG1/HRG1 and its receptors ERBB3 and ERBB4 are expressed in lung alveolar cells, similar to EGF and its receptor EGFR/ERBB1 (figure EV1D)."

4. The data presented in Reviewer Fig. E, concerning the differentiation potential of CEACAM6+ versus CEACAM6- organoids, are important for readers to fully appreciate the distinct behaviors of these cell populations in culture. These results should therefore be included in the Supplemental Figs.

A: We now added reviewer figure E as figure EV2F to the manuscript.

5. Line 125, this appears to reference the wrong figure - should be Fig. EV1C.

A: We corrected this mistake.

Figure for reviewers removed

Dear Hans, dear Antonella,

Thank you for submitting the revised version of your manuscript. I have now evaluated your amended manuscript and concluded that the remaining minor concerns have been sufficiently addressed.

I am thus pleased to inform you that your manuscript has been accepted for publication in the EMBO Journal.

Best regards,

Daniel

Daniel Klimmeck, PhD
Senior Editor
The EMBO Journal
EMBO
Postfach 1022-40
Meyerhofstrasse 1
D-69117 Heidelberg
contact@embojournal.org

Please note that it is The EMBO Journal policy for the transcript of the editorial process (containing referee reports and your response letters) to be published as an online supplement to each paper. If you should prefer removal of any referee-only figures included in the point-by-point response(s), e.g. because they may still be used for future publication or because they have been reproduced from published work by others, please do let us know immediately via response email.

More information is available here: <https://link.springer.com/partners/embo-press/editorial-policies#Peer%20review>